# THOMPSON SAMPLING ALGORITHM FOR STOCHASTIC GAMES

## ABSTRACT

We study a stochastic differential game with $N$ competitive players in a linear-quadratic framework with ergodic cost, where $d$-dimensional diffusion processes govern the state dynamics with an unknown common drift (matrix). Assuming a Gaussian prior on the drift, we use filtering techniques to update its posterior estimates. Based on these estimates, we propose a Thompson-sampling-based algorithm with dynamic episode lengths to approximate strategies. We show that the Bayesian regret for each player has an error bound of order $O(\sqrt{T \log(T)})$, where $T$ is the time-horizon, independent of the number of players. This implies that average regret per unit time goes to zero. Finally, we prove that the algorithm results in a Nash equilibrium.

## 1 INTRODUCTION

We consider a non-zero-sum stochastic differential game (SDG) with $N$ competitive players, where each player's state evolves as a linear diffusion driven by a $d$-dimensional Brownian motion. Players do not interact directly through actions or dynamics; coupling arises solely through the cost functionals. Each player minimizes an individual ergodic quadratic cost by choosing a strategy that affects the drift, penalizing deviations of the common state from a fixed reference. A key feature is that all players share a common but unknown drift matrix. Each player observes their own state, learns the drift, and adapts their control. We characterize a Nash equilibrium in this partially observed game and develop a numerical scheme to approximate the players' equilibrium strategies.

Stochastic differential games (SDGs) provide a continuous-time framework of modeling strategic interactions among multiple players whose states are controlled diffusion processes. The study of differential games dates back to the seminal work of Isaacs (1965). For the stochastic extension and its PDE treatments, see for example Friedman (1975). SDGs arise naturally in many real-world applications, where multiple agents interact through their dynamics and cost structures to optimize individual or collective objectives. In large systems, players often interact via the average behavior of the population, leading to mean field game settings, which have been extensively studied. Notable works include Lasry & Lions (2006; 2007), Huang et al. (2006), Carmona & Delarue (2018a;b), and Bensoussan et al. (2016); Li & Zhang (2008) for linear-quadratic settings.

In our SDG, players are coupled via the state vector, not empirical distributions, so we do not use mean field game theory. Our formulation follows Bardi & Priuli (2014), where each player observes their own independent dynamics with known drift parameters. The authors derive a system of $N$ nonlinear PDEs of the HJB-KFP type, providing conditions for existence and uniqueness in quadratic value functions, and Nash equilibria in the form of affine feedbacks. In our model, the drift parameter is common but unknown, introducing a learning component. Since players observe only their own state trajectories and use open-loop strategies, the structure retains a form of independence, allowing us to build on techniques from Bardi & Priuli (2014).

In most of the literature analyzing SDGs, the players have the full distributional assumption of the underlying system. However, such assumptions are overly idealized, as there is often uncertainty not only in the system dynamics but also in the model parameters themselves. This demands that players learn about the system over time before they act. Such learning is commonly incorporated through stochastic filtering theory. The statistical study of unknown parameters of the underlying system dates back to the foundational work Wald (1947) and Wald & Wolfowitz (1950), whose

proposed methods are still widely applied across various engineering domains today. In our work, we assume that the players share a common unknown prior, but they only observe their own state process. We adopt a Bayesian framework and use standard methods in continuous-time stochastic filtering theory. For the general tools and treatment of such problems, we refer the reader to Liptser & Shiryayev (1978). We assume a Gaussian prior as it is the conjugate prior for our model and thus ensures tractable posterior updates. In line with sequential analysis methods, we introduce appropriate stopping criteria that capture the tradeoff between the time cost and the accuracy of the posterior estimate. Application-wise, the common drift feature captures public exogenous signals (e.g., macroeconomic/regulatory), while heterogeneity enters via player-specific dynamics, priors, and noise. Agents update their beliefs and act independently without exchanging private data. There are many related potential applications, and we highlight one: the electricity consumers aim to minimize their long-term average electricity cost, where the underlying is the individually assigned electricity price related to their consumption and the rate of mean-reversion (government-provided information). The decision makers filter the common information independently and solve their own optimization problems. A related setup appears in Aïd et al. (2025), where our example aligns with the receiver's perspective.

In this paper, we propose a numerical scheme to accommodate the learning feature and approximate the equilibrium strategies of each player based on the Thompson Sampling (TS) algorithm. TS is a natural Bayesian approach where one begins with a prior belief over the unknown parameters and, at each decision point, updates this belief to form a posterior. The algorithm then samples from the posterior distribution of the parameters for each action and selects the action that maximizes the expected reward based on these samples. It has been shown to perform well empirically across a wide range of problems; see, for example, Chapelle & Li (2011); Scott (2010); Agrawal & Goyal (2012). While TS is typically formulated in a discrete-time setting, we adopt a continuous-time framework to leverage the results in Bardi & Priuli (2014). This avoids the need to derive discrete-time analogs of the problem in consideration under full information. In recent years, a growing body of work has focused on the theoretical analysis of TS in different settings and studied their expected Bayesian regret bounds. For example, see Ouyang et al. (2017; 2020), and Gagrani et al. (2021). Our algorithm and regret analysis are motivated by the approach in the last three references, where the authors propose adaptive episode lengths to balance the trade-off between learning the unknown state and minimizing the long-term average cost. They establish a regret bound depending on the number of types of players, and of order $O(\sqrt{T \log(T)})$, where $T$ is the time horizon of the game. Notably, our assumption on the underlying process is much weaker, as we do not assume the independence of the players, and yet as one of our main results, we still achieve a regret bound of the same order in $T$, and independent of the number of players. As we elaborate later, this is made possible by the specific information structure of our formulation. The regret upper-bound of order $O(\sqrt{T \log(T)})$ we obtain matches the best known upper-bound for LQ systems. For a broad literature that achieves this result, see Faradonbeh et al. (2018a;b); Abbasi-Yadkori & Szepesvári (2011); Ibrahimi et al. (2012); Ouyang et al. (2017; 2020); Abeille & Lazaric (2018); Cohen et al. (2019); Shirani Faradonbeh et al. (2022). By contrast, obtaining lower bounds is generally harder. Some recent works have established a lower bound of order $O(\sqrt{T})$. See Simchowitz & Foster (2020); Ziemann & Sandberg (2024); Tsiamis et al. (2022). In our work, we adopt a vanilla TS algorithm. We want to point out that some other randomized exploration strategies that are studied in the recent years, such as Ishfaq et al. (2023); Hsu et al. (2024); Ishfaq et al. (2021); Lee & Oh (2024), can be adapted to our problem as well.

Another key result is that the TS algorithm yields a Nash equilibrium. In other words, no player would unilaterally deviate from their TS-proposed strategy, assuming all others continue following theirs, as it does not reduce their ergodic cost. What makes this particularly innovative is that, unlike much of the existing literature focused on optimization, our approach centers on establishing equilibrium behavior in a multi-agent setting, which presents distinct analytical challenges, offering a novel contribution. The main step is to construct a coupling between the state dynamics under the TS setting and those under full information, and compare their evolution over time. We show that the time-averaged difference between the two processes vanishes. With a few additional steps for the full-information model, we demonstrate that the dynamics under the TS algorithm inherit its key features from the full-information process. To the best of our knowledge, our work gives one of the first approaches for partial information in a game setting that yields an equilibrium and a sublinear regret bound, extending beyond single-agent control problems and full-information stochastic differential games.

From the perspective of reinforcement learning, the SDG we study can be viewed as a continuous-time multi-agent decision problem. In this light, our contribution is to design and analyze a learning algorithm in this rich environment: we propose a TS procedure and show a Bayesian regret bound together with an equilibrium property of the learned strategies. The SDG framework allows us to model strategic interactions and partial information in continuous time while still obtaining explicit characterizations of equilibrium policies. It provides theoretically grounded models and algorithms beyond the standard discrete-time, fully observed settings. While much of the MARL literature focuses on discrete-time Markov games, rigorous guarantees for continuous-time and partially observed multi-agent environments are much less explored. Our framework helps bridge this gap by providing constructive learning guarantees and explicit equilibrium characterizations within a continuous-time SDG model.

The paper is organized as follows. Section 2 formulates the $N$-player game under full information and discusses equilibrium strategies. Section 3 covers incomplete information and learning, introduces a TS-based algorithm, and presents our main results: Theorem 3.1 (Bayesian regret bound) and Theorem 3.2 (Nash equilibrium). Section 4 reports numerical experiments, and Section 5 concludes. Technical proofs and examples are in the appendices.

## 2 THE ERGODIC $N$-PLAYER GAME WITH FULL INFORMATION

### 2.1 N-PLAYER ERGODIC GAME

In this section, we summarize the results from Bardi & Priuli (2014) for ergodic $N$-player games with open-loop controls. In this setup, we assume that all players are fully informed about the model's parameters. This serves as the benchmark for the model with unknown parameters. Consider a probability space $(\Omega, \mathcal{F}, \mathbb{P})$ that supports $N$ independent $d$-dimensional standard Brownian motions, denoted as $(W_t^i)_{t \geq 0}$, $i \in [N] := \{1, 2, \ldots, N\}$. For each $i \in [N]$, denote by $(\mathcal{F}_t^i)_{t \geq 0}$ the augmentation of the filtration generated by $W^i$.

Consider a game with open-loop controls between $N$ players. We use $X_t^i \in \mathbb{R}^d$ and $\alpha_t^i \in \mathbb{R}^d$ to denote, respectively, the state and action of player $i$ at time $t$. The system starts at a given initial state[1] $\boldsymbol{X}_0 = (X_0^i)_{i \in [N]} \in \mathbb{R}^{Nd}$. The state dynamics of the players are given by

$$dX_t^i = (AX_t^i - \alpha_t^i)dt + \sigma^i dW_t^i, \qquad i = 1, \ldots, N, \tag{2.1}$$

where $A, \sigma^i \in \mathbb{R}^{d \times d}$ are given matrices with $\det(\sigma^i) \neq 0$,

**Definition 2.1.** *A strategy $(\alpha_t^i)_{t \geq 0}$ is called an* admissible strategy *if it is a process adapted to $\mathcal{F}_t^i$, the expectation $\mathbb{E}^i[\|\alpha_t^i\|^2]$ is bounded for any $t \geq 0$, and the corresponding state dynamics $X_t^i$ satisfy:*

- *For any $t \geq 0$, $\mathbb{E}^i[\|X_t^i\|^2] < \infty$.*
- *$X_t^i$ is ergodic in the following sense: there exists a probability measure $\pi^i$ on $\mathbb{R}^d$ such that $\int_{\mathbb{R}^d} \|x\|^2 d\pi^i < \infty$ and*

$$\lim_{T \to \infty} \frac{1}{T} \mathbb{E}^i\Big[ \int_0^T h(X_t^i)dt \Big] = \int_{\mathbb{R}^d} h(x)d\pi^i(x) \tag{2.2}$$

*locally uniformly w.r.t. the initial state $X_0^i$ for all polynomial functions of degree at most 2.*

Denote by $\mathcal{A}^i$ the set of all admissible strategies of player $i$. Given an initial state $\boldsymbol{X}_0 = (X_0^1, \ldots, X_0^N) \in \mathbb{R}^{Nd}$ and an admissible strategy profile $\boldsymbol{\alpha} = (\alpha^1, \ldots, \alpha^n) \in \prod_{i=1}^n \mathcal{A}^i$, the ergodic cost for player $i$ is $J^i : \mathbb{R}^{Nd} \times \mathbb{R}^{Nd} \to [0, \infty)$, defined as

$$J^i(\boldsymbol{X}_0, \boldsymbol{\alpha}) = \liminf_{T \to \infty} \frac{1}{T} \mathbb{E}^i\Big[ \int_0^T F^i(\boldsymbol{X}_t, \alpha_t^i) \, dt \Big], \tag{2.3}$$

where the state dynamics $\boldsymbol{X}_t$ depend on all players' strategies $\alpha^1, \alpha^2, \ldots, \alpha^n$. The running cost $F^i : \mathbb{R}^{Nd} \times \mathbb{R}^d \to [0, \infty)$ is given by

$$F^i(\boldsymbol{x}, a) := \tilde{F}^i(\boldsymbol{x}) + \frac{1}{2} a^T R^i a \qquad \text{where} \qquad \tilde{F}^i(\boldsymbol{x}) := (\boldsymbol{x} - \overline{\boldsymbol{x}}_i)^T \mathcal{Q}^i (\boldsymbol{x} - \overline{\boldsymbol{x}}_i), \tag{2.4}$$

---

[1]We assume a deterministic initial state, but the results extend readily to random initial states that are not necessarily independent across players, as in ergodic settings, the initial state does not affect the analysis.

and for each $i \in [N], \overline{\boldsymbol{x}}_i \in \mathbb{R}^{Nd}$ is a given reference position for the players. Additionally, $R^i \in \mathbb{R}^{d \times d}$ and $\mathcal{Q}^i \in \mathbb{R}^{Nd \times Nd}$ are given matrices.

Given an admissible strategy profile $\boldsymbol{\alpha} = (\alpha^1, \ldots, \alpha^N) \in \prod_{i=1}^n \mathcal{A}^i$ and an individual admissible strategy $\hat{\alpha} \in \mathcal{A}^i$, denote $[\boldsymbol{\alpha}^{-i}; \hat{\alpha}] := [\alpha^1, \ldots, \alpha^{i-1}, \hat{\alpha}, \alpha^{i+1}, \ldots, \alpha^N]$ as the strategy profile generated from $\boldsymbol{\alpha}$ by replacing $\alpha^i$ with $\hat{\alpha}$.

**Definition 2.2.** *Given an initial state $\boldsymbol{X}_0 \in \mathbb{R}^{Nd}$, a strategy profile $\boldsymbol{\alpha} = (\alpha^1, \ldots, \alpha^N)$ with $\alpha^i \in \mathcal{A}$ for any $i \in [N]$, is called* Nash equilibrium *if for every $i \in [N]$ and every $\hat{\alpha} \in \mathcal{A}^i$*
$$J^i(\boldsymbol{X}_0, \boldsymbol{\alpha}) \leq J^i(\boldsymbol{X}_0, [\boldsymbol{\alpha}^{-i}; \hat{\alpha}]).$$

The cost is defined as a long-run cost. Leveraging ergodicity, we now define a form of the expected cost under the stationary distribution that the players in $[N] \setminus \{i\}$ eventually converge to. For this, denote by $m^j, j \in [N]$ a density of a distribution on $\mathbb{R}^d$ as well as
$$\boldsymbol{m}^{-i} := (m^1, \ldots, m^{i-1}, m^{i+1}, \ldots, m^N),$$
and set the function
$$f^i(x, \alpha; \boldsymbol{m}^{-i}) := \int_{\mathbb{R}^{(N-1)d}} F^i(\xi^1, \ldots, \xi^{i-1}, x, \xi^{i+1}, \ldots, \xi^N, \alpha) \prod_{j \neq i} dm^j(\xi^j),$$
$$\tilde{f}^i(x; \boldsymbol{m}^{-i}) := \int_{\mathbb{R}^{(N-1)d}} \tilde{F}^i(\xi^1, \ldots, \xi^{i-1}, x, \xi^{i+1}, \ldots, \xi^N) \prod_{j \neq i} dm^j(\xi^j),$$
and $\varsigma^i := \frac{1}{2}(\sigma^i)(\sigma^i)^T \in \mathbb{R}^{d \times d}$ for $i \in [N]$. Also, set the *Hamiltonian* $H^i : \mathbb{R}^d \times \mathbb{R}^d \to \mathbb{R}$ as
$$H^i(x, p) := \max_{a \in \mathbb{R}^d} \left\{ -\frac{1}{2} a^T R^i a - p^T (Ax - a) \right\} = \frac{1}{2} p^T (R^i)^{-1} p - p^T Ax.$$

The system of HJB-KFP equations associated with the game (2.3) is given by
$$\begin{cases} -\text{tr}\left(\varsigma^i D^2 v^i(x)\right) + H^i(x, \nabla v^i(x)) + \lambda^i = \tilde{f}^i(x; \boldsymbol{m}^{-i}), \\ -\text{tr}\left(\varsigma^i D^2 m^i(x)\right) - \text{div}\left(m^i(x) \frac{\partial H^i}{\partial p}(x, \nabla v^i(x))\right) = 0, \qquad i \in [N]. \\ \int_{\mathbb{R}^d} m^i(x)\, dx = 1, \quad m^i > 0, \end{cases} \tag{2.5}$$

For fixed stationary $\boldsymbol{m}^{-i}$ of the other players, the first equation in (2.5) is the stationary Hamiltonian-Jacobi-Bellman equation for player $i$, which characterizes the relative value function $v^i$ (measures how much additional long-run cost is incurred when starting from state $x$ instead of the stationary distribution) and the optimal long-run average cost $\lambda^i$ when player $i$ uses the optimal feedback policy. The second equation is the Kolmogorov forward (Fokker-Planck) equation, which identifies $m^i$ as the stationary distribution of $X^i$ under the feedback policy.

The solution is $(\lambda^i, v^i, m^i)_{i=1}^N$, where $m^i$ is the density of the *stationary distribution* of the player $i$'s state, $\lambda^i \in \mathbb{R}$ is the *value* of the ergodic control problem from the point of view of player $i$, given the distribution vector $(m^1, \ldots, m^N)$; we refer to $v^i : \mathbb{R}^d \to \mathbb{R}$ as the *relative value* for player $i$. We denote by $\nabla v(x)$ the gradient of function $v$, and by $D^2 v(x)$ its Hessian matrix.

Define the matrix $\mathcal{B} \in \mathbb{R}^{Nd \times Nd}$ as a block matix:
$$\mathcal{B} := (\mathcal{B}_{ij})_{i,j=1,\ldots,N} \qquad \text{where} \qquad \mathcal{B}_{ij} := -Q_{ij}^i - \frac{1}{2} \delta_{ij} A^T R^i A \in \mathbb{R}^{d \times d}, \tag{2.6}$$
and $\delta_{ij}$ is the Kronecker delta. The matrices $Q_{jk}^i$ are $d \times d$ blocks of $\mathcal{Q}^i$, thus $Q_{ij}^i$ denotes the $(i, j)$-th $d \times d$ block of the matrix $\mathcal{Q}^i$. Define $\boldsymbol{p} = (p_1, \ldots p_n) \in \mathbb{R}^{Nd}$ with the $i$-th component
$$p_i := -\sum_{j=1}^N Q_{ij}^i \overline{x}_i^j \in \mathbb{R}^d. \tag{2.7}$$

Also, denote $[\mathcal{B}, \boldsymbol{p}] := (\mathcal{B}^1, \ldots, \mathcal{B}^{Nd}, \boldsymbol{p})$, where $\mathcal{B}^j$ are the columns of the matrix $\mathcal{B}$.

Throughout the paper, we make the following assumption, borrowed from Bardi & Priuli (2014).

**Assumption 2.1.** *The following conditions hold:*

*(A1)* $\forall i \in [N]$, *there exists a unique symmetric positive definite matrix $Y$ that solves the* algebraic Riccati equation
$$\frac{1}{2} Y \varsigma^i R^i \varsigma^i Y = \frac{1}{2} A^T R^i A + Q_{ii}^i. \tag{2.8}$$

*(A2) The block matrix $\mathcal{B}$ defined by (2.6) is invertible.*

*(A3) For any $i \in [N]$, $\sigma^i$ in (2.1) is an invertible matrix, $R^i$ in (2.4) and $Q_{ii}^i$ are real symmetric and positive definite matrices, and $\mathcal{Q}^i$ in (2.4) is a symmetric matrix.*

*(A4) For any $i \in [N]$, $\exists \rho^i > 0$, $\mathcal{Q}^i$ satisfies $\lambda_{\min}(Q_{ii}^i) - \sum_{j \neq i} \|Q_{ij}^i\| \geq \rho^i$, where $\lambda_{\min}(Q_{ii}^i)$ is the smallest eigenvalue of $Q_{ii}^i$.*

We provide two examples, one of a symmetric system and one of a nondefective system, in A.1

**Remark 2.1.** *Each $d \times d$ block $Q_{jk}^i$ represents the cost for player $i$ incurred by per unit of deviation of players $j$ and $k$ from their reference positions $\overline{x}_i^j$ and $\overline{x}_i^k$ respectively. The assumption $Q_{ii}^i > 0$ implies that player $i$'s own reference position $\overline{x}_i^i$ is a preferred position for the player. This condition simplifies calculations, but for the validity of Proposition 2.1 it can be relaxed to $Q_{ii}^i + A^T R^i A / 2 > 0$.*

**Proposition 2.1** (Theorem 3.1 in Bardi & Priuli (2014))**.** *Let Assumption 2.1 be in force. Then, the HJB-KFP equation system (2.5) admits a unique solution $(v^i, m^i, \lambda^i)_{i \in [N]}$ with a quadratic function $v^i$, and it is of the form*

$$v^i(x) = \tfrac{1}{2} x^T \Lambda^i x + (\rho^i)^T x, \qquad \lambda^i \in [0, \infty), \qquad m^i = \mathcal{N}(\eta^i, (\Upsilon^i)^{-1}), \tag{2.9}$$

*where $\Upsilon^i$ is the unique solution of the Riccati Equation (2.8) which is a real symmetric positive definite matrix, $\Lambda^i = R^i(\varsigma^i \Upsilon^i + A)$, $\eta = (\eta^1, \ldots, \eta^N)$ solves $\mathcal{B}\eta = \boldsymbol{p}$, $\rho^i = -R^i \varsigma^i \Upsilon^i \eta^i$, and*

$$\lambda^i = F_0^i - \tfrac{1}{2}(\eta^i)^T \Upsilon^i \varsigma^i R^i \varsigma^i \Upsilon^i \eta^i + \mathrm{tr}(\varsigma^i R^i \varsigma^i \Upsilon^i + \varsigma^i R^i A), \tag{2.10}$$

*where*

$$\begin{aligned}
F_0^i := {}& (\overline{\boldsymbol{x}}_i^j)^T Q_{ii}^i \overline{\boldsymbol{x}}_i^i - (\overline{\boldsymbol{x}}_i^j)^T \Big( \sum_{j \neq i} Q_{ij}^i (\eta^j - \overline{\boldsymbol{x}}_i^j) \Big) - \Big( \sum_{j \neq i} (\eta^j - \overline{\boldsymbol{x}}_i^j)^T Q_{ji}^j \Big) \overline{\boldsymbol{x}}_i^i \\
& + \sum_{j,k \neq i, j \neq k} (\eta^j - \overline{\boldsymbol{x}}_i^j)^T Q_{jk}^j (\eta^k - \overline{\boldsymbol{x}}_i^k) + \sum_{j \neq i} \Big( tr(Q_{jj}^j (\Sigma^i)^{-1}) + (\eta^j - \overline{\boldsymbol{x}}_i^j)^T Q_{jj}^j (\eta^j - \overline{\boldsymbol{x}}_i^j) \Big).
\end{aligned}$$

*Moreover, the affine feedback strategies*

$$\bar{a}^i(x; A) = (R^i)^{-1} \nabla v^i(x) = (\varsigma^i \Upsilon^i + A) x - \varsigma^i \Upsilon^i \eta^i, \qquad x \in \mathbb{R}^d, \ i \in [N], \tag{2.11}$$

*provide a Nash equilibrium strategy $\alpha_t^i := \bar{a}^i(X_t^i; A)$, $t \geq 0$, for all initial positions $\boldsymbol{X}_0 \in \mathbb{R}^{Nd}$, among the admissible strategies, and $J^i(\boldsymbol{X}_0, \boldsymbol{\alpha}) = \lambda^i$ for all $\boldsymbol{X}_0$ and all $i$.*

Plugging in the equilibrium strategy profile, we find that the dynamics of the players follow Ornstein–Uhlenbeck processes, governed by the stochastic differential equation:

$$\begin{aligned}
dX_t^i &= ((A - (R^i)^{-1}\Lambda^i)X_t^i - (R^i)^{-1}\rho^i)dt + \sigma^i dW_t^i \\
&= (-\varsigma^i \Upsilon^i X_t^i + \varsigma^i \Upsilon^i \eta^i)dt + \sigma^i dW_t^i, \qquad\qquad i = 1, \ldots, N, \tag{2.12}
\end{aligned}$$

and can be explicitly written as:

$$X_t^i = e^{-\varsigma^i \Upsilon^i t} X_0^i + (I - e^{-\varsigma^i \Upsilon^i t})\eta^i + \int_0^t e^{-\varsigma^i \Upsilon^i (t-s)} \sigma^i dW_s^i, \qquad t \geq 0. \tag{2.13}$$

## 3 LEARNING FOR THE $N$-PLAYER GAME

In this section, we build on the results of the previous section and consider the same model, with the key difference that players are now uncertain about the parameter $A$. We begin by setting up the model, then define a TS algorithm for the players, evaluate its regret, and verify that it constitutes an equilibrium. To address this, we expand the probability space to incorporate the players' prior distributions over the parameter $A$.

Consider a probability space $(\Omega, \mathcal{F}, \mathbb{P})$ supporting $N$ independent $d$-dimensional standard Brownian motions, denoted as $(W_t^i)_{t \geq 0}$, $i \in [N]$ and a prior information set $\mathcal{G} = \cup_{i=1}^N \mathcal{G}^i \subset \mathcal{F}$ independent of $(\mathcal{F}_t^i)_{t \geq 0}, i \in [N]$. For each $i \in [N]$, denote by $(\mathcal{F}_t^i)_{t \geq 0}$ the augmentation of the filtration generated by $W^i$. The sigma-algebra $\mathcal{G}^i$ represents the information available to player $i$ a priori regarding the matrix $A$, and we define $\mathbb{P}^i(\cdot) := \mathbb{P}(\cdot | \mathcal{G}^i)$. We also use $\mathbb{E}^i$ to denote the expectation under the prior distribution of player $i$ over $A$.

In order to write down the prior distribution over the inputs of the matrix $A$ and more importantly, in order to derive the formulas for the posterior process, we follow the framework in Liptser & Shiryayev (1978), which requires the matrix $A$ to be written as a vector.

For any matrix $M \in \mathbb{R}^{d \times d}$, define its vectorized form by stacking its rows: $M^{(v)} := [M(1), M(2), \ldots, M(d)]^T \in \mathbb{R}^{d^2}$, where $M(j)$ is the $j$-th row of $M$. Conversely, for any vector $M^{(v)} \in \mathbb{R}^{d^2}$, define the corresponding matrix $M \in \mathbb{R}^{d \times d}$ by

$$M := \left[ M^{(v)}[1:d], M^{(v)}[d+1:2d], \ldots, M^{(v)}[d(d-1)+1:d^2] \right]^T, \qquad (3.1)$$

where $M^{(v)}[i:j]$ denotes the subvector consisting of the $i$-th to $j$-th entries of $M^{(v)}$.

One more thing before setting up the prior for $A^{(v)}$ is that when we want to emphasize the dependence of parameters $\lambda^i, \Lambda^i, \rho^i, \eta^i$ and $\Upsilon^i$ from Proposition 2.1 on the matrix $A$, we will use the notation $\lambda^i(A), \Lambda^i(A), \rho^i(A), \eta^i(A)$ and $\Upsilon^i(A)$ respectively.

**Assumption 3.1** (Prior distribution). *For any $i \in [N]$, under $\mathbb{P}^i$, $A^{(v)} \sim \mathcal{N}(\mu_0^i, \Sigma_0^i)|_{\Theta^{(v)}}$, where $\mu_0^i \in \mathbb{R}^{d^2}$, and $\Sigma_0^i \in \mathbb{R}^{d^2 \times d^2}$ is a symmetric positive definite matrix, satisfying $\mathrm{tr}(\Sigma_0^i) < \infty$ (P-a.s.). $\mathcal{N}(\mu_0^i, \Sigma_0^i)|_{\Theta^{(v)}}$ denotes the projection of distribution $\mathcal{N}(\mu_0^i, \Sigma_0^i)$ on the set $\Theta^{(v)} \subset \mathbb{R}^{Nd}$. The support $\Theta^{(v)}$ satisfies the following: there are constants $M_A, M_\lambda, M_\Upsilon, C, c > 0$ such that:*

*1. For all $A^{(v)} \in \Theta^{(v)}$, let $A$ be the matrix corresponding to $A^{(v)}$. $A$ satisfies*
$$\|A\| \le M_A, \quad |\lambda^i(A)| \le M_\lambda, \quad \|\Upsilon^i(A)\| \le M_\Upsilon, \quad \|\eta^i(A)\| \le M_\eta.$$

*2. For any $A^{(v)}, \hat{A}^{(v)} \in \Theta^{(v)}$, let their corresponding matrices be $A$ and $\hat{A}$. For any $t \ge 0$,*
$$\left\| e^{\left(A - \hat{A} - \varsigma^i \Upsilon^i(\hat{A})\right)t} \right\| \le e^{-ct},$$

*where throughout this paper, $\|\cdot\|$ denotes the Frobenius norm for matrices.*

In our assumption, (2) enforces exponential decay to assure the stability of the following TS algorithm, which is the continuous-time equivalent of Ouyang et al. (2020, Assumption 2). It differs from Ouyang et al. (2020, Assumption 1) and Assumptions (A3) and (A4) in Gagrani et al. (2021), as we do not require independence between the columns of $\hat{A}$. This condition holds if, for example, $A - \hat{A} - \varsigma\Upsilon$ is symmetric negative definite and so $\|e^{(A - \hat{A} - \varsigma\Upsilon)}\| < 1$.

**Remark 3.1.** *We assume that the prior distribution is Gaussian for the explicit tractability. In fact, the analysis of our algorithm introduced later in Section 3.1 relies on the fact that the posterior variance is decreasing, which follows from the law of total variance. Accordingly, the same regret bound holds for any prior. Following Ekström et al. (2022) who consider sequentially estimating the unknown drift of a Brownian motion under an arbitrary prior, we can characterize a tractable posterior for our Gaussian process under any prior. Consistent with this, our experiments in A.9.5 shows that our TS algorithm remains effective across different priors.*

Since the model now includes a prior distribution on the matrix $A$ we need to adjust the definition of admissible strategies. Loosely speaking, we say that a strategy $(\alpha_t^i)_{t \ge 0}$ is *admissible in the model with uncertainty about the matrix $A$* if each Player $i$ uses only information from their respective Brownian motion, the prior distribution, and other posterior sampling. More formally:

**Definition 3.1.** *The strategy $(\alpha_t^i)_{t \ge 0}$ is admissible for Player $i$ if it satisfies Definition 2.1, with the filtration $(\mathcal{F}_t^i)_{t \ge 0}$ replaced by a filtration generated by (a) $\mathcal{G}^i$, (b) $(\mathcal{F}_t^i)_{t \ge 0}$, and (c) randomization devices, independent of $\mathcal{G}^i \cup (\mathcal{F}_t^i)_{t \ge 0}$, which serve for sampling from Player $i$'s posterior distributions.*

### 3.1 Thompson Sampling algorithm

We describe the TS algorithm for an arbitrary player $i \in \mathbb{N}$. Each player $i$ operates in episodes $[t_k^i, t_{k+1}^i)_{k=0}^\infty$, as outlined below.

**Episode $k = 0$:**

- Player $i$ samples at time $t_0^i := 0$ a parameter $\hat{A}_0^{i,(v)} \in \mathbb{R}^{d^2}$ from the prior distribution $\mathcal{N}(\mu_0^i, \Sigma_0^i)|_{\Theta^{(v)}}$, independently of the other random elements in the model. Player $i$ calculates its feedback strategy using the sample, which we now write in a matrix form $\hat{A}_0^i$

(following (3.1)). We define $\Upsilon_0^i = \Upsilon^i(\hat{A}_0^i)$ and $\eta_0^i = \eta^i(\hat{A}_0^i)$. Let the feedback control be
$\bar{a}^i(x; \hat{A}_0^i) = (\varsigma^i \Upsilon_0^i + \hat{A}_0^i)x - \varsigma^i \Upsilon_0^i \eta_0^i$.

- Player $i$ continuously observes its own state process and calculates the process of the posterior distribution on $A^{(v)}$ according to Proposition 3.1 given below, which claims that the posterior distribution at time $t$ is $\mathcal{N}(\mu_t^i, \Sigma_t^i)$ for some (random) mean $\mu_t^i$ and covariance matrix $\Sigma_t^i$, whose expressions are provided in Proposition 3.1.
- For the first episode, let the stopping criterion be

$$t_1^i := \min\{t > t_0^i + 1 : \det(\Sigma_t^i) < 0.5 \det(\Sigma_{t_0^i}^i) \text{ or } t - t_0^i \geq 2\}. \tag{3.2}$$

**Episodes $k \geq 1$:**

- Assume that the algorithm is defined for Player $i$ for the episodes $0, \ldots, k-1$.
- Player $i$ samples at time $t_k^i$ a parameter $\hat{A}_k^{i,(v)} \in \mathbb{R}^{d^2}$ from the posterior distribution $\mathcal{N}(\mu_{t_k^i}^i, \Sigma_{t_k^i}^i)|_{\Theta^{(v)}}$, independently of the other random elements in the model. Player $i$ calculates its feedback strategy using the sample, which we now write in a matrix form $\hat{A}_k^i$ (following (3.1)). Here, for each episode $k$, we define $\Upsilon_k^i := \Upsilon^i(\hat{A}_k^i)$ and $\eta_k^i := \eta^i(\hat{A}_k^i)$ as the Riccati solution and affine term corresponding to the sampled parameter $\hat{A}_k^i$. Let the feedback control be $\bar{a}^i(x; \hat{A}_k^i) = (\varsigma^i \Upsilon_k^i + \hat{A}_k^i)x - \varsigma^i \Upsilon_k^i \eta_k^i$.
- Player $i$ continuously observes its own state process and calculates the process of the posterior distribution on $A^{(v)}$ according to Proposition 3.1 given below, which claims that the posterior distribution at time $t$ is $\mathcal{N}(\mu_t^i, \Sigma_t^i)$ for some (random) mean $\mu_t^i$ and covariance matrix $\Sigma_t^i$, whose expressions are provided in Proposition 3.1.
- Let $t_k^i$ and $T_k^i$ denote the starting point and the length of episode $k$ for player $i$, respectively. The episode $k$ ends if the determinant value of $\Sigma_t^i$ falls below half of its value at the beginning of the episode, or if the length of episode $k+1$ is more than the length of episode $k$, i.e.,

$$t_{k+1}^i := \min\{t \geq t_k^i + 1 : \det(\Sigma_t^i) < 0.5 \det(\Sigma_{t_k^i}^i) \text{ or } t - t_k^i \geq T_k^i + 1\}. \tag{3.3}$$

**Remark 3.2.** *In the algorithm, we do not necessarily assume the independence of sampling across players, but only that samplings of the same player are independent.*

As a result of this algorithm, the dynamics of the observed process $(\hat{X}_t^i)_{t \geq 0}$ can be written as

$$\begin{aligned}
d\hat{X}_t^i &= (A\hat{X}_t^i - \bar{a}^i(\hat{X}_t^i; \hat{A}_k^i))dt + \sigma^i dW_t^i \\
&= [(A - \hat{A}_k^i - \varsigma^i \Upsilon_k^i)\hat{X}_t^i + \varsigma^i \Upsilon_k^i \eta_k^i]dt + \sigma^i dW_t^i, \qquad t \in [t_k^i, t_{k+1}^i), \quad k \in \mathbb{N}.
\end{aligned}$$

and its explicit solution is given recursively by

$$\begin{aligned}
\hat{X}_t^i =\ & e^{(A - \hat{A}_k^i - \varsigma^i \Upsilon_k^i)(t - t_k)}\hat{X}_{t_k^i}^i - (A - \hat{A}_k^i - \varsigma^i \Upsilon_k^i)^{-1}(I - e^{(A - \hat{A}_k^i - \varsigma^i \Upsilon_k^i)(t - t_k)})\varsigma^i \Upsilon_k^i \eta_k^i \\
& + \int_{t_k^i}^t e^{(A - \hat{A}_k^i - \varsigma^i \Upsilon_k^i)(t - s)}\sigma^i dW_s^i, \qquad\qquad\qquad t \in [t_k^i, t_{k+1}^i), \quad k \in \mathbb{N}.
\end{aligned}$$

---

**Algorithm 1:** Thompson Sampling for Stochastic Games - Implementation in discrete time steps

---

**Initialize player $i$:** Initialize prior distribution of $\hat{A}_0^{i,(v)}$ : $(\mu_0^i, \Sigma_0^i)$. Let $t_0^i = 0, T_0^i = 0, k = 0$;

**for** $t = \Delta t, 2\Delta t, \ldots$ **do**

    Observe current states $\hat{X}_t^i$;

    Update strategy $\bar{a}^i(x; \hat{A}_k^i)$ by $\bar{a}^i(x; \hat{A}_k^i) \leftarrow$ **Feedback Strategy-i**$(\hat{X}_t^i)$;

**end for**

**Function: Feedback Strategy-i**$(\hat{X}_t^i)$;

    **global var** $t$;

    Update $(\mu_t^i, \Sigma_t^i)$ according to posterior distribution;

    **if** $t - t_k^i \geq 1$ **and** $(t - t_k^i \geq T_{k-1}^i + 1$ **or** $\det(\Sigma_t^i) < 0.5 \det(\Sigma_{t_k^i}^i))$ **then**

        Update $T_k^i \leftarrow t - t_k^i, \quad k \leftarrow k + 1, \quad t_k^i \leftarrow t$;

        Sample $\hat{A}_k^{i,(v)}$ from the posterior distribution $\mathcal{N}(\mu_{t_k^i}^i, \Sigma_{t_k^i}^i)|_{\Theta^{(v)}}$;

    **return** $(\varsigma^i \Upsilon_k^i + \hat{A}_k^i)\hat{X}_t^i - \varsigma^i \Upsilon_k^i \eta_k^i$;

---

For the rest of the paper, denote by $\{\hat{\mathcal{F}}_t\}_{t \geq 0}$ the filtration generated by the underlying Brownian motions, as well as the randomization devices hosted on $(\Omega, \mathcal{F}, \mathbb{P})$. Here, the randomization devices are used by Player $i$ to sample from the posterior Gaussian distribution.

The following proposition describes the evolution of the posterior process. It follows from Liptser & Shiryayev (1978, Theorem 12.8).

**Proposition 3.1** (Posterior distribution). *Let Assumption 3.1 be in force. Then under $\mathbb{P}^i$, $A^{(v)}|\hat{\mathcal{F}}_t^i \sim \mathcal{N}(\mu_t^i, \Sigma_t^i)|_{\Theta^{(v)}}$, where $\mu_t^i$ and $\Sigma_t^i$ are given by the formulas*

$$\mu_t^i = \left[ I_{d^2} + \Sigma_{t_k^i}^i \left( (\sigma^i(\sigma^i)^T)^{-1} \otimes \int_{t_k^i}^t \hat{X}_s^i (\hat{X}_s^i)^T ds \right) \right]^{-1}$$

$$\times \left[ \mu_{t_k^i}^i + \Sigma_{t_k^i}^i \int_{t_k^i}^t (I_d \otimes \hat{X}_s^i)(\sigma^i(\sigma^i)^T)^{-1} \Big( d\hat{X}_s^i + \big( \varsigma^i \Upsilon_k^i (\hat{X}_s^i - \eta_k^i) + (I_d \otimes (\hat{X}_s^i)^T) \hat{A}_k^{i,(v)} \big) ds \Big) \right],$$

$$\Sigma_t^i = \left[ I_{d^2} + \Sigma_{t_k^i}^i \left( (\sigma^i(\sigma^i)^T)^{-1} \otimes \int_{t_k^i}^t \hat{X}_s^i (\hat{X}_s^i)^T ds \right) \right]^{-1} \Sigma_{t_k^i}^i, \qquad t \in [t_k^i, t_{k+1}^i),$$

*where $I_{d^2}$ is the $d^2 \times d^2$ identity matrix and $\otimes$ stands for the Kronecker product.*

It is worth mentioning that the statement in Liptser & Shiryayev (1978, Theorem 12.8) deals with an untruncated prior. However, it is standard that if for two measures $\nu$ and $\mu$, satisfying $\nu = f \cdot \mu$ via the Radon–Nikodym derivative $f$, then restricting both $\mu$ and $\nu$ to a measurable set $S$ gives $\nu_S = f \cdot \mathbf{1}_S \cdot \mu_S$, so the Radon–Nikodym derivative of $\nu_S$ with respect to $\mu_S$ is $f \cdot \mathbf{1}_S$.

## 3.2 MAIN RESULTS

The Bayesian *regret* of an admissible strategy profile $\alpha \in \mathcal{A}^i$ over a horizon $T$ is defined as

$$R^i(\alpha, T) := \mathbb{E}^i \Big[ \int_0^T f^i(\hat{X}_t^i, \alpha_t; \boldsymbol{m}^{-i}) dt - T\lambda^i(A) \Big],$$

where $f^i(\hat{X}_t^i, \alpha_t; \boldsymbol{m}^{-i})$ represents player $i$'s cost at time $t$ under the algorithm, if other players' states follow their stationary distribution. $\lambda^i(A)$ denotes the equilibrium cost of player $i$ under known parameter $A$, from Proposition 2.1.

Denote by $\hat{\alpha}^i$ the TS algorithm strategy for Player $i \in [N]$:

$$\hat{\alpha}_t^i := \bar{a}^i(\hat{X}_t^i; \hat{A}_k^i), \qquad \forall t \in [t_k^i, t_{k+1}^i), \quad k = 0, 1, \dots \tag{3.4}$$

**Theorem 3.1.** *Under Assumptions 2.1 and 3.1, the regret is upper bounded as follows:*

$$R^i(\hat{\alpha}^i, T) \leq O\Big( \min(\|\sigma^i\|^3 \sqrt{d}, d) \sqrt{T \log(T)} \Big).$$

The key idea behind the proof of the regret estimate in Theorem 3.1 is a decomposition of $R^i(T)$ into three parts, using the HJB equation and Itô's lemma. We then bound each part separately and combine the results to obtain the overall regret bound.

**Proposition 3.2.** *Under Assumptions 2.1 and 3.1,*

$$R^i(\hat{\alpha}^i, T) = R_0^i(\hat{\alpha}^i, T) + R_1^i(\hat{\alpha}^i, T) + R_2^i(\hat{\alpha}^i, T), \quad where \tag{3.5}$$

$$R_0^i(\hat{\alpha}^i, T) := \mathbb{E}^i \left[ \int_0^T \lambda^i(\hat{A}_{k(t)}^i) \, dt - T\lambda^i(A) \right], \quad sampling\ error\ regret,$$

$$R_1^i(\hat{\alpha}^i, T) := \mathbb{E}^i \big[ v^i(X_0^i) - v^i(\hat{X}_T^i) \big], \quad cumulative\ strategy\ effect,$$

$$R_2^i(\hat{\alpha}^i, T) := \mathbb{E}^i \left[ \int_0^T (\nabla v^i(\hat{X}_t^i))^\top (A - \hat{A}_{k(t)}^i) \hat{X}_t^i \, dt \right], \quad model\ mismatch\ regret.$$

*where $k(t)$ is the episode label $k$ such that $t \in [t_k^i, t_{k+1}^i)$.*

The proof of Theorem 3.1 follows from the bounds of the regret terms as in the following proposition.

**Proposition 3.3.** *Under Assumptions 2.1 and 3.1, the terms in Proposition 3.2 are bounded as:*

1. $R_0^i(\hat{\alpha}^i, T) \leq O(d\sqrt{T \log(T)})$.
2. $R_1^i(\hat{\alpha}^i, T) \leq \|\sigma^i\|^2 O(1)$.
3. $R_2^i(\hat{\alpha}^i, T) \leq O(\|\sigma^i\|^3 \sqrt{dT \log(T)})$.

The following proposition provides a rate of convergence from the sampled parameter $\hat{A}_{k(t)}^i$ to the true drift matrix $A$. It is crucial in proving the bound for $R_2^i(\hat{\alpha}^i, T)$. In particular, it states that the integral of the expected squared deviation between $A$ and $\hat{A}_{k(t)}^i$ grows at most logarithmically in the time horizon $T$.

**Proposition 3.4.** *For any player $i$, $\int_0^T \mathbb{E}^i \left[ \|A - \hat{A}_{k(t)}^i\|^2 \right] dt \leq O(d\|\sigma^i\|^2 \log(T))$.*

Our next main result shows that the strategy profile of the TS algorithm constitutes a Nash equilibrium.

**Theorem 3.2.** *Let Assumptions 2.1 and 3.1 be in force, and that for each $i \in [N]$, $\frac{1}{2}\left(\varsigma^i \Upsilon^i + (\varsigma^i \Upsilon^i)^T\right)$ is positive definite. Let $(\hat{\alpha}_t^1, \ldots, \hat{\alpha}_t^N)_{t \geq 0}$ be the strategy profile where all the players follow the TS algorithm. Then, this profile constitutes a Nash equilibrium. Moreover, the dynamics of the players are ergodic with the same limiting distributions with densities $m^i$'s as in the full information problem.*

The key to proving the Nash equilibrium is showing that the dynamics under the TS algorithm, $(\hat{X}_t^i)_{t \geq 0}$, closely match those under the true parameter $A$, $(X_t^i)_{t \geq 0}$ from (2.12), with $A$ randomized at $t = 0$ according to the prior and observed by Player $i$. The process $(X_t^i)_{t \geq 0}$ is used as an auxiliary process. Specifically, we couple both processes by initializing them with the same state $\hat{X}_0^i = X_0^i$ and the same Brownian motion $(W_t^i)_{t \geq 0}$. We then show that:

**Proposition 3.5.** *Under Assumptions 2.1 and 3.1 as well as that $\frac{1}{2}\left(\varsigma^i \Upsilon^i + (\varsigma^i \Upsilon^i)^T\right)$ is positive definite for any $i \in [N]$, we have*

$$\int_0^T \mathbb{E}^i[\|\hat{X}_t^i - X_t^i\|^2]dt \leq O\left(\|\sigma^i\|^3 \sqrt{dT \log(T)}\right),$$

*where here and in the proof, for any vector $v \in \mathbb{R}^d$, $\|v\|$ denotes the $l^2$-norm of $v$.*

This, along with the ergodicity of $(X_t^i)_{t \geq 0}$ from Proposition 2.1, implies the ergodicity of $(\hat{X}_t^i)_{t \geq 0}$ and that its limiting distribution is $m^i$, ensuring admissibility of $\hat{\alpha}^i$. Additionally, leveraging the quadratic costs, we extract the Nash equilibrium conditions from $(X_t^i)_{t \geq 0}$ and apply them to $(\hat{X}_t^i)_{t \geq 0}$.

The following proposition provides a convergence rate for the TS strategy itself, measuring how fast the TS control $\hat{\alpha}_t^i$ approaches the equilibrium feedback control in the full information case. We denote

$$(\alpha_t^i)^* = \bar{a}^i(X_t^i; A), \tag{3.6}$$

as the equilibrium policy under the known parameter $A$, following from (2.11).

**Proposition 3.6.** *Under Assumptions Assumption 2.1 and Assumption 3.1, the TS feedback strategy $\hat{\alpha}^i$ of player $i \in [N]$ satisfies*

$$\mathbb{E}^i \left[ \int_0^T \|\hat{\alpha}_t^i - (\alpha_t^i)^*\|^2 dt \right] \leq O\left( \max\left(\|\sigma^i\|^{\frac{13}{2}} d^{\frac{1}{4}}, \|\sigma^i\|^5 d^{\frac{1}{2}}\right) T^{\frac{3}{4}} (\log T)^{\frac{1}{2}} \right).$$

## 4 Numerical Experiments

We evaluate the empirical performance of our TS algorithm in the $N$-player stochastic differential game described in Section 3. We simulate the game with $N = 10, 50, 100$ players, where each player is assigned a different diffusion matrix $\sigma^i$ and a distinct cost structure $(Q^i, R^i)$. We focus on a single player and compute their cumulative regret over time.

To evaluate the scaling behavior and variance, we repeat the simulation for $n = 10, 50$, and $100$ sample paths. The results are shown in Figure 1: panel (a) plots $R(T)$ as a function of time $T$, and panel (b) shows the scaled regret $R(T)/\sqrt{T \log(T)}$. The shaded region indicates $0.2$ times the standard deviation of regret over sample paths at each time step.

As seen in Figure 3, the regret grows sublinearly with time, and the normalized regret $R(T)/\sqrt{T \log(T)}$ remains bounded as $T$ increases, in agreement with the theoretical upper bound $O\big(\sqrt{T \log(T)}\big)$ established in Theorem 3.1. Simulation details are provide in Appendix A.9.1.

Additional experiments, including results in higher dimensions, with different prior distributions, and comparisons against baseline strategies, are provided in Appendix A.9.

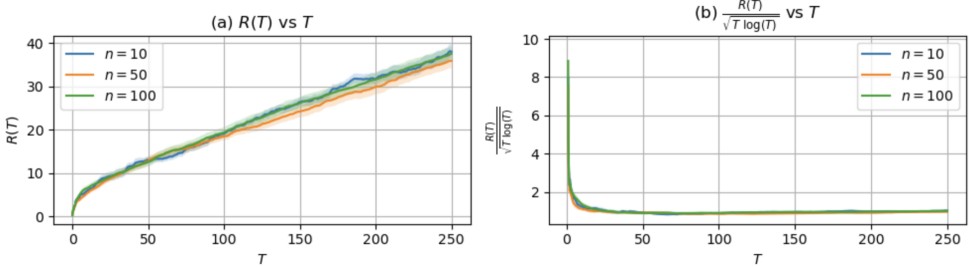

Figure 1: Regret vs time.

## 5 CONCLUSION AND FUTURE OUTLOOK

In this work, we proposed a Thompson Sampling algorithm for a linear-quadratic stochastic differential game with an unknown common drift matrix. Under an open-loop partial observation setting, we designed an episodic strategy using posterior sampling and established a sublinear Bayesian regret bound of order $O(\sqrt{T \log(T)})$. Furthermore, we showed that the algorithm leads to a Nash equilibrium. Our theoretical findings are supported by numerical experiments, which confirm the sublinear growth of cumulative regret.

Looking ahead, an important extension is to study the case where players have heterogeneous parameters and the game admits a closed-loop structure. Namely, each player observes the full state vector of all other players. As a first step towards this goal, it will be necessary to analyze the fully observed stochastic game. This would bridge the gap between classical Nash theory under full information and modern learning-based approaches under partial observability.

ETHICS STATEMENT

This research complies with the ICLR code of Ethics. It is purely theoretical and does not involve human subjects, personal data, or proprietary datasets. The authors declare no conflicts of interest and assume full responsibility for the content of the manuscript.

REPRODUCIBILITY STATEMENT

Our theoretical results are fully reproducible from the algorithm and the complete proofs provided in the main text and the appendix. To illustrate the theoretical findings, we provide code implementing the numerical experiments, which is included in the supplementary materials (anonymized for review). Similar qualitative behaviors can be reproduced by running the released code with reasonable parameter choices.

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

# A  APPENDIX

In the following appendices, we present the proofs of the main results of the paper. The proof of the regret bound is inspired by the argument in Ouyang et al. (2017; 2020), and Gagrani et al. (2021). The proofs of Theorem 3.2 and Proposition 3.5 rely on the game-theoretic structure of the model rather than an optimal control framework, and are therefore more novel.

We start with two examples, satisfying Assumption 2.1 as well as that $\frac{1}{2}\big(\varsigma^i\Upsilon^i + (\varsigma^i\Upsilon^i)^T\big)$ is positive definite for any $i$, which is required for the proof of Theorem 3.2. These examples were originally detailed in Bardi & Priuli (2014): the first with a symmetric parameter matrix $A$, and the second with a nondefective matrix $A$, which can be transformed into a symmetric system.

## A.1  EXAMPLES

First we introduce the definition of nearly identical players from Bardi & Priuli (2014, Definition 4.2), which serves as a standing assumption in the following two examples.

**Definition A.1.** *We say that the players are* nearly identical *if the following conditions are satisfied:*

- *For any $i, j, k \in [N]$, $F^i(\ldots, x^j, \ldots, x^k, \ldots, \alpha) = F^i(\ldots, x^k, \ldots, x^j, \ldots, \alpha)$.*

- *For any $i \in [N]$, the cost matrix $\mathcal{Q}^i$ is symmetric. View $\mathcal{Q}^i$ as composed of $N \times N$ blocks of size $d \times d$. The diagonal block $Q_{ii}^i = Q^*$, which is identical across all players.*

- *For any $i \in [N]$, the vector $\overline{\boldsymbol{x}}_i \in \mathbb{R}^{Nd}$, the $j$-th $d \times 1$ block is the same across all $j \neq i$ and all $i \in [N]$, denoted by $\Delta$. The $i$-th block of $\overline{\boldsymbol{x}}_i$ satisfies $\overline{x}_i^i = H$.*

- *In the dynamics in (2.1), $\sigma^i = \sigma \in \mathbb{R}^{d \times d}$ for all $i \in [N]$.*

- *In the cost function (2.4), $R^i = R \in \mathbb{R}^{d \times d}$ for all $i \in [N]$.*

### A.1.1  SYMMETRIC SYSTEM

Consider an $N$-player game with dynamics (2.1) and costs (2.4). Assume that players are nearly identical, and for all $i \in [N]$

- (a)  In the dynamics in (2.1), $A \in \mathbb{R}^{d \times d}$ is symmetric and $\sigma = sI_d$ with $s \in \mathbb{R}\backslash\{0\}$.
- (b)  In the cost function (2.4), $R^i = rI_d$ with $r > 0$.
- (c)  Each off-diagonal block of $\mathcal{Q}^i$ satisfies $Q_{ij}^i = Q_{ji}^i = \check{Q}/2$, for any $j \neq i$, $i, j \in [N]$.

Then both matrices $\varsigma = \frac{s^2}{2}I_d$ and $R = rI_d$ commute with any other matrix. Consider the algebraic Riccati equation (2.8), we find that

$$r\frac{s^4}{8}\Upsilon^2 = Q^* + \frac{r}{2}A^2,$$

which implies

$$\Upsilon = \frac{2}{s^2}\sqrt{\frac{2}{r}Q^* + A^2},$$

as $\frac{2}{r}Q^* + A^2$ is symmetric and positive definite. Since $\varsigma = \frac{s^2}{2}I_d$, it commutes with $\Upsilon$, and hence $\varsigma\Upsilon$ is symmetric and positive definite.

Plug in (2.7) and (2.6), we have

$$\boldsymbol{p} = -\Big(Q^*H + \frac{N-1}{2}\check{Q}\Delta\Big)$$

$$\mathcal{B} = -\Big(Q^* + \frac{r}{2}A^2 + \frac{N-1}{2}\check{Q}\Big).$$

so $\mathcal{B}$ is invertible. Thus the linear system $\mathcal{B}\eta = \boldsymbol{p}$ has the unique solution

$$\eta = (-\mathcal{B})^{-1}\Big(Q^*H + \frac{N-1}{2}\check{Q}\Delta\Big).$$

In conclusion, conditions (a), (b) and (c) are sufficient to guarantee the existence of a unique solution to (2.5) of the form (2.9), and hence of a unique affine Nash equilibrium strategy given by

$$\bar{a}(x; A) = R^{-1}(\Lambda x + \rho) = Ax + \varsigma \Upsilon(x - \eta) = Ax + \sqrt{\frac{2}{r}Q^* + A^2}(x - \eta).$$

### A.1.2  NONDEFECTIVE SYSTEM

A matrix $M \in \mathbb{R}_{d \times d}$ is said to be *non-defective* if for every eigenvalue $\lambda$ in the spectrum of $M$, the corresponding eigenspace has dimension equal to the multiplicity of $\lambda$ or, equivalently, when there exists a basis of $\mathbb{R}^d$ consisting of right (or left) eigenvectors of $M$.

The following proposition follows from (Bardi & Priuli, 2014, Proposition 6.1):

**Proposition A.1.** *Let $M$ be any $d \times d$ real matrix. Then the following properties hold:*

*(i) There exists an invertible and symmetric symmetrizer for $M$, i.e., there exists a matrix $Y \in \mathbb{R}^{d \times d}$ such that*

$$\det(Y) \neq 0, \quad Y^T = Y, \quad YM = M^T Y.$$

*(ii) If $M$ is nondefective, then the symmetrizer $Y$ can be chosen positive definite.*

*(iii) If $M$ is nondefective, $\sigma$ is invertible, and we consider a linear SDE*

$$dx_t = (Mx_t - \alpha_t)dt + \sigma dW_t, \tag{A.1}$$

*then there exists a linear change of coordinates $x \mapsto \xi$ such that* (A.1) *can be rewritten in the form*

$$d\xi_t = (\widetilde{M}\xi_t - \tilde{\alpha}_t)dt + \tilde{\sigma}dW_t$$

*with $\widetilde{M}$ symmetric matrix and $\tilde{\sigma}$ invertible.*

Consider a differential game with $N$ nearly identical players, and $A \in \mathbb{R}^{d \times d}$ is nondefective.

From the proof of (Bardi & Priuli, 2014, Proposition 6.1), we can denote $Y = P^T Z^2 P$ as the symmetrizer matrix for $A$ with $P$ orthogonal matrix and $Z$ a diagonal and positive definite matrix. Accordingly,

$$\tilde{A} = Z^{-1}PYAP^T Z^{-1}, \qquad \tilde{\alpha} = ZP\alpha, \qquad \tilde{\sigma} = ZP\sigma.$$

Then the new game will have costs given by

$$\tilde{J}^i(\Xi, \tilde{\alpha}^1, \dots, \tilde{\alpha}^N) := \liminf_{T \to \infty} \frac{1}{T}\mathbb{E}^i\Big[ \int_0^T \Big( \frac{(\tilde{\alpha}^i)^T \tilde{R}\tilde{\alpha}^i}{2} + \sum_{j,k=1}^N (\Xi^j - \overline{\Xi}_i^j)^T \tilde{Q}_{jk}^i(\Xi^k - \overline{\Xi}_i^k) \Big)dt \Big],$$

where the new variables $\Xi, \overline{\Xi}_1, \dots, \overline{\Xi}_N \in \mathbb{R}^{Nd}$ and $\tilde{\alpha}^1, \dots, \tilde{\alpha}^N \in \mathbb{R}^d$ satisfy for $k \in [N]$

$$\Xi^k = ZPX^k, \qquad \overline{\Xi}_i^k = ZP\overline{x}_i^k, \qquad \tilde{\alpha}^k = ZP\alpha^k,$$

and the matrices $\tilde{R}$ and $\tilde{Q}_{jk}^i$ are given by

$$\tilde{R} = Z^{-1}PRP^{-1}Z^{-1}, \quad \tilde{Q}_{jk}^i = Z^{-1}PQ_{jk}^i P^{-1}Z^{-1},$$

Then we can replace (a), (b) in Appendix A.1.1 by

(a') dynamics (2.1) are given by drift matrices $A$ nondefective and diffusion matrices $\sigma = sP^T Z^{-1}$ with $s \in \mathbb{R} \setminus \{0\}$.

(b') matrix $R$ in control costs (2.4) satisfies $R = rY$ with $r > 0$.

Thus after the change of coordinates $\xi = ZPx$, this problem becomes the symmetric system problem, it is possible to repeat the arguments of Appendix A.1.1.

## A.2 PROOF OF PROPOSITION 3.2

In order to analyze the regret, we decompose it as follows:

$$R^i(\hat{\alpha}^i, T) = \mathbb{E}^i \Big[ \int_0^T f^i(\hat{X}_t^i, \hat{\alpha}_t^i; \boldsymbol{m}^{-i}) dt - T\lambda^i(A) \Big] \tag{A.2}$$

$$= \mathbb{E}^i \Big[ \int_0^T \lambda^i(\hat{A}_{k(t)}^i) dt - T\lambda^i(A) \Big] + \mathbb{E}^i \Big[ \int_0^T \Big( f^i(\hat{X}_t^i, \hat{\alpha}_t^i; \boldsymbol{m}^{-i}) - \lambda^i(\hat{A}_{k(t)}^i) \Big) dt \Big]$$

where $k(t)$ is the episode label $k$ such that $t \in [t_k^i, t_{k+1}^i)$. We now expand the last expectation on the right. By the HJB equation (2.5),

$$f^i(\hat{X}_t^i, \hat{\alpha}_t^i; \boldsymbol{m}^{-i}) - \lambda^i(\hat{A}_{k(t)}^i) = -\text{tr}\left(\varsigma^i D^2 v^i(\hat{X}_t^i)\right) - (\nabla v^i(\hat{X}_t^i))^T (\hat{A}_{k(t)}^i \hat{X}_t^i - \hat{\alpha}_t^i). \tag{A.3}$$

Here $v^i$ is the function defined in the solution of HJB-KFP equation system, given in (2.9). With our eyes towards the first term on the right-hand side, Apply Itô's lemma to $v^i(\hat{X}_t^i)$:

$$v^i(\hat{X}_T^i) = v^i(\hat{X}_0^i) + \int_0^T (\nabla v^i(\hat{X}_t^i))^T d\hat{X}_t^i + \int_0^T \text{tr}\left(\varsigma^i D^2 v^i(\hat{X}_t^i)\right) dt,$$

Expand $d\hat{X}_t^i$ and rearrange terms to get

$$\mathbb{E}^i \Big[ \int_0^T \text{tr}\left(\varsigma^i D^2 v^i(\hat{X}_t^i)\right) dt \Big] = \mathbb{E}^i \Big[ v^i(\hat{X}_T^i) - v^i(\hat{X}_0^i) - \int_0^T (\nabla v^i(\hat{X}_t^i))^T (A\hat{X}_t^i - \hat{\alpha}_t^i) dt \Big]. \tag{A.4}$$

Combining (A.2), (A.3), and (A.4), the regret can be split as in (3.5). $\qquad\square$

## A.3 AUXILIARY RESULTS

In this section, we introduce three lemmas. The first bounds the $p$-th moment of the maximum state norm until time $T$, the second bounds the number of episodes until time $T$, and together they are applied in the proof of Proposition 3.3. The third lemma establishes a Lipschitz condition of parameters $\Upsilon^i(A)$ and $\eta^i(A)$ with respect to $A$, which will be used in the proof of Proposition 3.5.

We first establish the lemma on the boundedness of the $p$-th moment of the maximum state norm. For any continuous function $y : [0, \infty) \to \mathbb{R}^d$, define $\|y\|_T^* := \max_{0 \le t \le T} \|y(t)\|$ as the maximum norm of $y(t)$ along the trajectory over $[0, T]$.

Recall the dynamics of the process $(X_t^i)_{t \ge 0}$ from equation (2.13), where we assume that the matrix $A$ is randomized at time $t = 0$ according to the prior distribution specified in Assumption 3.1. We further assume that this realization of $A$ is observed by the Player. In the sequel, we employ this process as an auxiliary tool to analyze $(\hat{X}_t^i)_{t \ge 0}$. This is motivated by the fact that the behavior of $(X_t^i)_{t \ge 0}$ is already well-understood from Proposition 2.1, and our aim is to project its features onto the process $(\hat{X}_t^i)_{t \ge 0}$.

**Lemma A.1.** *For any $p \ge 1$, we have* [2]

$$\mathbb{E}^i \left[ \left( \|\hat{X}^i\|_T^* \right)^p \right] \le C(\|\sigma^i\|)^p \qquad \text{and} \qquad \mathbb{E}^i \left[ \left( \|X^i\|_T^* \right)^p \right] \le \hat{C}(\|\sigma^i\|)^p,$$

*where $C$ and $\hat{C}$ are constants depending only on $p$, $M_\Upsilon$, $M_\eta$, $\|X_0^i\|^p$ and constant $c$ from Assumption 3.1.*

Next step is to bound the number of update periods up to time $T$. Recall the definition of episodes from (3.3). Each episode terminates either when the determinant of the covariance matrix drops below half of that at the beginning of the episode, or when its length exceeds that of the previous one. Denote the number of episodes by player $i$ until time $T$ as $K_T^i$,

$$K_T^i := \max\{k : t_k^i \le T\}. \tag{A.5}$$

In the following lemma, we will establish a bound for $K_T^i$:

---

[2]This provides a stronger bound than (Gagrani et al., 2021, Lemma 4), since they take a maximum over $W_t^i$ in every time step, whereas we apply the Burkholder–Davis–Gundy inequality over the entire time interval.

**Lemma A.2.** $K_T^i$ *is bounded as follows:*

$$K_T^i \leq O\left(d\sqrt{T \log\left(\frac{T}{d}\|(\sigma^i)(\sigma^i)^T)^{-1}\| \cdot (\|\hat{X}^i\|_T^*)^2\right)}\right).$$

As a preliminary step toward bounding $\int_0^T \|X_t^i - \hat{X}_t^i\|^2 dt$ in Proposition 3.5, we require control of the coefficients that appear in the dynamics of $X_t^i$ and $\hat{X}_t^i$, particularly, $\Upsilon^i(\cdot)$ and $\eta^i(\cdot)$. The following lemma provides the needed Lipschitz condition, which are used in Appendix A.6.

**Lemma A.3.** *Let Assumption 2.1 and Assumption 3.1 hold. For each $i \in [N]$, there exist finite constants $L_\Upsilon^i, L_\eta^i > 0$ such that, for all $A, \hat{A}$ whose vectorized form $A^{(v)}, \hat{A}^{(v)} \in \Theta^{(v)}$,*

$$\|\Upsilon^i(A) - \Upsilon^i(\hat{A})\| \leq L_\Upsilon^i \|A - \hat{A}\|, \qquad \|\eta^i(A) - \eta^i(\hat{A})\| \leq L_\eta^i \|A - \hat{A}\|.$$

### A.3.1 PROOF OF LEMMA A.1.

The proof of the second bound is similar to that of the first; however, the first is more involved. Therefore, we present only the detailed proof of the first bound.

It can be written explicitly using the notation $k(t)$ from Proposition 3.2 in the following non-recursive form:

$$\hat{X}_t^i = e^{\int_0^t (A - \hat{A}_{k(s)}^i - \varsigma^i \Upsilon_{k(s)}^i) ds} X_0^i + \int_0^t e^{\int_s^t (A - \hat{A}_{k(r)}^i - \varsigma^i \Upsilon_{k(r)}^i) dr} (\varsigma^i \Upsilon_{k(s)}^i \eta_{k(s)}^i ds + \sigma^i dW_s^i), \qquad t \geq 0,$$

where $k(t)$ is the episode label $k$ such that $t \in [t_k^i, t_{k+1}^i)$.

Taking norms on both sides, we obtain by the triangle inequality

$$\|\hat{X}_t^i\| \leq \left\|e^{\int_0^t (A - \hat{A}_{k(s)}^i - \varsigma^i \Upsilon_{k(s)}^i) ds}\right\| \cdot \|X_0^i\| + \int_0^t \left\|e^{\int_s^t (A - \hat{A}_{k(r)}^i - \varsigma^i \Upsilon_{k(r)}^i) dr} \varsigma^i \Upsilon_{k(s)}^i \eta_{k(s)}^i\right\| ds$$
$$+ \left\|\int_0^t e^{\int_s^t (A - \hat{A}_{k(r)}^i - \varsigma^i \Upsilon_{k(r)}^i) dr} \sigma^i dW_s^i\right\|. \tag{A.6}$$

We are now bounding the terms on the right-hand side of the above. For this, from Assumption 3.1, we note first that

$$\left\|e^{\int_s^t (A - \hat{A}_{k(r)}^i - \varsigma^i \Upsilon_{k(r)}^i) dr}\right\| \leq e^{-c(t-s)}. \tag{A.7}$$

Then, we can bound the first two terms on the right-hand side of (A.6) as follows

$$\left\|e^{\int_0^t (A - \hat{A}_{k(s)}^i - \varsigma^i \Upsilon_{k(s)}^i) ds}\right\| \cdot \|X_0^i\| + \int_0^t \left\|e^{\int_s^t (A - \hat{A}_{k(r)}^i - \varsigma^i \Upsilon_{k(r)}^i) dr} \varsigma^i \Upsilon_{k(s)}^i \eta_{k(s)}^i\right\| ds$$
$$\leq e^{-ct}\|X_0^i\| + \int_0^t e^{-c(t-s)}\|\varsigma^i \Upsilon_{k(s)}^i \eta_{k(s)}^i\| ds \tag{A.8}$$
$$\leq e^{-ct}\|X_0^i\| + \frac{1}{c}(1 - e^{-ct}) M_\Upsilon M_\eta \|\varsigma^i\|.$$

We now turn to the third term on the right-hand side of (A.6). By the Burkholder–Davis–Gundy Inequality, there exists a constant $C_p$ that depends solely on $p$, such that,

$$\mathbb{E}^i\left[\left(\sup_{t \in [0,T]} \left\|\int_0^t e^{\int_s^t (A - \hat{A}_{k(r)}^i - \varsigma^i \Upsilon_{k(r)}^i) dr} \sigma^i dW_s^i\right\|\right)^p\right]$$
$$\leq C_p \mathbb{E}^i\left[\left(\int_0^T \left\|e^{\int_s^t (A - \hat{A}_{k(r)}^i - \varsigma^i \Upsilon_{k(r)}^i) dr} \sigma^i\right\|^2 ds\right)^{p/2}\right]$$
$$\leq C_p \mathbb{E}^i\left[\left(\int_0^T \|\sigma^i\|^2 \cdot \left\|e^{\int_s^T (A - \hat{A}_{k(r)}^i - \varsigma^i \Upsilon_{k(r)}^i) dr}\right\|^2 ds\right)^{p/2}\right].$$

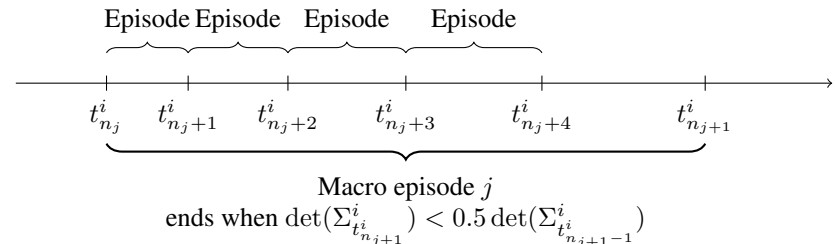

Macro episode $j$
ends when $\det(\Sigma^i_{t^i_{n_{j+1}}}) < 0.5\det(\Sigma^i_{t^i_{n_{j+1}-1}})$

Figure 2: An illustration of macro episode $j$. All episodes $[t^i_k, t^i_{k+1})$ within macro episode $j$, except the last one $[t^i_{n_{j+1}-1}, t^i_{n_{j+1}})$, do not satisfy the determinant condition $\det\left(\Sigma^i_{t^i_k}\right) < 0.5\det\left(\Sigma^i_{t^i_{k-1}}\right)$.

Apply (A.7) to the right handside,

$$
C_p\mathbb{E}^i\Big[\Big(\int_0^T \|\sigma^i\|^2 \cdot \Big\|e^{\int_s^T (A-\hat{A}^i_{k(r)}-\varsigma^i\Upsilon^i_{k(r)})dr}\Big\|^2 ds\Big)^{p/2}\Big]
$$

$$
\leq C_p\mathbb{E}^i\Big[\Big(\int_0^T \|\sigma^i\|^2 e^{-2c(T-s)}ds\Big)^{p/2}\Big] \tag{A.9}
$$

$$
\leq C_p\Big(\frac{1}{2c}\|\sigma^i\|^2\Big)^{p/2}.
$$

Taking the supremum over the interval $[0, T]$ on both sides of (A.6), taking the $p$-th power, and applying (A.8), we obtain:

$$
(\|\hat{X}^i\|^*_T)^p \leq 3^{p-1}\Big(\sup_{t\in[0,T]}\Big(\Big\|e^{\int_0^t (A-\hat{A}^i_{k(s)}-\varsigma^i\Upsilon^i_{k(s)})ds}\Big\|\Big)\cdot\|X^i_0\|\Big)^p
$$

$$
+ 3^{p-1}\Big(\sup_{t\in[0,T]}\int_0^t \Big\|e^{\int_s^t (A-\hat{A}^i_{k(r)}-\varsigma^i\Upsilon^i_{k(r)})dr}\varsigma^i\Upsilon^i_{k(s)}\eta^i_{k(s)}\Big\|ds\Big)^p
$$

$$
+ 3^{p-1}\Big(\sup_{t\in[0,T]}\Big\|\int_0^t e^{\int_s^t (A-\hat{A}^i_{k(r)}-\varsigma^i\Upsilon^i_{k(r)})dr}\sigma^i dW^i_s\Big\|\Big)^p
$$

$$
\leq 3^{p-1}\|X^i_0\|^p + 3^{p-1}\Big(\frac{1}{c}M_\Upsilon M_\eta\|\varsigma^i\|\Big)^p
$$

$$
+ 3^{p-1}\Big(\sup_{t\in[0,T]}\Big\|\int_0^t e^{\int_s^t (A-\hat{A}^i_{k(r)}-\varsigma^i\Upsilon^i_{k(r)})dr}\sigma^i dW^i_s\Big\|\Big)^p.
$$

Taking expectations on both sides and using (A.9), we obtain

$$
\mathbb{E}^i[(\|\hat{X}^i\|^*_T)^p] \leq 3^{p-1}\|X^i_0\|^p + 3^{p-1}\Big(\frac{1}{c}M_\Upsilon M_\eta\|\varsigma^i\|\Big)^p + 3^{p-1}C_p\Big(\frac{1}{2c}\Big)^{p/2}\|\sigma^i\|^p.
$$

The last bound is independent of $T$. Hence, finalizes the proof.

$\square$

### A.3.2 PROOF OF LEMMA A.2.

Define *the $j$-th macro episodes* $[t^i_{n_j}, t^i_{n_{j+1}})$, $j = 0, 1, 2, \ldots$ where $n_0 = 0$ and

$$
t^i_{n_{j+1}} = \min\{t^i_k > t^i_{n_j} : \det\left(\Sigma^i_{t^i_k}\right) < 0.5\det\left(\Sigma^i_{t^i_{k-1}}\right)\} \tag{A.10}
$$

See Figure 2 for an illustration.

Let $M$ be the number of macro episodes until time $T$.

The proof is carried out in three main steps. In the first step, we bound $K^i_T$ in terms of $M$ and $T$. In the second step, we bound $M$ using the determinant and trace of $(\Sigma^i_0)^{-1}$. Finally, in the third step, we combine the two bounds to complete the argument.

**Step 1: Bounding $K_T^i$ using $M$ and $T$.** By (A.10), the number of episodes contained in $j$-th macro episode is $n_{j+1} - n_j$ and the total number of episodes contained within these $M$ macro episodes is

$$\sum_{j=0}^{M-1} (n_{j+1} - n_j) = n_M.$$

Then $n_M \leq K_T^i$. Let $\widetilde{T}_j^i = \sum_{k=n_j}^{n_{j+1}-1} T_k^i$ be the length of the $j$-th macro episode. By definition, in the $j$-th macro episode $[t_{n_j}^i, t_{n_{j+1}}^i)$, the first $n_{j+1} - n_j - 1$ episodes $\{[t_k^i, t_{k+1}^i)\}_{k \in [n_j, n_{j+1}-1]}$ are triggered by the interval length stopping criterion: $T_k^i = t_k^i - t_{k-1}^i = T_{k-1}^i + 1$. Only the last episode $[t_{n_{j+1}-1}^i, t_{n_{j+1}}^i)$ is triggered by the determinant criterion that $\det(\Sigma_{t_{n_{j+1}}^i}^i) < 0.5 \det(\Sigma_{t_{n_{j+1}-1}^i}^i)$. Hence

$$\widetilde{T}_j^i = \sum_{l=n_j}^{n_{j+1}-2} T_l^i + T_{n_{j+1}-1}^i = \sum_{l=1}^{n_{j+1}-n_j-1} (T_{n_j-1}^i + l) + T_{n_{j+1}-1}^i$$

$$\geq \sum_{l=1}^{n_{j+1}-n_j-1} (l+1) + 1 = 0.5(n_{j+1} - n_j)(n_{j+1} - n_j + 1).$$

Thus $n_{j+1} - n_j < \sqrt{2\widetilde{T}_j^i}$ for all $j = 0, \ldots, M$. Then by Cauchy–Schwarz inequality, we obtain

$$n_M = \sum_{j=0}^{M-1} (n_{j+1} - n_j) \leq \sum_{j=0}^{M-1} \sqrt{2\widetilde{T}_j^i} \leq \sqrt{M \sum_{j=0}^{M-1} 2\widetilde{T}_j^i} = \sqrt{2Mt_{n_M}^i}. \tag{A.11}$$

Denote the number of episodes in the time interval $[t_{n_M}^i, T)$ by $k_T^i$. The episodes in $[t_{n_M}^i, T)$ are $\{[t_k^i, t_{k+1}^i)\}_{k \in [n_M, n_M+k_T^i-1]}$, all triggered by the interval length stopping criterion: $T_k^i = t_k^i - t_{k-1}^i = T_{k-1}^i + 1$. Then,

$$T - t_{n_M}^i = \sum_{l=n_M}^{n_M+k_T^i-1} T_l^i = \sum_{l=1}^{k_T^i} (T_{n_M-1}^i + l)$$

$$\geq \sum_{l=1}^{k_T^i} (1+l) = 0.5 k_T^i (3 + k_T^i),$$

and as a result,

$$k_T^i \leq \sqrt{2(T - t_{n_M}^i)}. \tag{A.12}$$

Combine (A.11) and (A.12), we obtain

$$K_T^i = n_M + k_T^i \leq \sqrt{2Mt_{n_M}^i} + \sqrt{2(T - t_{n_M}^i)}.$$

Consider the function $f(t) = \sqrt{2Mt} + \sqrt{2(T-t)}$, which reaches its maximum $\max_{t \in [0,T)} f(t) = \sqrt{2(M+1)T}$ at $t^* = \frac{MT}{M+1}$. Then

$$K_T^i \leq \sqrt{2Mt_{n_M}^i} + \sqrt{2(T - t_{n_M}^i)} \leq \sqrt{2(M+1)T}. \tag{A.13}$$

**Step 2: Bounding $M$.** In this part, we relate $M$ to the determinant and trace of $(\Sigma_0^i)^{-1}$. To this end, we analyze the behavior of $(\Sigma_t^i)^{-1}$.

By Proposition 3.1, for any episode $k$ and any $t \in [t_k^i, t_{k+1}^i]$,

$$(\Sigma_t^i)^{-1} = (\Sigma_{t_k^i}^i)^{-1} + (\sigma^i(\sigma^i)^T)^{-1} \otimes \int_{t_k^i}^t \hat{X}_s^i (\hat{X}_s^i)^T ds. \tag{A.14}$$

Hence,

$$\det\left((\Sigma_t^i)^{-1}\right) = \det\left((\Sigma_{t_k^i}^i)^{-1}\right) \cdot \det\left(I_{d^2} + \Sigma_{t_k^i}^i\left((\sigma^i(\sigma^i)^T)^{-1} \otimes \int_{t_k^i}^t \hat{X}_s^i(\hat{X}_s^i)^T ds\right)\right).$$

Now, both matrices $\left((\sigma^i(\sigma^i)^T)^{-1} \otimes \int_{t_k^i}^t \hat{X}_s^i(\hat{X}_s^i)^T ds\right)$ and $\Sigma_{t_k^i}^i$ are symmetric positive semi-definite. By Sylvester's identity,

$$\det\left(I_{d^2} + \Sigma_{t_k^i}^i\left((\sigma^i(\sigma^i)^T)^{-1} \otimes \int_{t_k^i}^t \hat{X}_s^i(\hat{X}_s^i)^T ds\right)\right)$$

$$= \det\left(I_{d^2} + (\Sigma_{t_k^i}^i)^{1/2}\left((\sigma^i(\sigma^i)^T)^{-1} \otimes \int_{t_k^i}^t \hat{X}_s^i(\hat{X}_s^i)^T ds\right)(\Sigma_{t_k^i}^i)^{1/2}\right)$$

$$\geq 1.$$

We thus obtain
$$\det\left((\Sigma_t^i)^{-1}\right) \geq \det\left((\Sigma_{t_k^i}^i)^{-1}\right), \qquad \forall t \in [t_k^i, t_{k+1}^i).$$

Then, $\det\left((\Sigma_{t_{k+1}^i}^i)^{-1}\right) \geq \det((\Sigma_{t_k^i}^i)^{-1})$ for each $k$. Since the macro episode starts when the determinant stopping criterion is triggered, for each $j$,

$$\det\left((\Sigma_{t_{n_{j+1}}^i}^i)^{-1}\right) \geq 2\det\left((\Sigma_{t_{n_{j+1}-1}^i}^i)^{-1}\right) \geq 2\det\left((\Sigma_{t_{n_j}^i}^i)^{-1}\right).$$

Then we have

$$\det((\Sigma_T^i)^{-1}) \geq \det((\Sigma_{t_{n_M}^i}^i)^{-1}) \geq 2\det((\Sigma_{t_{n_{M-1}}^i}^i)^{-1}) \geq \cdots > 2^M \det((\Sigma_0^i)^{-1}).$$

By the inequality of arithmetic and geometric means, and since $(\Sigma_t^i)^{-1}$ is symmetric and positive semi-definite, we have for any $t \geq 0$:

$$\left(\frac{\text{tr}((\Sigma_t^i)^{-1})}{d^2}\right)^{d^2} \geq \det((\Sigma_t^i)^{-1}), \tag{A.15}$$

which implies,

$$\frac{\text{tr}((\Sigma_T^i)^{-1})}{d^2} \geq \det((\Sigma_T^i)^{-1})^{1/d^2} \geq 2^{M/d^2} \det((\Sigma_0^i)^{-1})^{1/d^2}. \tag{A.16}$$

Note that we obtained $M$ on the right-hand side. Hence, we now estimate the left-hand side.

Now, recall (A.14). By recursion,

$$(\Sigma_T^i)^{-1} = (\Sigma_{t_k^i}^i)^{-1} + (\sigma^i(\sigma^i)^T)^{-1} \otimes \int_{t_k^i}^T \hat{X}_s^i(\hat{X}_s^i)^T ds$$

$$= (\Sigma_0^i)^{-1} + \sum_{j=0}^k \left((\sigma^i(\sigma^i)^T)^{-1} \otimes \int_{t_j^i}^{t_{j+1}^i \wedge T} \hat{X}_s^i(\hat{X}_s^i)^T ds\right).$$

Hence,

$$\text{tr}((\Sigma_T^i)^{-1}) = \text{tr}((\Sigma_0^i)^{-1}) + \text{tr}\left(\sum_{j=0}^k \left((\sigma^i(\sigma^i)^T)^{-1} \otimes \int_{t_j^i}^{t_{j+1}^i \wedge T} \hat{X}_s^i(\hat{X}_s^i)^T ds\right)\right) \tag{A.17}$$

Next, we have that

$$\text{tr}\left((\sigma^i(\sigma^i)^T)^{-1} \otimes \int_{t_j^i}^{t_{j+1}^i \wedge T} \hat{X}_s^i(\hat{X}_s^i)^T ds\right) = \text{tr}((\sigma^i(\sigma^i)^T)^{-1})\text{tr}\left(\int_{t_j^i}^{t_{j+1}^i \wedge T} \hat{X}_s^i(\hat{X}_s^i)^T ds\right)$$

$$\leq d \cdot \|(\sigma^i(\sigma^i)^T)^{-1}\| \cdot \int_{t_j^i}^{t_{j+1}^i \wedge T} (\|\hat{X}^i\|_T^*)^2 ds.$$

Putting these bounds back in (A.17), we get

$$
\begin{aligned}
\operatorname{tr}((\Sigma_T^i)^{-1}) &\leq \operatorname{tr}((\Sigma_0^i)^{-1}) + d\|(\sigma^i)(\sigma^i)^T)^{-1}\| \sum_{j=0}^{k} \int_{t_j^i}^{t_{j+1}^i \wedge T} (\|\hat{X}^i\|_T^*)^2 ds \\
&= \operatorname{tr}((\Sigma_0^i)^{-1}) + dT\|(\sigma^i)(\sigma^i)^T)^{-1}\| \cdot (\|\hat{X}^i\|_T^*)^2.
\end{aligned}
$$

Plugging in this bound in (A.16), we manage to connect $M$ and the determinant and trace of $(\Sigma_0^i)^{-1}$, as follows:

$$
\operatorname{tr}((\Sigma_0^i)^{-1}) + dT\|(\sigma^i)(\sigma^i)^T)^{-1}\| \cdot (\|\hat{X}^i\|_T^*)^2 \geq d^2 \det((\Sigma_0^i)^{-1})^{1/d^2} 2^{M/d^2}.
$$

Taking the logarithm on both sides and multiplying by $d^2$, we obtain a bound for $M$:

$$
M \leq \frac{d^2}{\log 2} \log \Big( \frac{\operatorname{tr}((\Sigma_0^i)^{-1}) + dT\|(\sigma^i)(\sigma^i)^T)^{-1}\| \cdot (\|\hat{X}^i\|_T^*)^2}{d^2 \det((\Sigma_0^i)^{-1})^{1/d^2}} \Big).
$$

**Step 3: Combining the bounds.** Finally, plug back in (A.13), we have

$$
K_T^i \leq \sqrt{\frac{2d^2 T}{\log 2} \log \Big( \frac{\operatorname{tr}((\Sigma_0^i)^{-1})}{d^2 \det((\Sigma_0^i)^{-1})^{1/d^2}} + \frac{T\|(\sigma^i)(\sigma^i)^T)^{-1}\|}{d \det((\Sigma_0^i)^{-1})^{1/d^2}} (\|\hat{X}^i\|_T^*)^2 \Big) + 2T},
$$

which is,

$$
K_T^i \leq O\Big( d\sqrt{T \log \Big( \frac{T}{d} \|(\sigma^i)(\sigma^i)^T)^{-1}\| \cdot (\|\hat{X}^i\|_T^*)^2 \Big)} \Big).
$$

$\square$

### A.3.3 PROOF OF LEMMA A.3

**Step 1: $\Upsilon^i(A)$ is Lipschitz w.r.t. $A$.**

From the Riccati equation (2.8)

$$
Y \varsigma^i R^i \varsigma^i Y = A^T R^i A + 2Q_{ii}^i,
$$

Define

$$
S^i(A) := A^T R^i A + 2Q_{ii}^i, \qquad M^i := \varsigma^i R^i \varsigma^i.
$$

Then the solution has the explicit form

$$
\Upsilon^i(A) = (M^i)^{-1/2} \left[ (M^i)^{1/2} S^i(A)(M^i)^{1/2} \right]^{1/2} (M^i)^{-1/2}.
$$

For symmetric positive definite matrices $\Upsilon^i(A), \Upsilon^i(\hat{A})$, we have the following inequality

$$
\begin{aligned}
&\|\Upsilon^i(A) - \Upsilon^i(\hat{A})\| \\
&\leq \|(M^i)^{-1/2}\|^2 \left\| \left[ (M^i)^{1/2} S^i(A)(M^i)^{1/2} \right]^{1/2} - \left[ (M^i)^{1/2} S^i(\hat{A})(M^i)^{1/2} \right]^{1/2} \right\| \\
&\leq \|(M^i)^{-1/2}\|^2 \frac{\|(M^i)^{1/2} [S^i(A) - S^i(\hat{A})] (M^i)^{1/2}\|}{\sqrt{\lambda_{\min}((M^i)^{1/2} S^i(A)(M^i)^{1/2})} + \sqrt{\lambda_{\min}((M^i)^{1/2} S^i(\hat{A})(M^i)^{1/2})}}.
\end{aligned}
$$

The last inequality follows from the identity, where we denote $Y := (M^i)^{1/2} S^i(A)(M^i)^{1/2}$, $Z := (M^i)^{1/2} S^i(\hat{A})(M^i)^{1/2}$,

$$
(Y^{1/2} - Z^{1/2})Y^{1/2} + Z^{1/2}(Y^{1/2} - Z^{1/2}) = Y - Z.
$$

Then

$$
\|Y^{1/2} - Z^{1/2}\| \leq \frac{\|Y - Z\|}{\sqrt{\lambda_{\min}(Y)} + \sqrt{\lambda_{\min}(Z)}}
$$

For any $A \in \Theta$, we have the uniform lower bound

$$
\lambda_{\min}((M^i)^{1/2} S^i(A)(M^i)^{1/2}) \geq \lambda_{\min}((M^i)^{1/2} Q_{ii}^i (M^i)^{1/2}) > 0,
$$

so the denominator is bounded below by

$$2\sqrt{\lambda_{\min}((M^i)^{1/2}Q_{ii}^i(M^i)^{1/2})}.$$

Hence,

$$\|\Upsilon^i(A) - \Upsilon^i(\hat{A})\| \leq \frac{\|M^i\| \cdot \|(M^i)^{-1/2}\|^2}{2\sqrt{\lambda_{\min}((M^i)^{1/2}Q_{ii}^i(M^i)^{1/2})}} \, \|S^i(A) - S^i(\hat{A})\|.$$

Using

$$\|S^i(A) - S^i(\hat{A})\| = \|A^T R^i A - \hat{A}^T R^i \hat{A}\| \leq \|R^i\| \, (\|A\| + \|\hat{A}\|) \, \|A - \hat{A}\|,$$

it follows that

$$\|\Upsilon^i(A) - \Upsilon^i(\hat{A})\| \leq \frac{\|M^i\| \cdot \|(M^i)^{-1/2}\|^2 \, \|R^i\|}{2\sqrt{\lambda_{\min}((M^i)^{1/2}Q_{ii}^i(M^i)^{1/2})}} \, (\|A\| + \|\hat{A}\|) \, \|A - \hat{A}\|.$$

Finally, since $\|A\|, \|\hat{A}\| \leq M_A$ on $\Theta$, it follows that

$$\|\Upsilon^i(A) - \Upsilon^i(\hat{A})\| \leq \frac{\|M^i\| \cdot \|(M^i)^{-1/2}\|^2 \, \|R^i\|}{\sqrt{\lambda_{\min}((M^i)^{1/2}Q_{ii}^i(M^i)^{1/2})}} \, M_A \, \|A - \hat{A}\|.$$

Let $L_\Upsilon^i := \frac{\|M^i\| \cdot \|(M^i)^{-1/2}\|^2 \, \|R^i\|}{\sqrt{\lambda_{\min}((M^i)^{1/2}Q_{ii}^i(M^i)^{1/2})}} \, M_A$, we have $\|\Upsilon^i(A) - \Upsilon^i(\hat{A})\| \leq L_\Upsilon^i \|A - \hat{A}\|$.

**Step 2:** $\eta^i(A)$ **is Lipschitz w.r.t.** $A$

Recall from (2.6) that the block matrix $\mathcal{B}$ depends on $A$ through

$$\mathcal{B}(A) := \big(\mathcal{B}_{ij}(A)\big)_{i,j=1}^N, \qquad \mathcal{B}_{ij}(A) := -Q_{ij}^i - \frac{1}{2}\,\delta_{ij}\, A^T R^i A \in \mathbb{R}^{d \times d}.$$

For any $A, \hat{A} \in \Theta$, we have

$$\|\mathcal{B}(A) - \mathcal{B}(\hat{A})\| \leq \frac{1}{2} \max_j \|A^T R^i A - \hat{A}_{k(t)}^{i\,T} R^i \hat{A}\|$$

$$\leq \frac{1}{2} \|R^i\| \, (\|A\| + \|\hat{A}\|) \, \|A - \hat{A}\|$$

$$\leq M_A \, \|R^i\| \, \|A - \hat{A}\|.$$

From $\mathcal{B}(A)\,\eta(A) = \boldsymbol{p}$, the resolvent identity yields

$$\eta(A) - \eta(\hat{A}) = \mathcal{B}(A)^{-1} \big[\, \mathcal{B}(\hat{A}) - \mathcal{B}(A) \,\big]\, \mathcal{B}(\hat{A})^{-1}\, \boldsymbol{p},$$

$$\|\eta(A) - \eta(\hat{A})\| \leq \big(\sup_{A' \in \Theta} \|\mathcal{B}(A')^{-1}\|\big)^2 \|\mathcal{B}(\hat{A}) - \mathcal{B}(A)\| \, \|\boldsymbol{p}\|.$$

From Assumption 2.1 and the fact that $R^i$ is positive definite, there exists $\rho^i > 0$ such that, for any $A \in \Theta$,

$$\lambda_{\min}\big(Q_{ii}^i + \frac{1}{2}A^T R^i A\big) \, - \, \sum_{j \neq i} \|Q_{ij}^i\| \geq \lambda_{\min}(Q_{ii}^i) \, - \, \sum_{j \neq i} \|Q_{ij}^i\| \geq \rho^i,$$

Then

$$\lambda_{\min}\big(-\mathcal{B}(A)\big) \, \geq \, \rho^i, \qquad \|\mathcal{B}(A)^{-1}\| \, \leq \, \frac{1}{\rho^i}.$$

Combining these bounds gives

$$\|\eta(A) - \eta(\hat{A})\| \leq \frac{1}{(\rho^i)^2} \, \|\mathcal{B}(\hat{A}) - \mathcal{B}(A)\| \, \|\boldsymbol{p}\|$$

$$\leq \frac{M_A \, \|R^i\| \, \|\boldsymbol{p}\|}{(\rho^i)^2} \, \|A - \hat{A}\|.$$

Let $L_\eta^i := \frac{M_A \, \|R^i\| \, \|\boldsymbol{p}\|}{(\rho^i)^2}$, we have $\|\eta^i(A) - \eta^i(\hat{A})\| \leq L_\eta^i \|A - \hat{A}\|$.

$$\square$$

### A.4 Proof of Proposition 3.3

We now bound the three parts of the regret.

#### A.4.1 Proof of (1): Regret bound of $R_0^i(\hat{\alpha}^i, T)$.

Note that we can rewrite the regret in the summation form using $K_T^i$, provided in (A.5), and bound it as follows,

$$R_0^i(\hat{\alpha}^i, T) = \mathbb{E}^i \Big[ \sum_{k=0}^{K_T^i} \left( T_k^i \wedge (T - t_k^i) \right) \lambda^i(\hat{A}_k^i) - T\lambda^i(A) \Big]$$

$$\leq \mathbb{E}^i \Big[ \sum_{k=0}^{\infty} \mathbb{1}_{\{t_k \leq T\}} T_k^i \lambda^i(\hat{A}_k^i) \Big] - T\mathbb{E}^i[\lambda^i(A)] \tag{A.18}$$

$$= \sum_{k=0}^{\infty} \mathbb{E}^i \Big[ \mathbb{1}_{\{t_k \leq T\}} T_k^i \lambda^i(\hat{A}_k^i) \Big] - T\mathbb{E}^i[\lambda^i(A)].$$

Without loss of generality, we assume that $\lambda^i(\hat{A}_k^i)$ and $\lambda^i(A)$ are nonnegative for all $k \in \mathbb{N}$. Otherwise, from Assumption 3.1, their absolute value are bounded by $M_\lambda$. We may add $M_\lambda$ to both $\lambda^i(\hat{A}_k^i)$ and $\lambda^i(A)$ to ensure non-negativity.

Recall the definition of episodes from (3.3) for $k \geq 1$. Then, $T_k^i \leq T_{k-1}^i + 1$. Also, recall that the value in the game is nonnegative due to the quadratic cost structure and Assumption 2.1. Hence,

$$\mathbb{E}^i \Big[ \mathbb{1}_{\{t_k \leq T\}} T_k^i \lambda^i(\hat{A}_k^i) \Big] \leq \mathbb{E}^i \Big[ \mathbb{1}_{\{t_k \leq T\}} (T_{k-1}^i + 1)\lambda^i(\hat{A}_k^i) \Big] = \mathbb{E}^i \Big[ \mathbb{1}_{\{t_k \leq T\}} (T_{k-1}^i + 1)\lambda^i(A) \Big], \tag{A.19}$$

where the last equality follows from the tower property, by conditioning on $\hat{\mathcal{F}}_{t_k^i}^i$. Indeed, $t_k^i$, $T_{k-1}^i$, and $\hat{A}_k^i$ are all $\hat{\mathcal{F}}_{t_k^i}^i$-measurable, and $\hat{A}_k^i$ and $A$ share the same (posterior) distribution given $\hat{\mathcal{F}}_{t_k^i}^i$.

For $k = 0$, from (3.2), $T_0^i \leq 2$. By the same argument as above,

$$\mathbb{E}^i[T_0^i \lambda^i(\hat{A}_k^i)] \leq 2\mathbb{E}^i[\lambda^i(\hat{A}_k^i)] = 2\mathbb{E}^i[\lambda^i(A)].$$

Plugging in (A.19) in (A.18), we get

$$R_0^i(\hat{\alpha}^i, T) \leq \sum_{k=1}^{\infty} \mathbb{E}^i \Big[ \mathbb{1}_{\{t_k \leq T\}} (T_{k-1}^i + 1)\lambda^i(A) \Big] + 2\mathbb{E}^i[\lambda^i(A)] - T\mathbb{E}^i[\lambda^i(A)]$$

$$= \mathbb{E}^i \Big[ \sum_{k=1}^{K_T^i} (T_{k-1}^i + 1)\lambda^i(A) \Big] + 2\mathbb{E}^i[\lambda^i(A)] - T\mathbb{E}^i[\lambda^i(A)]$$

$$= \mathbb{E}^i[(K_T^i + 2)\lambda^i(A)] + \mathbb{E}^i \Big[ \Big( \sum_{k=0}^{K_T^i} T_{k-1}^i - T \Big)\lambda^i(A) \Big]$$

$$\leq M_\lambda \mathbb{E}^i[K_T^i] + 2M_\lambda,$$

where in the last inequality, we used the bound from Assumption 3.1, $\lambda^i(A) \leq M_\lambda$, and the bound $\sum_{k=0}^{K_T^i} T_{k-1}^i \leq T$.

Finally, applying first, Lemma A.2, Jensen's inequality, and finally, Lemma A.1, we obtain

$$R_0^i(\hat{\alpha}^i, T) \leq O\Big( d\mathbb{E}^i \Big[ \sqrt{T \log \Big( \frac{T}{d} \|((\sigma^i)(\sigma^i)^T)^{-1}\| \cdot (\|\hat{X}^i\|_T^*)^2 \Big)} \Big] \Big)$$

$$\leq O\Big( d\sqrt{T \log \Big( \frac{T}{d} \|((\sigma^i)(\sigma^i)^T)^{-1}\| \cdot \mathbb{E}^i \big[ (\|\hat{X}^i\|_T^*)^2 \big] \Big)} \Big)$$

$$\leq O(d\sqrt{T \log(T)}).$$

$\square$

### A.4.2 PROOF OF (2): REGRET BOUND OF $R_1^i(\hat{\alpha}^i, T)$.

From the form of $v^i$, provided in Proposition 2.1, we get that

$$R_1^i(\hat{\alpha}^i, T) = \mathbb{E}^i \left[ v^i(X_0^i) - v^i(\hat{X}_T^i) \right]$$

$$= \mathbb{E}^i \left[ \frac{1}{2}(X_0^i)^T \Lambda^i X_0^i + (\rho^i)^T X_0^i - \frac{1}{2}(\hat{X}_T^i)^T \Lambda^i \hat{X}_T^i - (\rho^i)^T \hat{X}_T^i \right].$$

Take the form of $\Lambda^i$ and $\rho^i$ in Proposition 2.1. From Assumption 3.1, $\Upsilon^i$ and $\eta^i$ are bounded, then there exists $M_\Lambda, M_\rho > 0$, s.t.

$$\|\Lambda^i\| = \|R^i(\varsigma^i \Upsilon^i + A)\| \le \|R^i\|(\frac{1}{2}\|\sigma^i\|^2 M_\Upsilon + M_A) \le M_\Lambda, \tag{A.20}$$

$$\|\rho^i\| = \|R^i \varsigma^i \Upsilon^i \eta^i\| \le \|R^i\| \cdot \|\varsigma^i\| \cdot M_\Upsilon M_\eta \le M_\rho.$$

Thus

$$R_1^i(\hat{\alpha}^i, T) \le \frac{1}{2} M_\Lambda \cdot \|X_0^i\|^2 + M_\rho \cdot \|X_0^i\| + \frac{1}{2} M_\Lambda \cdot (\|\hat{X}^i\|_T^*)^2 + M_\rho \cdot \|\hat{X}^i\|_T^* \le \|\sigma^i\|^2 O(1).$$

$\square$

### A.4.3 PROOF OF (3): REGRET BOUND OF $R_2^i(\hat{\alpha}^i, T)$.

Using again the form of $v^i$, provided in Proposition 2.1 and the bounds in (A.20), we get that

$$R_2^i(\hat{\alpha}^i, T) = \mathbb{E}^i \Big[ \int_0^T (\nabla v^i(\hat{X}_t^i))^T (A - \hat{A}_{k(t)}^i) \hat{X}_t^i dt \Big]$$

$$= \mathbb{E}^i \Big[ \int_0^T \big( \Lambda^i(\hat{A}_{k(t)}^i)\hat{X}_t^i + \rho^i(\hat{A}_{k(t)}^i) \big)^T (A - \hat{A}_{k(t)}^i) \hat{X}_t^i dt \Big]$$

$$\le \mathbb{E}^i \Big[ \int_0^T (M_\Lambda \|\hat{X}_t^i\| + M_\rho) \cdot \|(A - \hat{A}_{k(t)}^i)\| \cdot \|\hat{X}_t^i\| dt \Big].$$

By Cauchy–Schwarz inequality, [3]

$$\mathbb{E}^i \Big[ \int_0^T \|(A - \hat{A}_{k(t)}^i)\| \cdot (M_\Lambda \|\hat{X}_t^i\|^2 + M_\rho \|\hat{X}_t^i\|) dt \Big]$$

$$\le \left( \int_0^T \mathbb{E}^i \Big[ \|A - \hat{A}_{k(t)}^i\|^2 \Big] dt \right)^{\frac{1}{2}} \left( \int_0^T \mathbb{E}^i \Big[ M_\Lambda \|\hat{X}_t^i\|^4 + 2M_\Lambda M_\rho \|\hat{X}_t^i\|^3 + M_\rho \|\hat{X}_t^i\|^2 \Big] dt \right)^{\frac{1}{2}}$$

From Proposition 3.4,

$$\left( \int_0^T \mathbb{E}^i \Big[ \|A - \hat{A}_{k(t)}^i\|^2 \Big] dt \right)^{1/2} \le O(\|\sigma^i\|\sqrt{d\log(T)}).$$

From Lemma A.1,

$$\int_0^T \mathbb{E}^i \Big[ M_\Lambda \|\hat{X}_t^i\|^4 + 2M_\Lambda M_\rho \|\hat{X}_t^i\|^3 + M_\rho \|\hat{X}_t^i\|^2 \Big] dt \le \|\sigma^i\|^4 O(T).$$

Combining both bounds yields

$$R_2^i(\hat{\alpha}^i, T) \le O(\|\sigma^i\|^3 \sqrt{dT \log(T)}).$$

$\square$

---

[3]For this proof we can follow similar steps to Gagrani et al. (2021) and estimate the expectation of the integral below by $\mathbb{E}^i \Big[ \int_0^T \|(\hat{X}_t^i)^T (A - \hat{A}_{k(t)}^i)\hat{X}_t^i\| dt \Big] \le \mathbb{E}^i \Big[ \|\hat{X}^i\|_T^* \int_0^T \|(\Sigma_t^i)^{-\frac{1}{2}}(A^{(v)} - \hat{A}_{k(t)}^{i,(v)})\| \cdot \|(I_d \otimes \hat{X}_t^i)(\Sigma_t^i)^{\frac{1}{2}}\| dt \Big] \le O\Big(d^2 \|\sigma^i\|^2 \sqrt{(T + d\sqrt{T\log(T)})\log(T)}\Big)$. Here one utilizes the fact that $(A - \hat{A})$ appears in the integral multiplied by $\hat{X}$, which enables us to make the connection to $(\Sigma_t^i)^{-\frac{1}{2}}(A^{(v)} - \hat{A}_{k(t)}^{i,(v)})$, a difference between Gaussian vectors, and $\|(I_d \otimes \hat{X}_t^i)(\Sigma_t^i)^{\frac{1}{2}}\|$, which is related to the derivative of $\det((\Sigma_t^i)^{-1})$. However, here we simply use the bound from Proposition 3.4, as in the proof of Proposition Proposition 3.5 we cannot use the same approach as above. Over there we are facing $\|A - \hat{A}\|\|X\|$ with the process $X$, rather than $\hat{X}$. So to achieve two goals simultaneously, we prove both bounds $R_2$ and Proposition 3.5 using the same bound.

## A.5 PROOF OF PROPOSITION 3.4

Consider $\mathbb{E}^i\big[\|A - \hat{A}^i_{k(t)}\|^2\big]$. Rewrite in vectorized form,

$$
\begin{aligned}
\mathbb{E}^i\big[\|A - \hat{A}^i_{k(t)}\|^2\big] &= \sum_{j=1}^d \sum_{l=1}^d \left( A_{j,l} - (\hat{A}^i_{k(t)})_{j,l} \right)^2 \\
&= \sum_{j=1}^d \sum_{l=1}^d \left( (A^{(v)})_{(j-1)d+l} - (\hat{A}^{i,(v)}_{k(t)})_{(j-1)d+l} \right)^2 \\
&= \mathbb{E}^i\big[\|A^{(v)} - \hat{A}^{i,(v)}_{k(t)}\|^2\big].
\end{aligned}
$$

By the triangle inequality,

$$
\mathbb{E}^i\big[\|A^{(v)} - \hat{A}^{i,(v)}_{k(t)}\|^2\big] \le 2\mathbb{E}\big[\|A^{(v)} - \mu_{t_k}\|^2\big] + 2\mathbb{E}\big[\|\mu_{t_k} - \hat{A}^{i,(v)}_{k(t)}\|^2\big].
$$

where

$$
\begin{aligned}
\mathbb{E}\big[\|A^{(v)} - \mu_{t_k}\|^2\big] &= \mathbb{E}\left[ \mathbb{E}\left[ \|A^{(v)} - \mu_{t_k}\|^2 | \mathcal{F}^i_{t_k} \right] \right] = \mathbb{E}[\mathrm{tr}(\Sigma_{t_k})], \\
\mathbb{E}\big[\|\mu_{t_k} - \hat{A}^{i,(v)}_{k(t)}\|^2\big] &= \mathbb{E}[\mathrm{tr}(\Sigma_{t_k})].
\end{aligned}
$$

Then

$$
\int_0^T \mathbb{E}^i\big[\|A - \hat{A}^i_{k(t)}\|^2\big]\, dt \le 4\mathbb{E}^i\Big[ \int_0^T \mathrm{tr}(\Sigma^i_{t_{k(t)}})dt \Big] = 4\int_0^T \mathbb{E}^i\big[\mathrm{tr}(\Sigma^i_{t_{k(t)}})\big]dt,
$$

where the last equality follows from Fubini's theorem.

Consider the alternative algorithm OS (one-shot at 0): we draw

$$
\tilde{A}^i \sim \mathcal{N}(\mu^i_0, \Sigma^i_0)
$$

once at time 0, and keep it fixed throughout the horizon $[0, T]$. We claim that

$$
\mathbb{E}^i\left[ \int_0^T \mathrm{tr}(\Sigma^i_t)\, dt \right] = \mathbb{E}^i\left[ \int_0^T \mathrm{tr}(\tilde{\Sigma}^i_t)\, dt \right],
$$

where $\tilde{\Sigma}^i_t$ denotes the posterior covariance under OS.

The key observation is that, under OS,

$$
\tilde{A}^i \,|\, \mathcal{F}^i_{t^i_k} \sim \mathcal{N}(\mu^i_{t^i_k}, \Sigma^i_{t^i_k}),
$$

which coincides with the posterior distribution used in Thompson Sampling at time $t^i_k$. Hence, conditional on $\mathcal{F}^i_{t^i_k}$, the law of the posterior under OS is the same, and thus the conditional expected values of the integral on the interval $[t^i_k, t^i_{k+1})$ coincide. Applying the tower property,

$$
\begin{aligned}
\mathbb{E}^i\left[ \int_0^T \mathrm{tr}(\Sigma^i_{t^i_k})\, dt \right] &= \mathbb{E}^i\left[ \sum_{k=0}^{K^i_T} \int_{t^i_k}^{t^i_{k+1} \wedge T} \mathrm{tr}(\Sigma^i_{t^i_k})\, dt \right] \\
&= \mathbb{E}^i\left[ \sum_{k=0}^\infty 1_{\{t^i_k \le T\}} \int_{t^i_k}^{t^i_{k+1} \wedge T} \mathrm{tr}(\Sigma^i_{t^i_k})\, dt \right] \\
&= \mathbb{E}^i\left[ \sum_{k=0}^\infty \mathbb{E}^i\left[ 1_{\{t^i_k \le T\}} \int_{t^i_k}^{t^i_{k+1} \wedge T} \mathrm{tr}(\Sigma^i_{t^i_k})\, dt \,\Big|\, \mathcal{F}^i_{t^i_k} \right] \right] \\
&= \mathbb{E}^i\left[ \sum_{k=0}^\infty 1_{\{t^i_k \le T\}} \mathbb{E}^i\left[ \int_{t^i_k}^{t^i_{k+1} \wedge T} \mathrm{tr}(\Sigma^i_{t^i_k})\, dt \,\Big|\, \mathcal{F}^i_{t^i_k} \right] \right].
\end{aligned}
$$

Conditioned[4] on $\mathcal{F}^i_{t^i_k}$, the posterior distribution under OS and TS is identical, then

$$\mathbb{E}^i\left[\int_0^T \mathrm{tr}(\Sigma^i_{t^i_k})\,dt\right] = \mathbb{E}^i\left[\sum_{k=0}^\infty 1_{\{t^i_k \leq T\}}\mathbb{E}^i\left[\int_{t^i_k}^{t^i_{k+1}\wedge T}\mathrm{tr}(\tilde{\Sigma}^i_{t^i_k})\,dt \;\middle|\; \mathcal{F}^i_{t^i_k}\right]\right]$$

$$= \mathbb{E}^i\left[\int_0^T \mathrm{tr}(\tilde{\Sigma}^i_{t^i_k})\,dt\right].$$

Recall from Assumption 3.1 that $A^{(v)}, \tilde{A}^{i,(v)} \in \Theta^{(v)}$. By assumption, we have

$$\left\|e^{(A-\tilde{A}^i-\varsigma^i\tilde{\Upsilon}^i)t}\right\| \leq e^{-ct}, \qquad t \geq 0,$$

for some constant $c > 0$, where $\tilde{\Upsilon}^i = \Upsilon^i(\tilde{A}^i)$ depends on $\tilde{A}^i$.

For any eigenvalue $\lambda$ of $(A - \tilde{A}^i - \varsigma^i\tilde{\Upsilon}^i)$, then

$$|e^{\lambda t}| = e^{t\Re(\lambda)} \leq e^{-ct}, \qquad t \geq 0,$$

where $\Re(\lambda)$ denotes the real part of $\lambda$. Hence $\Re(\lambda) \leq -c < 0$ for all eigenvalues $\lambda$, which shows that $A - \tilde{A}^i - \varsigma^i\tilde{\Upsilon}^i$ is Hurwitz.

Then, by Proposition 2.3 in Bardi & Priuli (2014), $\tilde{X}^i_t$ has a unique stationary distribution $\tilde{m}^i$ given by a multivariate Gaussian with mean $\tilde{\theta}^i := -(A - \tilde{A}^i - \varsigma^i\tilde{\Upsilon}^i)^{-1}\varsigma^i\tilde{\Upsilon}^i\tilde{\eta}^i$ and covariance matrix $\tilde{V}^i$ which satisfies

$$(A - \tilde{A}^i - \varsigma^i\tilde{\Upsilon}^i)\tilde{V}^i + \tilde{V}^i(A - \tilde{A}^i - \varsigma^i\tilde{\Upsilon}^i)^T + \sigma^i(\sigma^i)^T = 0,$$

and $\tilde{X}^i_t$ is ergodic w.r.t. such a measure $\tilde{m}^i$. Thus,

$$\frac{1}{t}\int_0^t \tilde{X}^i_s(\tilde{X}^i_s)^T ds \xrightarrow{\text{a.s.}} \tilde{V}^i + \tilde{\theta}^i(\tilde{\theta}^i)^T := \tilde{U}^i.$$

Next, we want to prove $\lambda_{\min}(\tilde{U}^i) > 0$. Write $B := A - \tilde{A}^i - \varsigma^i\tilde{\Upsilon}^i$, and observe that $\|B\| \leq 2M(A) + \frac{1}{2}\|\sigma_i\|^2 M(\Upsilon^i) =: L$. Then $\tilde{V}^i$ has the explicit representation:

$$\tilde{V}^i = \int_0^\infty e^{Bt}\sigma^i(\sigma^i)^T e^{B^T t}dt.$$

For any unit vector $u$, the quadratic form can be written as

$$u^T\tilde{V}^i u = \int_0^\infty u^T e^{Bt}\sigma^i(\sigma^i)^T e^{B^T t}u\,dt$$

$$= \int_0^\infty \|(\sigma^i)^T e^{B^T t}u\|^2 dt$$

$$\geq \lambda_{\min}(\sigma^i(\sigma^i)^T)\int_0^\infty \|e^{B^T t}u\|^2 dt.$$

Since $\|e^{B^T t}u\| \geq \sigma_{\min}(e^{B^T t})\|u\| = \sigma_{\min}(e^{B^T t})$, where $\sigma_{\min}(\cdot)$ is the smallest singular value of a matrix,

$$u^T\tilde{V}^i u \geq \lambda_{\min}(\sigma^i(\sigma^i)^T)\int_0^\infty \sigma_{\min}(e^{B^T t})^2\|u\|^2 dt$$

$$= \lambda_{\min}(\sigma^i(\sigma^i)^T)\int_0^\infty \sigma_{\min}(e^{B^T t})^2 dt.$$

---

[4]We can describe this setup more precisely by using two different probability measures on the same probability space and filtration: $\{\mathcal{F}^i_{t_k}\}_k$, which both support the same process. Under the original probability measure, written as $\mathbb{P}^i$, the process $\Sigma^i_{t_k}$ is associated with the TS sampling algorithm. Under a different measure, which we use for the OS algorithm and denote as $\tilde{\mathbb{P}}^i$, the process follows a different distribution. However, to avoid cumbersome notation and arguments, we abuse notation and use a different notation for the process under OS $\{\tilde{\Sigma}^i_{t_k}\}_k$. This way, we can talk about the two algorithms using different process names rather than constantly referring to different probability measures or expectations.

Take $\tau = \frac{1}{4L}$, $\|B\| \leq L$, then $\|B\| \cdot t \leq \frac{1}{4}$, we have

$$\|e^{Bt} - I\| \leq e^{\|B\|t} - 1 \leq 2\|B\|t, \qquad \forall 0 \leq t \leq \tau.$$

In addition, due to the fact that the smallest singular value $\sigma_{\min}(\cdot)$ is 1-Lipschitz with respect to the Frobenius norm, we have

$$\sigma_{\min}(e^{B^T t}) \geq \sigma_{\min}(I) - \|e^{Bt} - I\| \geq 1 - 2\|B\|t \geq 1 - 2Lt.$$

Since for any $t \in [0, \tau]$, $1 - 2Lt \geq \frac{1}{2}$,

$$u^T \tilde{V}^i u \geq \lambda_{\min}(\sigma^i(\sigma^i)^T) \int_0^\tau (\frac{1}{2})^2 dt = \frac{\lambda_{\min}(\sigma^i(\sigma^i)^T)}{8L}.$$

Then $\lambda_{\min}(\tilde{V}^i) \geq \frac{\lambda_{\min}(\sigma^i(\sigma^i)^T)}{8L}$, by Weyl's inequality,

$$\lambda_{\min}(\tilde{U}^i) = \lambda_{\min}\Big(\tilde{V}^i + \big((A - \tilde{A}^i - \varsigma^i \tilde{\Upsilon}^i)^{-1}\varsigma^i \tilde{\Upsilon}^i \tilde{\eta}^i\big)\big((A - \tilde{A}^i - \varsigma^i \tilde{\Upsilon}^i)^{-1}\varsigma^i \tilde{\Upsilon}^i \tilde{\eta}^i\big)^T\Big)$$

$$\geq \lambda_{\min}(\tilde{V}^i) + \lambda_{\min}\Big(\big((A - \tilde{A}^i - \varsigma^i \tilde{\Upsilon}^i)^{-1}\varsigma^i \tilde{\Upsilon}^i \tilde{\eta}^i\big)\big((A - \tilde{A}^i - \varsigma^i \tilde{\Upsilon}^i)^{-1}\varsigma^i \tilde{\Upsilon}^i \tilde{\eta}^i\big)^T\Big)$$

$$\geq \lambda_{\min}(\tilde{V}^i) \geq \frac{\lambda_{\min}(\sigma^i(\sigma^i)^T)}{8L} > 0.$$

Let

$$I_t^i := \frac{1}{t} \int_0^t \tilde{X}_s^i (\tilde{X}_s^i)^T ds.$$

For a tolerance $R = \frac{1}{2}\lambda_{\min}(\tilde{U}^i) > 0$, define the event

$$B_t^i := \Big\{ \|I_t^i - \tilde{U}^i\| \leq R \Big\}.$$

By Weyl's inequality,

$$\lambda_{\min}(I_t^i) \geq \lambda_{\min}(\tilde{U}^i) - \|I_t^i - \tilde{U}^i\| \geq \lambda_{\min}(\tilde{U}^i) - R \quad \text{on } B_t^i.$$

Recall $\tilde{\Sigma}_t^i = \big[(\Sigma_0^i)^{-1} + ((\sigma^i)(\sigma^i)^T)^{-1} \otimes (t \cdot I_t^i)\big]^{-1} \preceq \big(((\sigma^i)(\sigma^i)^T)^{-1} \otimes (t \cdot I_t^i)\big)^{-1}$, so

$$\mathrm{tr}(\tilde{\Sigma}_t^i) \leq \mathrm{tr}\big(((\sigma^i)(\sigma^i)^T) \otimes (t \cdot I_t^i)^{-1}\big) = \mathrm{tr}\big((\sigma^i)(\sigma^i)^T\big) \mathrm{tr}\big((t \cdot I_t^i)^{-1}\big) \leq \frac{d\|\sigma^i\|^2}{t \cdot \lambda_{\min}(I_t^i)}.$$

Therefore, on $B_t^i$,

$$\mathrm{tr}(\tilde{\Sigma}_t^i) \leq \frac{d\|\sigma^i\|^2)}{t\big(\lambda_{\min}(\tilde{U}^i) - R\big)} = \frac{2d\|\sigma^i\|^2}{t\lambda_{\min}(\tilde{U}^i)}.$$

On $(B_t^i)^c$ we simply use monotonicity $\tilde{\Sigma}_t^i \preceq \Sigma_0^i$ to get $\mathrm{tr}(\tilde{\Sigma}_t^i) \leq \mathrm{tr}(\Sigma_0^i)$.

Let $\delta_t^i := \mathbb{P}\big((B_t^i)^c\big)$, taking expectations,

$$\mathbb{E}\big[\mathrm{tr}(\tilde{\Sigma}_t^i)\big] \leq \frac{d\|\sigma^i\|^2}{t(\lambda_{\min}(\tilde{U}^i) - R)} (1 - \delta_t^i) + \delta_t^i \mathrm{tr}(\Sigma_0^i) \leq \frac{2d}{\lambda_{\min}(\tilde{U}^i)} \cdot \frac{1}{t} + \delta_t^i \mathrm{tr}(\Sigma_0^i)$$

The next step is to prove $\delta_t^i \leq 2d^2 \exp(-C^* t)$ for some constant $C^* > 0$.

Denote by $(\tilde{X}_t^i)_j$ as the $j$-th element of $\tilde{X}_t^i$, by $(\tilde{\theta}^i)_j$ as the $j$-th element of $\tilde{\theta}^i$, and by $(\tilde{V}^i)_{j,k}$ and $(\tilde{U}^i)_{j,k}$ as the $(j, k)$-th entries of the matrices $\tilde{V}^i$ and $\tilde{U}^i$. By Theorem 2.3 in Cattiaux & Guillin (2008), for any $1 \leq j, k \leq d$, we have that

$$\mathbb{P}\Big(\Big|\frac{1}{t}\int_0^t (\tilde{X}_s^i)_j (\tilde{X}_s^i)_k ds - (\tilde{U}^i)_{j,k}\Big| \geq \frac{R}{d}\Big) \leq 2\exp(-tH^*(\frac{R}{d} + (\tilde{U}^i)_{j,k})),$$

where $H^*(a) := \sup_{\lambda > 0}\left\{\lambda a - \log\left(\mathbb{E}_{\tilde{m}^i}[e^{\lambda(\tilde{X}^i)_j(\tilde{X}^i)_k}]\right)\right\}$. Recall that $\tilde{m}^i$ is the stationary measure of $\tilde{X}^i$. Then

$$\mathbb{E}_{\tilde{m}^i}[e^{\lambda(\tilde{X}^i)_j(\tilde{X}^i)_k}] = \frac{1}{\sqrt{1 - 2\lambda(\tilde{V}^i)_{j,k} - \lambda^2\left((\tilde{V}^i)_{j,j}(\tilde{V}^i)_{k,k} - (\tilde{V}^i)_{j,k}^2\right)}}$$
$$\cdot \exp\left(\frac{\lambda(\tilde{\theta}^i)_j(\tilde{\theta}^i)_k + \frac{1}{2}\lambda^2\left((\tilde{\theta}^i)_j^2\tilde{V}^i)_{k,k} + (\tilde{\theta}^i)_k^2\tilde{V}^i)_{j,j}\right)}{1 - 2\lambda(\tilde{V}^i)_{j,k} - \lambda^2\left((\tilde{V}^i)_{j,j}(\tilde{V}^i)_{k,k} - (\tilde{V}^i)_{j,k}^2\right)}\right)$$

This expression is valid for all $\lambda$ such that

$$1 - 2\lambda(\tilde{V}^i)_{j,k} - \lambda^2\left((\tilde{V}^i)_{j,j}(\tilde{V}^i)_{k,k} - (\tilde{V}^i)_{j,k}^2\right) > 0. \tag{A.21}$$

Take

$$\delta := \frac{1}{2\lambda_{\max}(\tilde{V}^i)} < \frac{1}{\sqrt{(\tilde{V}^i)_{j,j}(\tilde{V}^i)_{k,k}}}, \qquad \forall j, k$$

Since $\lambda_{\max}(\tilde{V}^i) \le \mathrm{tr}(\tilde{V}^i) = \int_0^\infty \|e^{\tilde{M}^i t}\sigma^i\|^2 \le \int_0^\infty e^{-2ct}\|\sigma^i\|^2 = \frac{\|\sigma^i\|^2}{2c}$, $\delta$ is well defined and $\delta > 0$. For every $\lambda \in (0, \delta)$, the condition (A.21) holds.

Define $\phi_{j,k}(\lambda) := \log\left(\mathbb{E}_{\tilde{m}^i}[e^{\lambda(\tilde{X}^i)_j(\tilde{X}^i)_k}]\right)$ and $F_{j,k}(\lambda) := \lambda(\frac{R}{d} + (\tilde{U}_{j,k}) - \phi_{j,k}(\lambda))$. Let us Taylor expand $\phi_{j,k}(\lambda)$ on $(0, \delta)$. Observe that $\phi_{j,k}(0) = 0, \phi'_{j,k}(0) = (\tilde{U}^i)_{j,k}$ and $\phi''_{j,k}(0) = \mathrm{Var}_{\tilde{m}^i}[(\tilde{X}^i)_j(\tilde{X}^i)_k] \ge 0$. By the continuity of $\phi''_{j,k}(\cdot)$, we can take

$$M_{\phi_{j,k}} := \sup_{0 < \lambda < \delta} \phi''_{j,k}(\lambda).$$

For $\lambda \in (0, \delta)$,

$$\phi_{j,k}(\lambda) = \phi_{j,k}(0) + \phi'_{j,k}(0)\lambda + \frac{1}{2}\phi''_{j,k}(\lambda_m)\lambda^2 \le (\tilde{U}^i)_{j,k}\lambda + \frac{1}{2}M_{\phi_{j,k}}\lambda^2,$$

where in the remainder term we have $0 < \lambda_m < \lambda$. Plug into $F_{j,k}(\lambda)$, we have

$$F_{j,k}(\lambda) \ge \lambda\frac{R}{d} - \frac{1}{2}M_{\phi_{j,k}}\lambda^2, \quad 0 < \lambda < \delta.$$

Observe that the RHS is a quadratic function in $\lambda$ and it crosses 0. Let us take $\lambda^*_{j,k} = \min\{\frac{\delta}{2}, \frac{R}{dM_{\phi_{j,k}}}\}$, then

$$F_{j,k}(\lambda^*_{j,k}) \ge \min\left\{F_{j,k}\left(\frac{\delta}{2}\right), F_{j,k}\left(\frac{R}{dM_{\phi_{j,k}}}\right)\right\} > 0.$$

It follows that $H^*(\frac{R}{d} + (\tilde{U}^i)_{j,k}) > F_{j,k}(\lambda^*_{j,k})$. Denote

$$C^* = \min_{1 \le j,k \le d} F_{j,k}(\lambda^*_{j,k}).$$

Hence we have that

$$\mathbb{P}\left(\left|\frac{1}{t}\int_0^t (\tilde{X}^i_s)_j(\tilde{X}^i_s)_k ds - (\tilde{U}^i)_{j,k}\right| \ge \frac{R}{d}\right) \le 2\exp(-C^*t), \qquad \forall 1 \le j,k \le d.$$

Since

$$\|I^i_t - \tilde{U}^i\| = \left(\sum_{1 \le j \le d}\sum_{1 \le k \le d}\left(\frac{1}{t}\int_0^t (\tilde{X}^i_s)_j(\tilde{X}^i_s)_k ds - (\tilde{U}^i)_{j,k}\right)^2\right)^{1/2}$$

Then

$$\mathbb{P}((B^i_t)^C) = \mathbb{P}(\|I^i_t - \tilde{U}^i\| \ge R)$$
$$\le \sum_{j,k}\mathbb{P}\left(\left|\frac{1}{t}\int_0^t (\tilde{X}^i_s)_j(\tilde{X}^i_s)_k ds - (\tilde{U}^i)_{j,k}\right| \ge \frac{R}{d}\right)$$
$$\le 2d^2\exp(-C^*t).$$

Let $S := t^i_{k(t)}$,

$$
\begin{aligned}
\int_0^T \mathbb{E}^i\big[\mathrm{tr}(\tilde{\Sigma}^i_{t^i_{k(t)}})\big]dt &= \int_0^T \mathbb{E}^i\big[\mathbb{E}^i\big[\mathrm{tr}(\tilde{\Sigma}^i_S) \mid S\big]\big]dt \\
&\leq \int_0^T \mathbb{E}^i\Big[\frac{2d}{\lambda_{\min}(\tilde{U}^i)} \cdot \frac{1}{S} \ + \ \delta^i_S\,\mathrm{tr}(\Sigma^i_0)\Big]dt \\
&\leq \int_0^2 \mathrm{tr}(\Sigma^i_0)dt + \int_2^T \Big[\frac{2d}{\lambda_{\min}(\tilde{U}^i)}\mathbb{E}^i\Big[\frac{1}{t^i_{k(t)}}\Big] \ + \ \mathbb{E}^i[\delta^i_{t^i_{k(t)}}]\,\mathrm{tr}(\Sigma^i_0)\Big]dt.
\end{aligned}
$$

For any $1 \leq k \leq K^i_T$, by (3.2) and (3.3),

$$
t^i_{k+1} - t^i_k \leq T^i_k \leq T^i_{k-1} + 1 \leq T^i_{k-1} + T^i_0 \leq t^i_k.
$$

Then for any $t \in [t^i_k, t^i_{k+1})$,

$$
\frac{2}{t} \geq \frac{2}{t^i_{k+1}} \geq \frac{1}{t^i_k}.
$$

Then

$$
\mathbb{E}^i\Big[\frac{1}{t^i_{k(t)}}\Big] = \mathbb{E}^i\Big[\sum_{k=1}^\infty \frac{1}{t^i_k}\mathbb{1}_{\{t^i_k \leq t < t^i_{k+1}\}}\Big] \leq \sum_{k=1}^\infty \mathbb{E}^i\Big[\frac{2}{t}\mathbb{1}_{\{t^i_k \leq t < t^i_{k+1}\}}\Big] = \frac{2}{t}.
$$

Thus

$$
\int_2^T \frac{2d}{\lambda_{\min}(\tilde{U}^i)}\mathbb{E}^i\Big[\frac{1}{t^i_{k(t)}}\Big] \leq \frac{2d}{\lambda_{\min}(\tilde{U}^i)}\int_2^T \frac{2}{t}dt \leq \frac{4d}{\lambda_{\min}(\tilde{U}^i)}\log(T).
$$

As $t^i_k \geq \frac{t}{2}$ for any $t \in [t^i_k, t^i_{k+1})$,

$$
\begin{aligned}
\int_2^T \mathbb{E}^i[\delta^i_{t^i_{k(t)}}]dt &\leq 2d^2\int_2^T \mathbb{E}^i[\exp(-C^* \cdot t^i_{k(t)})]dt \\
&\leq 2d^2\int_2^T \mathbb{E}^i\Big[\exp\Big(-C^* \cdot \frac{t}{2}\Big)\Big]dt \\
&\leq \frac{4d^2}{C^*}e^{-C^*}.
\end{aligned}
$$

Thus

$$
\begin{aligned}
\int_0^T \mathbb{E}^i\big[\mathrm{tr}(\tilde{\Sigma}^i_{t_{k(t)}})\big]dt &\leq 2\mathrm{tr}(\Sigma^i_0) + \frac{4d}{\lambda_{\min}(\tilde{U}^i)}\log(T) + \frac{4d^2}{C^*}e^{-C^*}\mathrm{tr}(\Sigma^i_0) \\
&= \Big(\frac{4d^2}{C^*}e^{-C^*} + 2\Big)\mathrm{tr}(\Sigma^i_0) + \frac{4d\|\sigma^i\|^2}{\lambda_{\min}(\tilde{U}^i)} \cdot \log(T).
\end{aligned}
$$

Finally,

$$
\int_0^T \mathbb{E}^i\Big[\|A - \hat{A}^i_{k(t)}\|^2\Big]dt \leq O(d\|\sigma^i\|^2\log(T)).
$$

$\square$

### A.6 PROOF OF PROPOSITION 3.5

Recall that we couple the processes $(\hat{X}^i_t)_{t \geq 0}$ and $(X^i_t)_{t \geq 0}$ by using the same Brownian motion $W^i$ and setting $\hat{X}^i_0 = X^i_0$. The process $X^i$ represents the trajectory of player $i$ assuming they randomize the parameter $A$ at time zero and observe its outcome, while $\hat{X}^i$ represents the process under the TS algorithm, where $A$ is unknown and sampled in episodes. The dynamics of $X^i$ are given by

$$
dX^i_t = (-\varsigma^i\Upsilon^i X^i_t + \varsigma^i\Upsilon^i\eta^i)dt + \sigma^i dW^i_t.
$$

The process $\hat{X}^i$ evolves according to

$$d\hat{X}_t^i = ((A - \varsigma^i \Upsilon_{k(t)}^i - \hat{A}_{k(t)}^i)\hat{X}_t^i + \varsigma^i \Upsilon_{k(t)}^i \eta_{k(t)}^i)dt + \sigma^i dW_t^i,$$

where $k(t)$ is the episode label $k$ such that $t \in [t_k^i, t_{k+1}^i)$.

Take derivative of $\|X_t^i - \hat{X}_t^i\|^2$ w.r.t. $t$, we have

$$\frac{d}{dt}\|X_t^i - \hat{X}_t^i\|^2 = \frac{d}{dt}(X_t^i - \hat{X}_t^i)^T(X_t^i - \hat{X}_t^i)$$

$$= 2(X_t^i - \hat{X}_t^i)^T \cdot \frac{d}{dt}(X_t^i - \hat{X}_t^i)$$

$$= 2(X_t^i - \hat{X}_t^i)^T(-\varsigma^i \Upsilon^i X_t^i + \varsigma^i \Upsilon^i \eta^i - (A - \varsigma^i \Upsilon_{k(t)}^i - \hat{A}_{k(t)}^i)\hat{X}_t^i - \varsigma^i \Upsilon_{k(t)}^i \eta_{k(t)}^i)$$

$$= 2(X_t^i - \hat{X}_t^i)^T \big( -\varsigma^i \Upsilon^i(X_t^i - \hat{X}_t^i) - (A - \hat{A}_{k(t)}^i + \varsigma^i(\Upsilon^i - \Upsilon_{k(t)}^i))\hat{X}_t^i \big)$$

$$\quad + 2(X_t^i - \hat{X}_t^i)^T \big( \varsigma^i(\Upsilon^i \eta^i - \Upsilon_{k(t)}^i \eta_{k(t)}^i) \big)$$

Take integral on both sides, by the coupling, $\|X_0^i - \hat{X}_0^i\|^2 = 0$,

$$\|X_T^i - \hat{X}_T^i\|^2 - 0$$

$$= -2\int_0^T (X_t^i - \hat{X}_t^i)^T \varsigma^i \Upsilon^i(X_t^i - \hat{X}_t^i)dt - 2\int_0^T (X_t^i - \hat{X}_t^i)^T(A - \hat{A}_{k(t)}^i)\hat{X}_t^i dt$$

$$\quad - 2\int_0^T (X_t^i - \hat{X}_t^i)^T \varsigma^i(\Upsilon^i - \Upsilon_{k(t)}^i)\hat{X}_t^i dt + 2\int_0^T (X_t^i - \hat{X}_t^i)^T \varsigma^i(\Upsilon^i \eta^i - \Upsilon_{k(t)}^i \eta_{k(t)}^i)dt.$$

Then,

$$\int_0^T (X_t^i - \hat{X}_t^i)^T \varsigma^i \Upsilon^i(X_t^i - \hat{X}_t^i)dt$$

$$= -\frac{1}{2}\|X_T^i - \hat{X}_T^i\|^2 - \int_0^T (X_t^i - \hat{X}_t^i)^T(A - \hat{A}_{k(t)}^i)\hat{X}_t^i dt$$

$$\quad - \int_0^T (X_t^i - \hat{X}_t^i)^T \varsigma^i(\Upsilon^i - \Upsilon_{k(t)}^i)\hat{X}_t^i dt + \int_0^T (X_t^i - \hat{X}_t^i)^T \varsigma^i(\Upsilon^i \eta^i - \Upsilon_{k(t)}^i \eta_{k(t)}^i))dt.$$

By assumption, $\frac{1}{2}\big(\varsigma^i \Upsilon^i + (\varsigma^i \Upsilon^i)^T\big)$ is positive definite. Hence, there exists a constant $c^i$ s.t. $\frac{1}{2}\big(\varsigma^i \Upsilon^i + (\varsigma^i \Upsilon^i)^T\big) \succeq c^i I_d$. Then

$$c^i \int_0^T \|X_t^i - \hat{X}_t^i\|^2 dt$$

$$\leq \int_0^T (X_t^i - \hat{X}_t^i)^T \frac{1}{2}\big(\varsigma^i \Upsilon^i + (\varsigma^i \Upsilon^i)^T\big)(X_t^i - \hat{X}_t^i)dt$$

$$= \frac{1}{2}\int_0^T (X_t^i - \hat{X}_t^i)^T \varsigma^i \Upsilon^i(X_t^i - \hat{X}_t^i)dt + \frac{1}{2}\int_0^T (X_t^i - \hat{X}_t^i)^T(\varsigma^i \Upsilon^i)^T(X_t^i - \hat{X}_t^i)dt$$

$$= \frac{1}{2}\int_0^T (X_t^i - \hat{X}_t^i)^T \varsigma^i \Upsilon^i(X_t^i - \hat{X}_t^i)dt + \frac{1}{2}\int_0^T \big((X_t^i - \hat{X}_t^i)^T \varsigma^i \Upsilon^i(X_t^i - \hat{X}_t^i)\big)^T dt$$

$$= \int_0^T (X_t^i - \hat{X}_t^i)^T \varsigma^i \Upsilon^i(X_t^i - \hat{X}_t^i)dt.$$

The last equality follows that since $(X_t^i - \hat{X}_t^i)^T \varsigma^i \Upsilon^i(X_t^i - \hat{X}_t^i)$ is scalar, it equals its transpose.

Since $-\frac{1}{2}\|X_T^i - \hat{X}_T^i\|^2 \leq 0$,

$$c^i \int_0^T \|X_t^i - \hat{X}_t^i\|^2 dt$$

$$\leq - \int_0^T (X_t^i - \hat{X}_t^i)^T (A - \hat{A}_{k(t)}^i) \hat{X}_t^i dt$$

$$- \int_0^T (X_t^i - \hat{X}_t^i)^T \varsigma^i (\Upsilon^i - \Upsilon_{k(t)}^i) \hat{X}_t^i dt$$

$$+ \int_0^T (X_t^i - \hat{X}_t^i)^T \varsigma^i (\Upsilon^i \eta^i - \Upsilon_{k(t)}^i \eta_{k(t)}^i)) dt.$$

Taking expectations on both sides, we get the following bound

$$\mathbb{E}^i \Big[ \int_0^T \|X_t^i - \hat{X}_t^i\|^2 dt \Big] \leq \frac{1}{c^i} \mathbb{E}^i \Big[ \Big| \int_0^T (X_t^i - \hat{X}_t^i)^T (A - \hat{A}_{k(t)}^i) \hat{X}_t^i dt \Big| \Big]$$

$$+ \frac{1}{c^i} \mathbb{E}^i \Big[ \Big| \int_0^T (X_t^i - \hat{X}_t^i)^T \varsigma^i (\Upsilon^i - \Upsilon_{k(t)}^i) \hat{X}_t^i dt \Big| \Big]$$

$$+ \frac{1}{c^i} \mathbb{E}^i \Big[ \Big| \int_0^T (X_t^i - \hat{X}_t^i)^T \varsigma^i (\Upsilon^i \eta^i - \Upsilon_{k(t)}^i \eta_{k(t)}^i)) dt \Big| \Big]$$

$$:= \frac{1}{c^i} [E_1 + E_2 + E_3].$$

Consider the first term and second term. For $E_1$

$$E_1 = \mathbb{E}^i \Big[ \Big| \int_0^T (X_t^i - \hat{X}_t^i)^T (A - \hat{A}_{k(t)}^i) \hat{X}_t^i dt \Big| \Big] \leq \mathbb{E}^i \Big[ \int_0^T \|(A - \hat{A}_{k(t)}^i)\| \cdot (\|X_t^i\| + \|\hat{X}_t^i\|) \cdot \|\hat{X}_t^i\| dt \Big]$$

For the second term $E_2$, from Lemma A.3,

$$\|\Upsilon^i - \Upsilon_{k(t)}^i\| \leq L_\Upsilon^i \|A - \hat{A}_{k(t)}^i\|, \ \|\eta^i - \eta_{k(t)}^i\| \leq L_\eta^i \|A - \hat{A}_{k(t)}^i\|.$$

Then

$$E_2 = \mathbb{E}^i \Big[ \Big| \int_0^T (X_t^i - \hat{X}_t^i)^T \varsigma^i (\Upsilon^i - \Upsilon_{k(t)}^i) \hat{X}_t^i dt \Big| \Big]$$

$$\leq L_\Upsilon^i \mathbb{E}^i \Big[ \int_0^T (\|X_t^i\| + \|\hat{X}_t^i\|) \cdot \|A - \hat{A}_{k(t)}^i\| \cdot \|\hat{X}_t^i\| dt \Big].$$

So for $E_1$ and $E_2$, we only need to bound $\mathbb{E}^i \Big[ \int_0^T (\|X_t^i\| + \|\hat{X}_t^i\|) \cdot \|A - \hat{A}_{k(t)}^i\| \cdot \|\hat{X}_t^i\| dt \Big]$. By Cauchy–Schwarz inequality, we first split the integrand into two factors:

$$\mathbb{E}^i \Bigg[ \int_0^T \|A - \hat{A}_{k(t)}^i\| \cdot (\|X_t^i\| + \|\hat{X}_t^i\|) \|\hat{X}_t^i\| dt \Bigg]$$

$$\leq \left( \int_0^T \mathbb{E}^i \Big[ \|A - \hat{A}_{k(t)}^i\|^2 \Big] dt \right)^{1/2} \left( \int_0^T \mathbb{E}^i \Big[ (\|X_t^i\| + \|\hat{X}_t^i\|)^2 \|\hat{X}_t^i\|^2 \Big] dt \right)^{1/2}$$

From Proposition 3.4,

$$\left( \int_0^T \mathbb{E}^i \Big[ \|A - \hat{A}_{k(t)}^i\|^2 \Big] dt \right)^{1/2} \leq O(\|\sigma^i\| \sqrt{d \log(T)}).$$

For the second factor, we apply Young's inequality,

$$\mathbb{E}^i \Big[ \|X_t^i\|^2 \cdot \|\hat{X}_t^i\|^2 \Big] \leq \frac{1}{2} \mathbb{E}^i [\|X_t^i\|^4] + \frac{1}{2} \mathbb{E}^i [\|\hat{X}_t^i\|^4],$$

then

$$\left(\int_0^T \mathbb{E}^i\Big[(\|X_t^i\| + \|\hat{X}_t^i\|)^2\|\hat{X}_t^i\|^2\Big]dt\right)^{1/2} \le \left(\int_0^T \frac{1}{2}\mathbb{E}^i\big[(\|X^i\|_T^*)^4\big] + \frac{3}{2}\mathbb{E}^i\Big[(\|\hat{X}^i\|_T^*)^4\Big]dt\right)^{1/2}$$

$$\le O(\|\sigma^i\|^2\sqrt{T}).$$

Combining both bounds yields

$$E_1 + E_2 \le O(\|\sigma^i\|^3\sqrt{dT\log(T)}).$$

Consider the third term,

$$E_3 = \mathbb{E}^i\Big[\big|\int_0^T (X_t^i - \hat{X}_t^i)^T\varsigma^i(\Upsilon^i\eta^i - \Upsilon_{k(t)}^i\eta_{k(t)}^i)dt\big|\Big]$$

$$= \mathbb{E}^i\Big[\big|\int_0^T (X_t^i - \hat{X}_t^i)^T\varsigma^i\big(\Upsilon^i(\eta^i - \eta_{k(t)}^i) + (\Upsilon^i - \Upsilon_{k(t)}^i)\eta_{k(t)}^i\big)dt\big|\Big]$$

$$\le (L_\Upsilon^i M_\eta + L_\eta^i M_\Upsilon)\|\varsigma^i\|\mathbb{E}^i\Big[\big|\int_0^T (X_t^i - \hat{X}_t^i)^T\|A - \hat{A}_{k(t)}^i\|dt\big|\Big]$$

$$\le (L_\Upsilon^i M_\eta + L_\eta^i M_\Upsilon)\|\varsigma^i\|\mathbb{E}^i\Big[\int_0^T \|A - \hat{A}_{k(t)}^i\|(\|\hat{X}_t^i\| + \|X_t^i\|)dt\Big].$$

By Cauchy–Schwarz inequality,

$$\mathbb{E}^i\Big[\int_0^T \|A - \hat{A}_{k(t)}^i\|(\|\hat{X}_t^i\| + \|X_t^i\|)dt\Big]$$

$$\le \left(\int_0^T \mathbb{E}^i\Big[\|A - \hat{A}_{k(t)}^i\|^2\Big]dt\right)^{1/2}\left(\int_0^T \mathbb{E}^i\Big[(\|\hat{X}_t^i\| + \|X_t^i\|)^2\Big]dt\right)^{1/2},$$

where

$$\left(\int_0^T \mathbb{E}^i\Big[(\|\hat{X}_t^i\| + \|X_t^i\|)^2\Big]dt\right)^{1/2}$$

$$\le \left(\int_0^T 2\mathbb{E}^i\big[(\|X^i\|_T^*)^2\big] + 2\mathbb{E}^i\Big[(\|\hat{X}^i\|_T^*)^2\Big]dt\right)^{1/2} \le O(\|\sigma^i\|\sqrt{T}),$$

and

$$\left(\int_0^T \mathbb{E}^i\Big[\|A - \hat{A}_{k(t)}^i\|^2\Big]dt\right)^{1/2} \le O(\|\sigma^i\|\sqrt{d\log(T)}).$$

Then

$$\mathbb{E}^i\Big[\int_0^T \|A - \hat{A}_{k(t)}^i\|(\|\hat{X}_t^i\| + \|X_t^i\|)dt\Big] \le O(\|\sigma^i\|^2\sqrt{dT\log(T)}). \tag{A.22}$$

Thus $E_3 \le O(\|\sigma^i\|^2\sqrt{dT\log(T)})$.

Combine $E_1, E_2, E_3$ together, we have that

$$\mathbb{E}^i\Big[\int_0^T \|X_t^i - \hat{X}_t^i\|^2dt\Big] \le O\Big(\|\sigma^i\|^3\sqrt{dT\log(T)}\Big).$$

Then

$$\lim_{T\to\infty}\frac{1}{T}\mathbb{E}^i\Big[\int_0^T \|\hat{X}_t^i - X_t^i\|^2dt\Big] = 0.$$

$$\square$$

## A.7 PROOF OF PROPOSITION 3.6

Recall $\bar{a}^i(x; A)$ from (2.11). Plug in $\hat{\alpha}_t^i = \bar{a}^i(\hat{X}_t^i; \hat{A}_{k(t)}^i)$ and $(\alpha_t^i)^* = \bar{a}^i(X_t^i; A)$, we have

$$
\mathbb{E}^i \Big[ \int_0^T \|\hat{\alpha}_t^i - (\alpha_t^i)^*\|^2 \Big]
$$

$$
= \mathbb{E}^i \Big[ \sum_{k=0}^{K_T^i} \int_{t_k^i}^{t_{k+1}^i \wedge T} \Big[ (\varsigma^i \Upsilon^i + \hat{A}_k^i) \hat{X}_t^i - \varsigma^i \Upsilon^i \eta^i \Big]^T R^i \Big[ (\varsigma^i \Upsilon^i + \hat{A}_k^i) \hat{X}_t^i - \varsigma^i \Upsilon^i \eta^i \Big] dt \Big]
$$

$$
- \mathbb{E}^i \Big[ \int_0^T \Big[ (\varsigma^i \Upsilon^i + A) X_t^i - \varsigma^i \Upsilon^i \eta^i \Big]^T R^i \Big[ (\varsigma^i \Upsilon^i + A) X_t^i - \varsigma^i \Upsilon^i \eta^i \Big] dt \Big]
$$

$$
= \mathbb{E}^i \Big[ \sum_{k=0}^{K_T^i} \int_{t_k^i}^{t_{k+1}^i \wedge T} \Big[ (\varsigma^i \Upsilon^i + \hat{A}_k^i) \hat{X}_t^i + (\varsigma^i \Upsilon^i + A) X_t^i - 2\varsigma^i \Upsilon^i \eta^i \Big]^T \cdot
$$

$$
\cdot R^i \Big[ (\varsigma^i \Upsilon^i + \hat{A}_k^i) \hat{X}_t^i - (\varsigma^i \Upsilon^i + A) X_t^i \Big] dt \Big].
$$

From Assumption 3.1,

$$
\big\| (\varsigma^i \Upsilon^i + \hat{A}_k^i) \hat{X}_t^i + (\varsigma^i \Upsilon^i + A) X_t^i - 2\varsigma^i \Upsilon^i \eta^i \big\|
$$

$$
\leq \big( \frac{1}{2} \|\sigma^i\|^2 M_\Upsilon + M_A \big) \big( \|\hat{X}_t^i\| + \|X_t^i\| \big) + \|\sigma^i\|^2 M_\Upsilon M_\eta
$$

$$
\leq \|\sigma^i\|^2 C_1 \big( \|\hat{X}^i\|_T^* + \|X^i\|_T^* \big) + \|\sigma^i\|^2 C_2,
$$

where $C_1, C_2$ are constants. Then

$$
\mathbb{E}^i \Big[ \int_0^T \|\hat{\alpha}_t^i - (\alpha_t^i)^*\|^2 \Big]
$$

$$
\leq \mathbb{E}^i \Big[ \|\sigma^i\|^2 \big( C_1 (\|\hat{X}^i\|_T^* + \|X^i\|_T^*) + C_2 \big) \sum_{k=0}^{K_T^i} \int_{t_k^i}^{t_{k+1}^i \wedge T} \big\| (\varsigma^i \Upsilon^i + \hat{A}_k^i) \hat{X}_t^i - (\varsigma^i \Upsilon^i + A) X_t^i \big\| dt \Big]
$$

$$
\leq \mathbb{E}^i \Big[ \|\sigma^i\|^2 \big( C_1 (\|\hat{X}^i\|_T^* + \|X^i\|_T^*) + C_2 \big) \int_0^T \big( \frac{1}{2} \|\sigma^i\|^2 M_\Upsilon + M_A \big) \|\hat{X}_t^i - X_t^i\| dt \Big]
$$

$$
+ \mathbb{E}^i \Big[ \|\sigma^i\|^2 \big( C_1 (\|\hat{X}^i\|_T^* + \|X^i\|_T^*) + C_2 \big) \sum_{k=0}^{K_T^i} \int_{t_k^i}^{t_{k+1}^i \wedge T} \|(\hat{A}_k^i - A) \hat{X}_t^i\| dt \Big],
$$

$$
\text{(A.23)}
$$

where the last inequality follows form $\big\| (\varsigma^i \Upsilon^i + \hat{A}_k^i) \hat{X}_t^i - (\varsigma^i \Upsilon^i + A) X_t^i \big\| \leq \big\| (\varsigma^i \Upsilon^i + A)(\hat{X}_t^i - X_t^i) \big\| + \big\| (\hat{A}_k^i - A) \hat{X}_t^i \big\|$.

Consider the first term, by Cauchy-Schwarz inequality, we have

$$
\mathbb{E}^i \Big[ \|\sigma^i\|^2 \big( C_1 (\|\hat{X}^i\|_T^* + \|X^i\|_T^*) + C_2 \big) \int_0^T \|\sigma^i\|^2 C_1 \|\hat{X}_t^i - X_t^i\| dt \Big]
$$

$$
\leq \Big( \mathbb{E}^i \Big[ T \|\sigma^i\|^4 \big( C_1 (\|\hat{X}^i\|_T^* + \|X^i\|_T^*) + C_2 \big)^2 \Big] \Big)^{1/2} \Big( \mathbb{E}^i \Big[ \|\sigma^i\|^4 (C_1)^2 \int_0^T \|\hat{X}_t^i - X_t^i\|^2 \Big] \Big)^{1/2}.
$$

Bound the left term from Lemma A.1 and the right term from Proposition 3.5, we have

$$
\mathbb{E}^i \Big[ \|\sigma^i\|^2 \big( C_1 (\|\hat{X}^i\|_T^* + \|X^i\|_T^*) + C_2 \big) \int_0^T \|\sigma^i\|^2 C_1 \|\hat{X}_t^i - X_t^i\| dt \Big] \leq O\big( \|\sigma^i\|^{\frac{13}{2}} \sqrt{T \sqrt{dT \log T}} \big).
$$

For the second term in (A.23), by Cauchy Schwarz inequality,

$$\mathbb{E}^i\Big[\|\sigma^i\|^2\big(C_1(\|\hat{X}^i\|_T^* + \|X^i\|_T^*) + C_2\big)\sum_{k=0}^{K_T^i}\int_{t_k^i}^{t_{k+1}^i \wedge T}\|(\hat{A}_k^i - A)\hat{X}_t^i\|dt\Big]$$

$$\leq \mathbb{E}^i\Big[\|\sigma^i\|^2 \cdot \|\hat{X}^i\|_T^*\big(C_1(\|\hat{X}^i\|_T^* + \|X^i\|_T^*) + C_2\big)\int_0^T\|\hat{A}_{k(t)}^i - A\|dt\Big]$$

$$\leq \Big(\mathbb{E}^i\Big[T\|\sigma^i\|^4(\|\hat{X}^i\|_T^*)^2\big(C_1(\|\hat{X}^i\|_T^* + \|X^i\|_T^*) + C_2\big)^2\Big]\Big)^{1/2}\Big(\mathbb{E}^i\Big[\int_0^T\|\hat{A}_{k(t)}^i - A\|^2\Big]\Big)^{1/2}.$$

From Lemma A.1 and Proposition 3.4, we get the bound

$$\mathbb{E}^i\Big[\|\sigma^i\|^2\big(C_1(\|\hat{X}^i\|_T^* + \|X^i\|_T^*) + C_2\big)\sum_{k=0}^{K_T^i}\int_{t_k^i}^{t_{k+1}^i \wedge T}\|(\hat{A}_k^i - A)\hat{X}_t^i\|dt\Big] \leq O(\|\sigma^i\|^5\sqrt{dT\log T}).$$

Combining the bounds for the first and the second term in (A.23), we have

$$\mathbb{E}^i\Big[\int_0^T\|\hat{\alpha}_t^i - (\alpha_t^i)^*\|^2\Big] \leq O\Big(\max\big(\|\sigma^i\|^{\frac{13}{2}}d^{\frac{1}{4}}, \|\sigma^i\|^5 d^{\frac{1}{2}}\big)T^{\frac{3}{4}}(\log T)^{\frac{1}{2}}\Big).$$

$\square$

## A.8    Proof of Theorem 3.2

In order to show that the profile $(\hat{\alpha}_t^1, \ldots, \alpha_t^N)_{t\geq 0}$ is a Nash equilibrium, we need to show that for any player $i \in [N]$,

    (1) The strategy $\hat{\alpha}^i$ is admissible.

    (2) Under strategy $\hat{\alpha}^i$, the cost for player $i$ from any initial position equals the value $\lambda^i$.

    (3) Under any other admissible strategy $\alpha^i$, the cost for player $i$ is greater than $\lambda^i$.

Recall (3.6), we denote $(\alpha_t^i)^* = \bar{a}^i(X_t^i; A)$ as the equilibrium policy under known parameter $A$. Also, recall (3.4), we denote $\hat{\alpha}_t^i = \bar{a}^i(\hat{X}_t^i; \hat{A}_k^i)$ as the optimal control under state $\hat{X}_t^i$ and parameter $\hat{A}_k^i$.

We divide the proof into three steps. First, we show that each player's strategy is admissible. Second, we prove that under the TS strategy $\hat{\alpha}^i$, the cost for player $i$, starting from any initial position, equals the value $\lambda^i$. Finally, we show that under any other admissible strategy $\alpha^i$, the cost for player $i$ exceeds $\lambda^i$.

### A.8.1    Proof of (1): Admissibility of $\hat{\alpha}^i$ and ergodicity of $(\hat{X}_t^i)_{t\geq 0}$ with $m^i$.

First, we prove that the strategy for each player is admissible. Recall the definition of admissibility from Definition 3.1, which relies on Definition 2.1. Recall that $\hat{\alpha}^i$ is defined recursively by:

$$\hat{\alpha}_t^i = \varsigma^i\Upsilon^i\hat{X}_t^i + \hat{A}_k^i\hat{X}_t^i - \varsigma^i\Upsilon^i\eta^i, \qquad \forall t \in [t_k^i, t_{k+1}^i).$$

The, it is obvious that $(\hat{\alpha}_t^i)$ is adapted to $\mathcal{F}_t^i$. Also, by Assumption 3.1,

$$\mathbb{E}^i[\|\hat{\alpha}_t^i\|^2] \leq 2\mathbb{E}^i\Big[\|\varsigma^i\Upsilon^i + \hat{A}_k^i\|^2 \cdot \|\hat{X}_t^i\|^2\Big] + 2\|\varsigma^i\Upsilon^i\eta^i\|^2$$

$$\leq 4(\|\varsigma^i\Upsilon^i\|^2 + M_A^2)\mathbb{E}^i[(\|\hat{X}^i\|_t^*)^2] + 2\|\varsigma^i\Upsilon^i\eta^i\|^2 < \infty,$$

where $\mathbb{E}^i[(\|\hat{X}^i\|_t^*)^2] < \infty$ follows from Lemma A.1, which also establishes the first bullet point in Definition 2.1.

Hence, it remains to prove that $\hat{X}_t^i$ is ergodic (with the same $m^i$ as $X_t$) in the following sense:

$$\lim_{T\to\infty}\frac{1}{T}\mathbb{E}^i\Big[\int_0^T h(\hat{X}_t^i)dt\Big] = \int h(x)dm^i(x)$$

locally uniformly with respect to the initial state $X_0^i$ for all functions $h$ which are polynomials of degree at most 2. To this end, consider a polynomial function $h$ of degree at most 2. More explicitly, $h(x) = x^T C_2 x + c_1^T x$, with $C_2 \in \mathbb{R}^{d \times d}$ and $c_1 \in \mathbb{R}^d$. Then,

$$\lim_{T \to \infty} \frac{1}{T} \mathbb{E}^i \Big[ \int_0^T h(\hat{X}_t^i) dt \Big]$$

$$= \lim_{T \to \infty} \frac{1}{T} \mathbb{E}^i \Big[ \int_0^T h(X_t^i) dt \Big]$$

$$+ \lim_{T \to \infty} \frac{1}{T} \mathbb{E}^i \Big[ \int_0^T c_1^T (\hat{X}_t^i - X_t^i) + (\hat{X}_t^i)^T C_2 \hat{X}_t^i - (X_t^i)^T C_2 X_t^i dt \Big]$$

$$= \lim_{T \to \infty} \frac{1}{T} \mathbb{E}^i \Big[ \int_0^T h(X_t^i) dt \Big] + \lim_{T \to \infty} \frac{1}{T} \mathbb{E}^i \Big[ \int_0^T (c_1^T + (\hat{X}_t^i + X_t^i)^T C_2)(\hat{X}_t^i - X_t^i) dt \Big].$$

We know from Proposition 2.1 that the strategies under Nash equilibrium are admissible, hence, the respective dynamics $X_t^i$ are ergodic with $m^i$ from the HJB-FKP equation:

$$\lim_{T \to \infty} \frac{1}{T} \mathbb{E}^i \Big[ \int_0^T h(X_t^i) dt \Big] = \int h(x) dm^i(x).$$

Hence, it remains to show that

$$\lim_{T \to \infty} \frac{1}{T} \mathbb{E}^i \Big[ \int_0^T (c_1^T + (\hat{X}_t^i + X_t^i)^T C_2)(\hat{X}_t^i - X_t^i) dt \Big] = 0. \tag{A.24}$$

By Cauchy–Schwartz inequality, We have

$$\lim_{T \to \infty} \frac{1}{T} \mathbb{E}^i \Big[ \int_0^T (c_1^T + 2(X_t^i)^T C_2)(\hat{X}_t^i - X_t^i) dt \Big]$$

$$\leq \lim_{T \to \infty} \frac{1}{T} \int_0^T \mathbb{E}^i \Big[ \| (c_1^T + 2(X_t^i)^T C_2)(\hat{X}_t^i - X_t^i) \| \Big] dt$$

$$\leq \lim_{T \to \infty} \sqrt{\mathbb{E}^i [(\|c_1\| + 2\|C_2\| \cdot \|X^i\|_T^*)^2]} \cdot \sqrt{\frac{1}{T} \int_0^T \mathbb{E}^i [\|\hat{X}_t^i - X_t^i\|^2] dt}.$$

Recall from Lemma A.1 that $\mathbb{E}^i \big[ (\|X^i\|_T^*)^p \big] \leq (\|\sigma^i\|)^p O(1)$. Together with Proposition 3.5, we obtain (A.24). $\qquad \square$

### A.8.2 PROOF OF (2): UNDER $\hat{\alpha}^i$, THE COST FOR PLAYER $i$ EQUALS $\lambda^i$.

In this part, we prove that under the TS algorithm strategy $(\hat{\boldsymbol{\alpha}}_t)_{t \geq 0}$, the cost for each player $i$ is equal to $\lambda^i$. For this, we add and subtract terms to the ergodic cost for Player $i$ as follows

$$J^i(\boldsymbol{X}_0, \hat{\boldsymbol{\alpha}}) = \liminf_{T \to +\infty} \frac{1}{T} \mathbb{E}^i \Big[ \int_0^T F^i(\hat{\boldsymbol{X}}_t, \hat{\alpha}_t^i) \Big]$$

$$= \liminf_{T \to +\infty} \frac{1}{T} \mathbb{E}^i \Big[ \int_0^T \tilde{F}^i(\hat{\boldsymbol{X}}_t) dt \Big] + \liminf_{T \to +\infty} \frac{1}{T} \mathbb{E}^i \Big[ \frac{1}{2} \int_0^T (\hat{\alpha}_t^i)^T R^i \hat{\alpha}_t^i dt \Big]$$

$$= \Big( \liminf_{T \to +\infty} \frac{1}{T} \mathbb{E}^i \Big[ \int_0^T \tilde{F}^i(\hat{\boldsymbol{X}}_t) dt \Big] - \liminf_{T \to +\infty} \frac{1}{T} \mathbb{E}^i \Big[ \int_0^T \tilde{F}^i(\boldsymbol{X}_t) dt \Big] \Big)$$

$$+ \Big( \liminf_{T \to +\infty} \frac{1}{T} \mathbb{E}^i \Big[ \frac{1}{2} \int_0^T (\hat{\alpha}_t^i)^T R^i \hat{\alpha}_t^i dt \Big] - \liminf_{T \to +\infty} \frac{1}{T} \mathbb{E}^i \Big[ \frac{1}{2} \int_0^T ((\alpha_t^i)^*)^T R^i (\alpha_t^i)^* dt \Big] \Big)$$

$$+ \Big( \liminf_{T \to +\infty} \frac{1}{T} \mathbb{E}^i \Big[ \int_0^T \tilde{F}^i(\boldsymbol{X}_t) dt \Big] + \liminf_{T \to +\infty} \frac{1}{T} \mathbb{E}^i \Big[ \frac{1}{2} \int_0^T ((\alpha_t^i)^*)^T R^i (\alpha_t^i)^* dt \Big] \Big)$$

$$=: I_1 + I_2 + I_3.$$

Next, we will estimate $I_1$, $I_2$, and $I_3$ accordingly.

**Proving $I_1 = 0$:** For $I_1$, plug in $\tilde{F}^i(\boldsymbol{x}) := (\boldsymbol{x} - \overline{\boldsymbol{x}}_i)^T \mathcal{Q}^i (\boldsymbol{x} - \overline{\boldsymbol{x}}_i)$,

$$
|I_1| = \left| \liminf_{T \to +\infty} \frac{1}{T} \mathbb{E}^i \Big[ \int_0^T \tilde{F}^i(\hat{\boldsymbol{X}}_t) dt \Big] - \liminf_{T \to +\infty} \frac{1}{T} \mathbb{E}^i \Big[ \int_0^T \tilde{F}^i(\boldsymbol{X}_t) dt \Big] \right|
$$

$$
\leq \limsup_{T \to +\infty} \frac{1}{T} \left| \mathbb{E}^i \Big[ \int_0^T (\hat{\boldsymbol{X}}_t - \bar{\boldsymbol{x}}_i)^T \mathcal{Q}^i (\hat{\boldsymbol{X}}_t - \bar{\boldsymbol{x}}_i) - (\boldsymbol{X}_t - \bar{\boldsymbol{x}}_i)^T \mathcal{Q}^i (\boldsymbol{X}_t - \bar{\boldsymbol{x}}_i) dt \Big] \right|
$$

$$
\leq \limsup_{T \to +\infty} \left| \frac{1}{T} \int_0^T \mathbb{E}^i \Big[ (\hat{\boldsymbol{X}}_t - \boldsymbol{X}_t)^T \mathcal{Q}^i (\hat{\boldsymbol{X}}_t + \boldsymbol{X}_t - 2\bar{\boldsymbol{x}}_i) \Big] dt \right|.
$$

$$
\leq \sqrt{\lambda_{\max}((\mathcal{Q}^i)^2)} \limsup_{T \to +\infty} \frac{1}{T} \int_0^T \mathbb{E}^i \Big[ \|\hat{\boldsymbol{X}}_t - \boldsymbol{X}_t\| \cdot \|\hat{\boldsymbol{X}}_t + \boldsymbol{X}_t - 2\bar{\boldsymbol{x}}_i\| \Big] dt.
$$

where $\lambda_{\max}((\mathcal{Q}^i)^2)$ is the biggest eigenvalue of $(\mathcal{Q}^i)^T \mathcal{Q}^i = (\mathcal{Q}^i)^2$, since $\mathcal{Q}^i$ is symmetric.
Apply Cauchy–Schwarz inequality,

$$
\limsup_{T \to +\infty} \frac{1}{T} \int_0^T \mathbb{E}^i \Big[ \|\hat{\boldsymbol{X}}_t - \boldsymbol{X}_t\| \cdot \|\hat{\boldsymbol{X}}_t + \boldsymbol{X}_t - 2\bar{\boldsymbol{x}}_i\| \Big] dt
$$

$$
\leq \sqrt{\limsup_{T \to +\infty} \frac{1}{T} \int_0^T \mathbb{E}^i \Big[ \|\hat{\boldsymbol{X}}_t - \boldsymbol{X}_t\|^2 \Big] dt} \cdot \sqrt{\limsup_{T \to +\infty} \frac{1}{T} \int_0^T \mathbb{E}^i \Big[ \|\hat{\boldsymbol{X}}_t + \boldsymbol{X}_t - 2\bar{\boldsymbol{x}}_i\|^2 \Big] dt}
$$

$$
\leq \sqrt{\limsup_{T \to +\infty} \frac{1}{T} \int_0^T \mathbb{E}^i \Big[ \|\hat{\boldsymbol{X}}_t - \boldsymbol{X}_t\|^2 \Big] dt}
$$

$$
\times \sqrt{3 \sup_{T \in [0,\infty)} \mathbb{E}^i \Big[ (\|\hat{\boldsymbol{X}}\|_T^*)^2 \Big] + \sup_{T \in [0,\infty)} \mathbb{E}^i \Big[ (\|\boldsymbol{X}\|_T^*)^2 \Big] + 4\|\bar{\boldsymbol{x}}_i\|^2}.
$$

Then,

$$
|I_1| \leq \sqrt{\lambda_{\max}((\mathcal{Q}^i)^2)} \sqrt{3} \sqrt{\limsup_{T \to +\infty} \frac{1}{T} \int_0^T \mathbb{E}^i \Big[ \|\hat{\boldsymbol{X}}_t - \boldsymbol{X}_t\|^2 \Big] dt}
$$

$$
\times \sqrt{\sup_{T \in [0,\infty)} \mathbb{E}^i \Big[ (\|\hat{\boldsymbol{X}}\|_T^*)^2 \Big] + \sup_{T \in [0,\infty)} \mathbb{E}^i \Big[ (\|\boldsymbol{X}\|_T^*)^2 \Big] + 4\|\bar{\boldsymbol{x}}_i\|^2}
$$

$$
\to 0,
$$

where the limit follows since the first square root tends to zero by Proposition 3.5, and the second square root is bounded by Lemma A.1.

**Proving $I_2 = 0$:** Plug in the explicit expressions of $(\alpha_t^i)^*$ and $\hat{\alpha}_t^i$, we get

$$
I_2 = \liminf_{T \to +\infty} \frac{1}{T} \mathbb{E}^i \Big[ \frac{1}{2} \int_0^T (\hat{\alpha}_t^i)^T R^i \hat{\alpha}_t^i dt \Big] - \liminf_{T \to +\infty} \frac{1}{T} \mathbb{E}^i \Big[ \frac{1}{2} \int_0^T ((\alpha_t^i)^*)^T R^i (\alpha_t^i)^* dt \Big]
$$

$$
= \frac{1}{2} \liminf_{T \to +\infty} \frac{1}{T} \mathbb{E}^i \Big[ \sum_{k=0}^{K_T^i} \int_{t_k^i}^{t_{k+1}^i \wedge T} \Big[ (\varsigma^i \Upsilon^i + \hat{A}_k^i) \hat{X}_t^i - \varsigma^i \Upsilon^i \eta^i \Big]^T R^i \Big[ (\varsigma^i \Upsilon^i + \hat{A}_k^i) \hat{X}_t^i - \varsigma^i \Upsilon^i \eta^i \Big] dt \Big]
$$

$$
- \frac{1}{2} \liminf_{T \to +\infty} \frac{1}{T} \mathbb{E}^i \Big[ \int_0^T \Big[ (\varsigma^i \Upsilon^i + A) X_t^i - \varsigma^i \Upsilon^i \eta^i \Big]^T R^i \Big[ (\varsigma^i \Upsilon^i + A) X_t^i - \varsigma^i \Upsilon^i \eta^i \Big] dt \Big].
$$

The proof follows the same reasoning as for $I_1$. Therefore, the details are omitted.

**Proving $I_3 = \lambda^i$:** By the ergodicity of $(X_t^i)_{t\geq 0}$ for any $i \in [N]$, hich follows from Proposition 2.1, we have

$$I_3 = \liminf_{T\to+\infty} \frac{1}{T}\mathbb{E}^i\Big[\int_0^T \tilde{F}^i(\boldsymbol{X}_t)dt\Big] + \liminf_{T\to+\infty}\frac{1}{T}\mathbb{E}^i\Big[\frac{1}{2}\int_0^T ((\alpha_t^i)^*)^T R^i(\alpha_t^i)^* dt\Big]$$

$$= \int_{\mathbb{R}^{Nd}} \tilde{F}^i(\boldsymbol{x})\prod_{j=1}^N dm^j(x^j) + \liminf_{T\to+\infty}\frac{1}{T}\mathbb{E}^i\Big[\frac{1}{2}\int_0^T ((\alpha_t^i)^*)^T R^i(\alpha_t^i)^* dt\Big]$$

$$= \int_{\mathbb{R}} \tilde{f}^i(x^i;\boldsymbol{m}^{-i})dm^i(x^i) + \liminf_{T\to+\infty}\frac{1}{T}\mathbb{E}^i\Big[\frac{1}{2}\int_0^T ((\alpha_t^i)^*)^T R^i(\alpha_t^i)^* dt\Big]$$

$$= \liminf_{T\to+\infty}\frac{1}{T}\mathbb{E}^i\Big[\int_0^T \Big[\tilde{f}^i(X_t^i;\boldsymbol{m}^{-i}) + \frac{1}{2}((\alpha_t^i)^*)^T R^i(\alpha_t^i)^*\Big]dt\Big].$$

Plug in $\tilde{f}^i$ according to the HJB equation (2.5) and expand the Hamiltonian function, we have

$$I_3 = \liminf_{T\to+\infty}\frac{1}{T}\mathbb{E}^i\Big[\int_0^T \Big[\lambda^i + H^i(X_t^i, \nabla v^i(X_t^i)) - \mathrm{tr}(\varsigma^i D^2 m^i(X_t^i)) + \frac{1}{2}((\alpha_t^i)^*)^T R^i(\alpha_t^i)^* dt\Big]\Big]$$

$$= \liminf_{T\to+\infty}\frac{1}{T}\mathbb{E}^i\Big[\int_0^T \Big[\lambda^i + \frac{1}{2}(\nabla v^i(X_t^i))^T (R^i)^{-1}\nabla v^i(X_t^i) - (\nabla v^i(X_t^i))^T X_t^i$$

$$- \mathrm{tr}(\varsigma^i D^2 m^i(X_t^i)) + \frac{1}{2}(\nabla v^i)^T (R^i)^{-1}\nabla v^i(X_t^i)\Big]dt\Big]$$

$$= \liminf_{T\to+\infty}\frac{1}{T}\mathbb{E}^i\Big[\int_0^T \Big[\lambda^i + (\nabla v^i(X_t^i))^T \big((R^i)^{-1}\nabla v^i(X_t^i) - X_t^i\big) - \mathrm{tr}(\varsigma^i D^2 m^i(X_t^i))\Big]dt\Big]$$

$$= \lambda^i - \limsup_{T\to+\infty}\frac{1}{T}\mathbb{E}^i\Big[\int_0^T \Big[(\Lambda^i X_t^i + \rho^i)^T \big(AX_t^i - (\alpha_t^i)^*\big) + \mathrm{tr}(\varsigma^i D^2 m^i(X_t^i))\Big]dt\Big].$$

By Itô's lemma applied to $v^i(X_t^i)$ and the form of $v^i$ from (2.9):

$$\mathbb{E}^i[v^i(X_T^i) - v^i(X_0^i)] = \mathbb{E}^i\Big[\int_0^T \Big[(\nabla v^i(X_t^i))^T(AX_t^i - (\alpha_t^i)^*) + \mathrm{tr}(\varsigma^i D^2 v^i(X_t^i))\Big]dt\Big]$$

$$= \mathbb{E}^i\Big[\int_0^T \Big[(\Lambda^i X_t^i + \rho^i)^T \big(AX_t^i - (\alpha_t^i)^*\big) + \mathrm{tr}(\varsigma^i D^2 m^i(X_t^i))\Big]dt\Big].$$

Hence,

$$I_3 = \lambda^i - \limsup_{T\to+\infty}\frac{1}{T}\mathbb{E}^i[v^i(X_T^i) - v^i(X_0^i)].$$

Finally, from Lemma A.1, we obtain

$$\limsup_{T\to+\infty}\frac{1}{T}\mathbb{E}^i[v^i(X_T^i) - v^i(X_0^i)] = 0,$$

which proves that $I_3 = \lambda^i$.

Combining $I_1, I_2$ and $I_3$, we have that under player $i$'s optimal strategy $\hat{\alpha}^i$, its cost is $\lambda^i$. $\qquad\square$

### A.8.3 PROOF OF (3): UNDER ANY OTHER ADMISSIBLE STRATEGY $\alpha^i$, THE COST FOR PLAYER $i$ IS GREATER THAN $\lambda^i$.

Assume that Player $i$ uses an arbitrary admissible strategy $\tilde{\alpha}^i$, while the other players $j \in [N] \setminus \{j\}$, use the TS algorithm strategies $(\hat{\alpha}_t^j)_{t\geq 0}$. Denote Player $i$'s dynamics by $(\tilde{X}_t^i)_{t\geq 0}$. By Itô's lemma,

$$\mathbb{E}^i[v^i(\tilde{X}_T^i) - v^i(\tilde{X}_0^i)]$$

$$= \mathbb{E}^i\Big[\int_0^T \Big[(\nabla v^i(\tilde{X}_t^i))^T(A\tilde{X}_t^i - \tilde{\alpha}_t^i) + \mathrm{tr}(\varsigma^i D^2 v^i(\tilde{X}_t^i))\Big]dt\Big]$$

$$= \mathbb{E}^i\Big[\int_0^T \Big[(\nabla v^i(\tilde{X}_t^i))^T A\tilde{X}_t^i - ((R^i)^{-1}\nabla v^i(\tilde{X}_t^i))^T R^i\tilde{\alpha}_t^i + \mathrm{tr}(\varsigma^i D^2 v^i(\tilde{X}_t^i))\Big]dt\Big].$$

From Assumption 2.1, $R^i$ is positive definite. Then by simple algebra, for any $\tilde{\alpha}_t^i$, we have

$$\frac{1}{2}(\tilde{\alpha}_t^i - (R^i)^{-1}\nabla v^i(\tilde{X}_t^i))^T R^i(\tilde{\alpha}_t^i - (R^i)^{-1}\nabla v^i(\tilde{X}_t^i))$$
$$= \frac{1}{2}(\tilde{\alpha}_t^i)^T R^i \tilde{\alpha}_t^i - ((R^i)^{-1}\nabla v^i(\tilde{X}_t^i))^T R^i \tilde{\alpha}_t^i + \frac{1}{2}(\nabla v^i(\tilde{X}_t^i))^T(R^i)^{-1}\nabla v^i(\tilde{X}_t^i)$$
$$\geq 0,$$

Then,

$$\mathbb{E}^i[v^i(\tilde{X}_T^i) - v^i(\tilde{X}_0^i)]$$
$$\geq \mathbb{E}^i\Big[\int_0^T \Big[\text{tr}(\varsigma^i D^2 v^i(\tilde{X}_t^i)) + (\nabla v^i(\tilde{X}_t^i))^T A\tilde{X}_t^i$$
$$- \frac{1}{2}(\nabla v^i(\tilde{X}_t^i))^T(R^i)^{-1}\nabla v^i(\tilde{X}_t^i) - \frac{1}{2}(\tilde{\alpha}_t^i)^T R^i \tilde{\alpha}_t^i\Big]dt\Big]$$
$$= \mathbb{E}^i\Big[\int_0^T \Big[\text{tr}(\varsigma^i D^2 v^i(\tilde{X}_t^i)) - H^i(\tilde{X}_t^i, \nabla v^i(\tilde{X}_t^i)) - \frac{1}{2}(\tilde{\alpha}_t^i)^T R^i \tilde{\alpha}_t^i\Big]dt\Big].$$

Thus from HJB-FKP equation in (2.5),

$$\frac{1}{T}\mathbb{E}^i[v^i(\tilde{X}_T^i) - v^i(\tilde{X}_0^i)] \geq \frac{1}{T}\mathbb{E}^i\Big[\int_0^T \Big(\lambda^i - \tilde{f}^i(\tilde{X}_t^i; \boldsymbol{m}^{-i}) - \frac{1}{2}(\alpha_t^i)^T R^i \alpha_t^i\Big)dt\Big]$$
$$= \lambda^i - \frac{1}{T}\mathbb{E}^i\Big[\int_0^T \Big(\tilde{f}^i(\hat{X}_t^i; \boldsymbol{m}^{-i}) + \frac{1}{2}(\tilde{\alpha}_t^i)^T R^i \tilde{\alpha}_t^i\Big)dt\Big].$$

Letting $\liminf_{T\to+\infty}$ on both sides, we have

$$\lambda^i \leq \liminf_{T\to+\infty} \frac{1}{T}\mathbb{E}^i\Big[\int_0^T \Big(\tilde{f}^i(\tilde{X}_u^i; \boldsymbol{m}^{-i}) + \frac{1}{2}(\tilde{\alpha}_u^i)^T R^i \tilde{\alpha}_u^i\Big)du\Big],$$

for any admissible strategy $\alpha^i$. Next, we show that

$$J^i(\tilde{X}, [\tilde{\alpha}^{-i}; \tilde{\alpha}^i]) = \liminf_{T\to+\infty} \frac{1}{T}\mathbb{E}^i\left[\int_0^T \Big(f^i(\tilde{X}_u; m^{-i}) + \frac{1}{2}(\tilde{\alpha}_u^i)^\top R^i \tilde{\alpha}_u^i\Big)du\right], \tag{A.25}$$

which finishes the proof.

Recall that the admissibility of $\tilde{\alpha}^i$ implies the existence of a measure with density $\tilde{m}^i$ such that $\tilde{X}^i$ satisfies ergodicity with this measure, see (2.2). Moreover, Players $j \in [N]\setminus\{i\}$ use the TS algorithm strategy $\hat{\alpha}$, which as we have shown in Appendix A.8.1, their corresponding dynamics $(\hat{X}_t^j)_{t\geq 0}$ are ergodic with $m^j$. For brevity, use the notation $\tilde{X}^j = \hat{X}^j$ and $\tilde{m}^j = m^j$ for $j \in [N]\setminus\{i\}$. This way, the dynamics of all the players under the strategy profile $[\hat{\boldsymbol{\alpha}}^{-i}; \tilde{\alpha}^i]$ have the same notation, it

simplifies the writing below. Therefore,

$$\lim_{T \to \infty} \frac{1}{T} \mathbb{E}^i \Big[ \int_0^T \tilde{f}^i(\hat{X}_t^i; \boldsymbol{m}^{-i}) dt \Big]$$

$$= \int_{\mathbb{R}^d} \Big[ \int_{\mathbb{R}^{(N-1)d}} (\boldsymbol{x} - \overline{\boldsymbol{x}}_i)^T \mathcal{Q}_i (\boldsymbol{x} - \overline{\boldsymbol{x}}_i) \prod_{j \neq i} d\tilde{m}^j(x^j) \Big] d\tilde{m}^i(x^i)$$

$$= \int_{\mathbb{R}^{Nd}} (\boldsymbol{x} - \overline{\boldsymbol{x}}_i)^T \mathcal{Q}_i (\boldsymbol{x} - \overline{\boldsymbol{x}}_i) \prod_{j=1}^N d\tilde{m}^j(x^j)$$

$$= \sum_{j=1}^n \int_{\mathbb{R}^d} \Big[ (x^j - \overline{x}_i^j)^T Q_{jj}^i (x^j - \overline{x}_i^j) \Big] d\tilde{m}^j(x^j)$$

$$+ \sum_{j=1}^N \sum_{k \neq j} \int_{\mathbb{R}^{2d}} \Big[ (x^j - \overline{x}_i^j)^T Q_{jk}^i (x^k - \overline{x}_i^k) \Big] d\tilde{m}^j(x^j) d\tilde{m}^k(x^k)$$

$$= \lim_{T \to +\infty} \frac{1}{T} \mathbb{E}^i \Big[ \sum_{j=1}^N \int_0^T \Big[ (\tilde{X}_t^j - \overline{x}_i^j)^T Q_{jj}^i (\tilde{X}_t^j - \overline{x}_i^j) \Big] dt \Big]$$

$$+ \lim_{T \to +\infty} \frac{1}{T} \mathbb{E}^i \Big[ \sum_{j=1}^N \sum_{k \neq j} \int_0^T \Big[ (\tilde{X}_t^j - \overline{x}_i^j)^T Q_{lk}^i (\tilde{X}_t^k - \overline{x}_i^k) \Big] dt \Big]$$

$$= \lim_{T \to +\infty} \frac{1}{T} \mathbb{E}^i \Big[ \int_0^T \tilde{F}^i(\tilde{X}_t^1, \dots, \tilde{X}_t^N) dt \Big].$$

This shows that (A.25). $\qquad \square$

## A.9 NUMERICAL EXPERIMENTS

### A.9.1 SIMULATION DETAILS

We consider a system with $N = 10$ players, each with state dimension $d = 2$. The true dynamics matrix is shared across players and given by $A = -0.5 \cdot I_d$. We track the regret of a fixed player with $\text{id} = 3$. The initial state of this player is set to $X_0^3 = \begin{bmatrix} 0 \\ 0.5 \end{bmatrix}$. The prior belief over the unknown parameter matrix is Gaussian:

$$\mu_0 = \mathbf{0}_{d^2 \times 1}, \quad \Sigma_0 = 0.01 \cdot I_{d^2}.$$

Each player's diffusion matrix $\sigma^i$ is independently sampled as:

$$\sigma^i = 0.5 \cdot I_d + 0.05 \cdot Z^i, \quad Z^i \sim \mathcal{N}(0, I_d).$$

The mean-field estimate for each player is a random vector $\bar{x}^i \in \mathbb{R}^{Nd}$, sampled independently.

To introduce mild heterogeneity in cost structures, for each player $i$, the cost matrices are given by:

- $Q^i = I_{Nd} + \epsilon \cdot E^i$, where $\epsilon = 0.05$ and $E^i$ is a symmetric matrix with i.i.d. Gaussian entries;

- $R^i = I_d + \epsilon \cdot F^i$, where $F^i$ is similarly defined.

Simulations are run for a total of 5000 time steps with a discretization step size $\delta_t = 0.05$. To evaluate statistical properties, the simulation is repeated over 10, 50, and 100 independent runs. We report the cumulative regret $R(T)$ and its normalized version $R(T)/\sqrt{T + d\sqrt{T}}$.

In addition to this baseline setup, we conduct several additional experiments (e.g., higher dimensions, different prior distributions). The specific simulation details for each experiment are provided in their corresponding subsections.

### A.9.2 THE SCALING BEHAVIOR OF THE REGRET

We evaluate the empirical performance of our TS algorithm in the $N$-player stochastic differential game described in Section 3. We simulate the game with $N = 10, 20, 50$ players, where each player is assigned a different diffusion matrix $\sigma^i$ and a distinct cost structure $(Q^i, R^i)$. We focus on a single player and compute their cumulative regret over time.

To evaluate the scaling behavior and variance, we repeat the simulation for $n = 10, 20$, and $50$ sample paths. The results are shown in Figure 1: panel (a) plots $R(T)$ as a function of time $T$, and panel (b) shows the scaled regret $R(T)/\sqrt{T \log(T)}$. The shaded region indicates $0.2$ times the standard deviation of regret over sample paths at each time step.

As seen in Figure 3, the regret grows sublinearly with time, and the normalized regret $R(T)/\sqrt{T \log(T)}$ remains bounded as $T$ increases, in agreement with the theoretical upper bound $O(\sqrt{T \log(T)})$ established in Theorem 3.1.

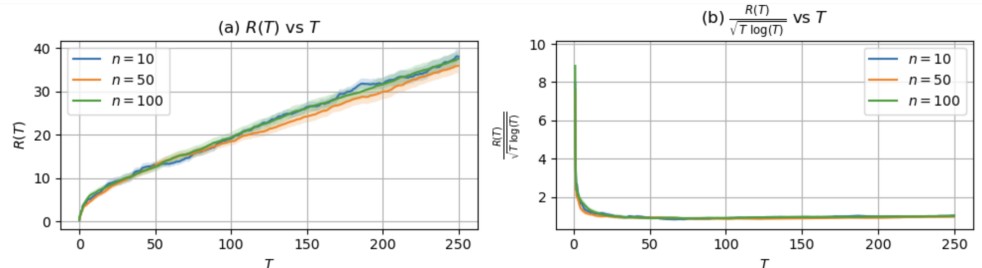

Figure 3: Regret vs time.

We also conduct additional experiments with a longer horizon $T = 1000$ (corresponding to 100,000 steps), using 10 sample paths, and report the results in Figure 4.

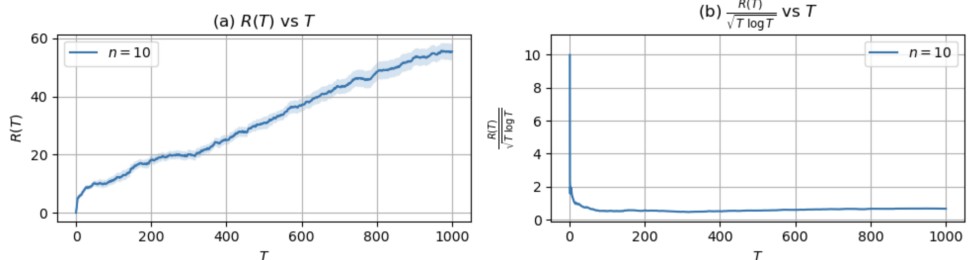

Figure 4: Regret vs time. ($T = 1000$)

### A.9.3 THE REGRET IN HIGHER DIMENSIONS

Similar to A.9.2, we evaluate the performance of the TS algorithm in higher dimensions. Specifically, we extend the experiments for $d \in \{2, 5, 10, 20\}$. While the raw regret $R(T)$ grows with $d$, the curves align after normalizing with respect to the factor $d\sqrt{T \log(T)}$. Results are shown in Figure 5.

### A.9.4 COMPARISON WITH CE AND BLIND SAMPLING

We compare our TS algorithm to two baselines. The first one is a posterior-mean controller that directly plugs the current posterior mean into the control law. It can perform slightly better in the very early stage, but TS consistently achieves lower cumulative regret over longer horizons. The second is Blind Sampling, which ignores data entirely and performs significantly worse. These results highlight the benefit of posterior-driven exploration. Detailed results are shown in Figure 6 and Figure 7.

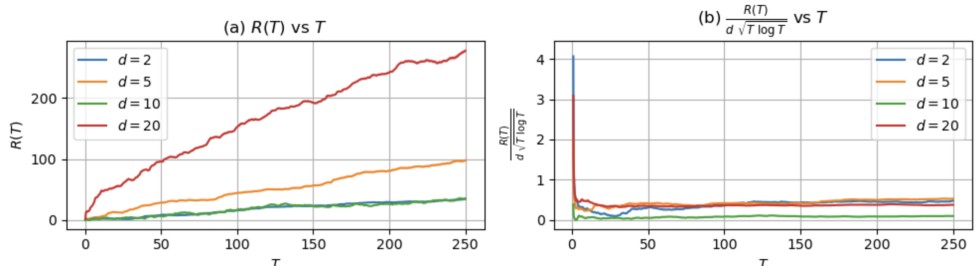

Figure 5: Regret vs time. (high dimensions)

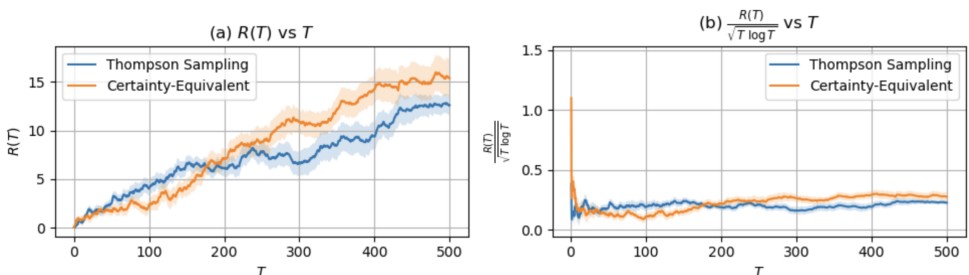

Figure 6: Regret vs time. (compared with CE)

### A.9.5   ROBUSTNESS TO PRIOR MISSPECIFICATION

In order to test the robustness of our algorithm to misspecification of the prior distribution, we run additional experiments with four prior families: Gaussian, t-distribution, exponential, and beta, all initialized with the same mean and covariance. For exponential and beta priors (defined element-wise), marginal parameters were chosen to match the target mean and variance. We tested both truncated priors (only samples yielding stable feedback matrices) and untruncated priors (no stability filter). Across all four families, the algorithm consistently achieves sublinear regret; heavy-tailed or asymmetric priors add early-stage variability but long-term behavior remains stable. The results are shown below in Figure 8.

### A.9.6   ABLATIONS ON HYPERPARAMETERS

We conduct a set of hyper-parameter ablation experiments on the prior distribution $\mathcal{N}(\mu_0, \Sigma_0)$.

First we vary the prior mean $\mu_0 = A_{true}, 0, 0.3 \cdot (1, 1, \ldots, 1)$, while keeping the covariance $\Sigma_0 = 0.01 \cdot I$ fixed. The results are shown in Figure 9. Across all settings, the normalized regret remains stable behavior, the misspecified priors also converge to the same scaling behavior as the well-specified prior.

We also vary the prior covariance matrix $\Sigma_0 = s_0^2 I$, with $s_0 = 0.1, 0.3, 1.0$, while fix the prior mean $\mu_0 = 0.3 \cdot (1, 1, \ldots, 1)$. The scaled regret curves also remain stable across all choices. The results are shown in Figure 10.

We also evaluate TS under non-isotropic prior covariance. In addition to the isotropic baseline $\Sigma_0 = 0.3^2 I$, we also consider (1) $\Sigma_0$ with diagonal entries $= 0.5$, off-diagonal entries $= 0.1$; (2) $\Sigma_0 = 0.3^2 I + 0.2^2 v v^T$, $v = (1, 1, \ldots, 1)^T$. The resulting regret curves are shown in Figure 11.

In all cases, the regret curve behaves stably and identically, which means the TS algorithm is robust to the hyper-parameters, matches our $O(\sqrt{T \log T})$ bound.

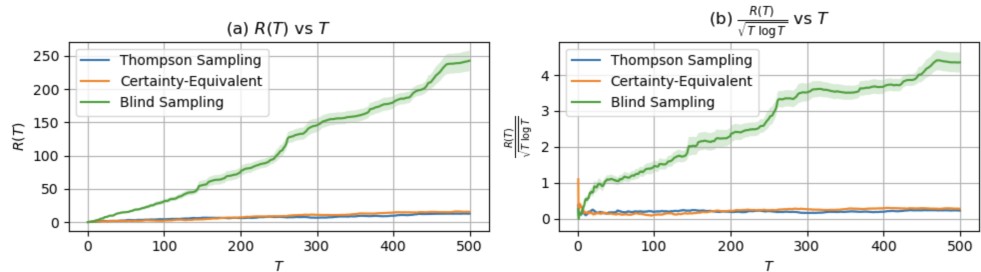

Figure 7: Regret vs time. (compared with Blind Sampling)

### A.9.7 Convergence rate to Nash equilibrium

We conduct a set of experiments to evaluate the empirical convergence rate to the Nash equilibrium. We report the following four quantities as functions of time $T$: (1) Parameter error: $\int_0^T \|A - A^i_{k(t)}\|^2 dt$; (2) State error: $\int_0^T \|X^i_t - \hat{X}^i_t\|^2 dt$; (3) Policy error: $\int_0^T \|(\alpha^i_t)^* - \hat{\alpha}^i_t\|^2 dt$; (4) Regret as in the paper. Each curve is plotted together with the theoretically predicted scaling established in our paper. As the parameter error only grows with the order of $\log T$, its behavior is more visible over a long time horizon; thus we extend the simulation interval to 10000. The results are shown in Figure 12. For the state error and the policy error, the empirical curves grow slower than the theoretical bounds, suggesting that the analysis is conservative and the true convergence rate may be sharper.

## A.10 Additional Details

### A.10.1 Usage of Large Language Model (LLM) for Manuscript Polishing

In preparing this manuscript, we used a large language model (LLM) only for the purpose of language polishing, grammar correction, and stylistic refinement. At no point was the LLM used to generate technical content, draft proofs, design experiments, interpret results, or propose new research directions.

All edits suggested by the LLM were reviewed, verified, and approved by the authors to ensure correctness, consistency, and scientific integrity. The authors take full responsibility for all content in the manuscript, including those parts refined via the LLM.

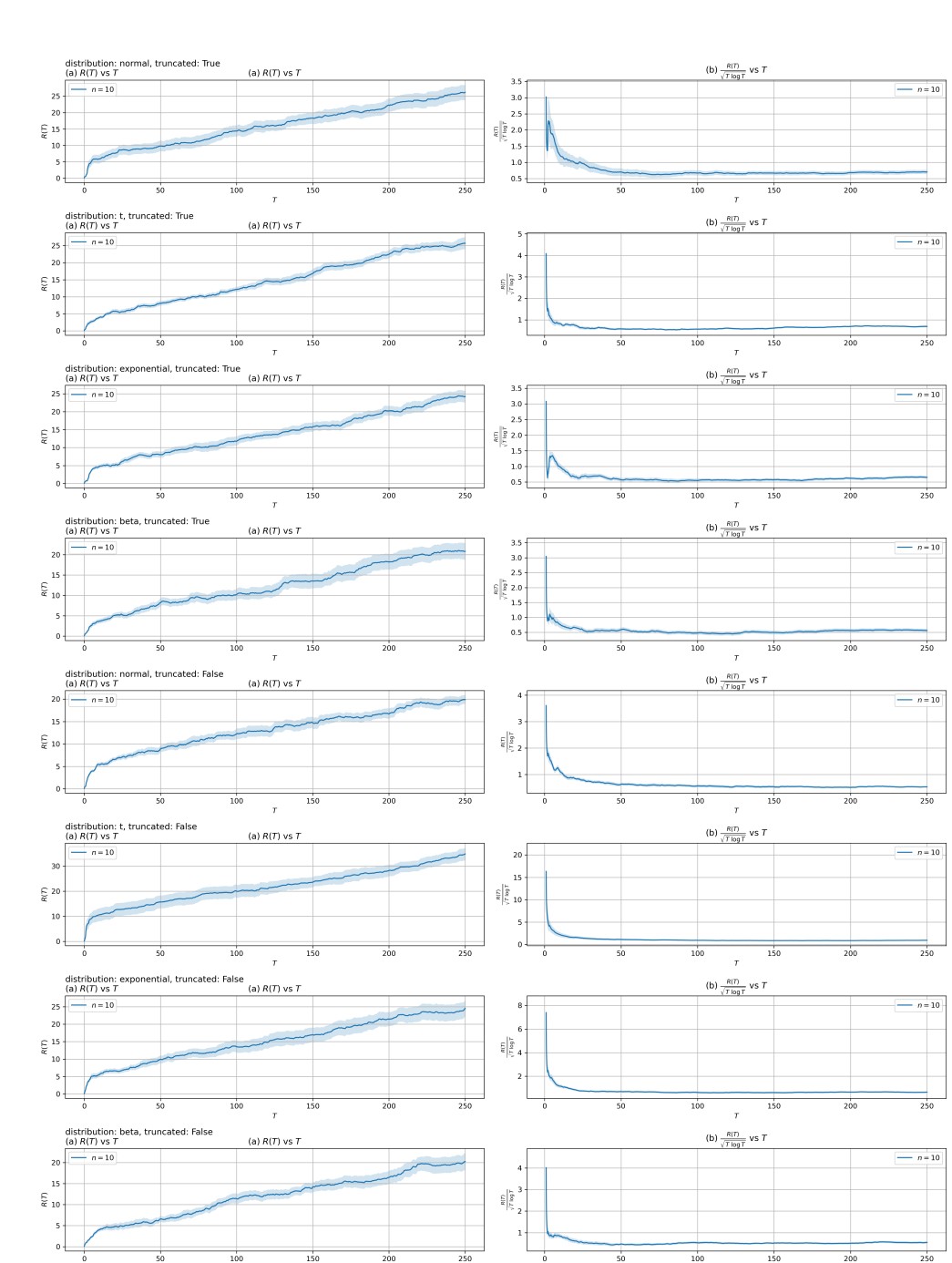

Figure 8: Regret vs time. (different prior distributions)

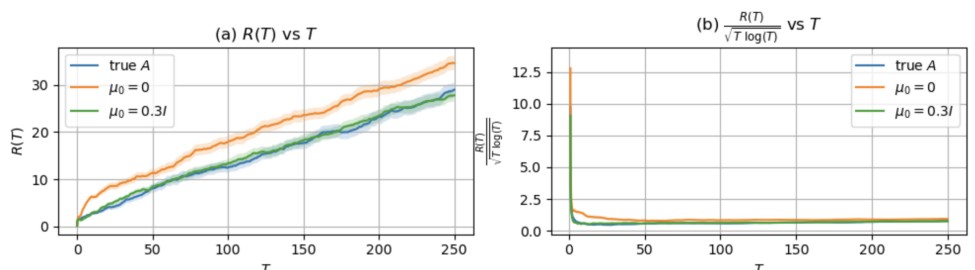

Figure 9: Ablation study on $\mu_0$

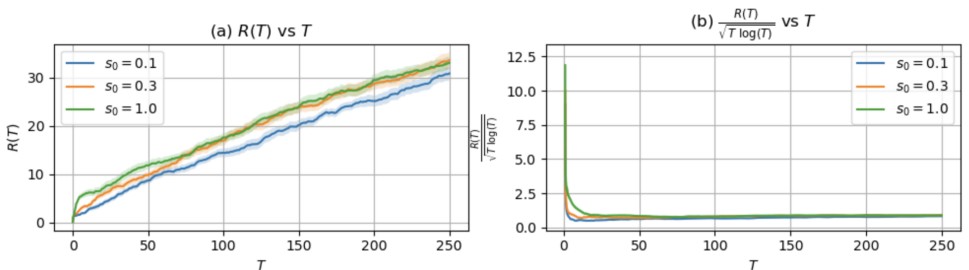

Figure 10: Ablation study on $\Sigma_0$

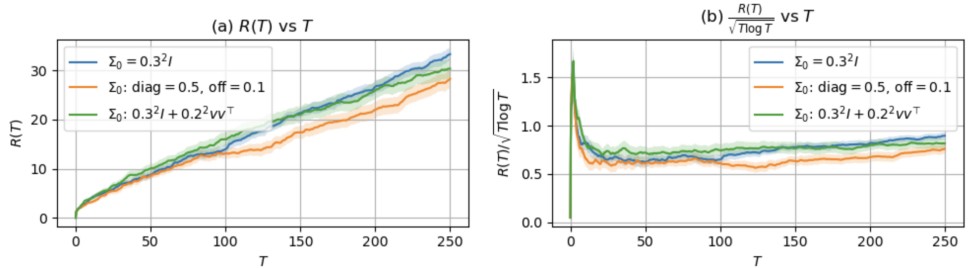

Figure 11: Ablation study on $\Sigma_0$

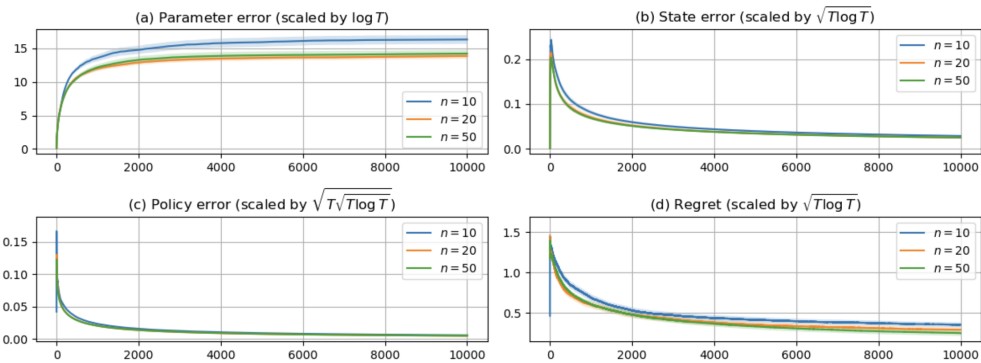

Figure 12: Convergence rate to Nash equilibrium

