# OpenReview forum: "Thompson Sampling Algorithm for Stochastic Games"
_ICLR.cc/2026/Conference — Submitted to ICLR 2026_

### Official Review · Reviewer_ywtJ · 2025-10-26

**Soundness:** 4
**Presentation:** 3
**Contribution:** 4
**Rating:** 8
**Confidence:** 3

**Summary:**

This paper extends Thompson Sampling (TS) to stochastic differential games in a linear-quadratic framework with ergodic cost. Bayesian regret of each player is proved to be of order $O(T\log T)$ which matches the best-known rates for single-player LQ control (up to logarithmic factors) and Nash equilibria can be found. Technically, the most challenging parts in this paper are two-folds: (1) each player has only access to their own states without overly idealized knowledge of the whole system; (2) the bound is independent of the number of players. The contribution of this paper is solid, the proofs are technically sound and the presentation is clear.

**Strengths:**

(1) This paper considers the partially obseravle setting, i.e., each agent only has access to its own states without knowledge of the whole system.

(2) The bound is independent of the number of players.

(3) The proofs are mathematically sound and well organized. The authors clearly justify their assumptions and discuss their appropriateness.

**Weaknesses:**

(1) It would be better to have more ablations on hyper-parameters in experiement part.

(2) It would be better to add experimental results on empirical convergence rate to Nash equilibria.

**Questions:**

(1) Is your regret bound tight in terms of dimension $d$ or other parameters?

(2) Is it possible to show last-iterate convergence rate to Nash equilibria?

---

> ### Author Response · Authors · 2025-11-21
>
> We thank the reviewer for their careful reading and detailed feedback. Below we respond to all comments. We address the weaknesses (W) first, followed by the questions (Q).
>
>
> *W1*: We have added the additional ablation studies on the hyper-paramters of the prior distribution $\mathcal{N}(\mu_0, \Sigma_0)$, and we include these results in the appendix (A.9.6) of the revised submission. If the reviewer is interested in additional hyper-parameter explorations, we would be happy to run further experiments and include them in the revised version.
>
> *W2*: Thank you for the helpful suggestion. We have added new experiments evaluating the empirical convergence rate to the Nash equilibrium in the appendix (A.9.7) of the revised submission.
>
> *Q1*: We do not know whether our regret bound is tight in the dimension $d$ or other parameters, what we can say is that it is reasonably competitive compared to existing literature. As in Gagrani et. al (2021), their regret scale is $\tilde{O}(\sigma^2 d^{1.5} \sqrt{T})$ ($\tilde{O}$ notation hides the $\log T$ factor). Our bound is $O \Big(\min(\|\sigma^i\|^3 \sqrt{d}, d) \sqrt{T \log(T)}\Big)$, which has a similar dependence on $T$ and is slightly tighter in moderately high dimensions. Nonetheless, establishing whether this dependence on $d$ is optimal remains open, and is an interesting direction for future work.
>
> *Q2*: Our analysis establishes a time-averaged convergence rate. My understanding of a last-iterate convergence rate is that it would characterize how the final state and control converge to the Nash equilibrium, as in our setting, $\mathbb{E}^i ||\hat{X}_T^i - X_T^i||^2$ and $\mathbb{E}^i ||\hat{\alpha}_T^i - (\alpha_T^i)^*||^2$. If this interpretation is inaccurate, we would be grateful if the reviewer could correct us.
>
> In contrast, our current analysis focuses on $\int_0^T \mathbb{E}^i |||\hat{X}_t^i - X_t^I||^2 dt$ and $\int_0^T \mathbb{E}^i ||\hat{\alpha}_t^i - (\alpha_t^i)^*||^2 dt$, which is a natural choice as the integral form is less sensitive and captures the accumulated discrepancy over time, while the last-iterate quantities can be more volatile.
>
> Obtaining such last-iterate guarantees is substantially more difficult, as one must control the instantaneous deviation between two diffusion process rather than the long-run averaged discrepancy. This typically would require some stronger stability and coupling arguments, which are beyond the scope of this paper. However, developing last-iterate convergence guarantees in this multi-player partial-information setting would be an interesting direction for future work.
>
>
> These are our responses to all comments raised by the reviewer. We are happy to address any further questions or suggestions.

---

### Official Review · Reviewer_6YKS · 2025-10-31

**Soundness:** 3
**Presentation:** 2
**Contribution:** 3
**Rating:** 6
**Confidence:** 3

**Summary:**

This paper studies an $N$-player nonzero-sum stochastic differential game with linear dynamics driven by a $d$-dimensional Brownian motion. Unlike most of the SDG literature, each player observes only their own state with a common unknown drift parameter. Thus uncertainty arises from both the state dynamics and the model parameters. Building on Bardi & Priuli (2014), they embed affine state-feedback strategies in a continuous-time TS policy for the LQ SDG. With an episode stopping rule, they prove a Bayesian regret bound of order $O\big(d||\sigma^i\||^3\sqrt{T\log T}\big)$, matching known TS rates in LQ control in terms of $T$. They also show the policy constitutes a Nash equilibrium. The experiments align with the theory.

**Strengths:**

* Proposes a continuous-time TS policy with adaptive episode lengths for an ergodic LQ SDG under partial information, where all players share an unknown drift parameter $A$ and learn it solely from their own state processes. The paper proves a Bayesian regret bound of order $O(\sqrt{T\log T})$ and establishes a corresponding equilibrium result.
* Experimental results support the theory, further demonstrating performance in higher-dimensional settings and robustness to prior misspecification.

**Weaknesses:**

* The assumption of a common unknown prior is somewhat restrictive. However, given the first focus on a partial-information SDG with both equilibrium and sublinear regret guarantees, it may be acceptable within the current problem setup.
* Section 2 is difficult to follow with the current explanation (especially for readers familiar with TS but not with SDGs like me). Providing more intuition about the parameters and their roles would improve accessibility.
* Some notation appears before being defined or is left undefined (see Question 1). Clarifying notation order and completeness would enhance readability.

**Questions:**

1. In Section 2, the authors use a norm notation $|| \cdot ||$. In Bardi & Priuli (2014) this notation denotes the spectral norm, i.e., the largest eigenvalue in the positive-semidefinite case. However, in Section 3 the same symbol is defined as the Frobenius norm. Please clarify whether the norm in Section 2 (especially in Assumption A4) refers to the Frobenius or the spectral norm.
   1.1. The notation $[\alpha^{-i}, \alpha]$ is defined in line 210 (p. 4), but it first appears on page 3. The definition should appear before its first use.
   1.2. In Proposition 3.2, the symbol $:=^{i}$ is used without definition. It likely means $:=E^{i}$; please make this explicit.

2. From Proposition 3.3, the obtained regret seems to scale as $O\big(\min(\|\sigma^{i}\|^{3}\sqrt{d},d)\sqrt{T\log T}\big)$, which is smaller than the form stated in Theorem 3.1. Can you clarify why Theorem 3.1 presents the looser bound? was it for simplicity, or are there technical reasons for preferring that form?

3. What is the main difficulty or bottleneck when extending the analysis to heterogeneous parameters $A^i$? I would (naively) expect the resulting regret to include $N$-dependent constants in such cases. The authors’ perspective on this point would be valuable.

### Comments

1. L75: the authors appear to cite Abeille & Lazaric (2018) as a paper studying the *Bayesian* regret of TS. However, this reference actually analyzes the *frequentist* regret of TS in linear–quadratic (LQ) control settings (as far as I understand).

2. There are duplicated references (e.g., Abeille & Lazaric 2018a,b; Bardi & Priuli 2014a,b). Please review the reference list carefully and remove any unintentional duplicates.

### Typos

1. L692: missing “*” in $Q$.
2. L1182-L1185: missing “(” before $(\sigma)$.
3. L1193: missing $1/2$ factor.
4. L1211: missing “(” before $\Lambda^i$.

---

> ### Author Response · Authors · 2025-11-21
>
> We thank the reviewer for their careful reading and detailed feedback. Below we respond to all comments. We address the weaknesses (W) first, followed by the questions (Q).
>
> *W1 \& Q3*: We interpret the first comment in weaknesses as referring to the assumption that all players share the same unknown drift matrix $A$, since the prior distribution itself is not identical across players. If this interpretation is inaccurate, we would be grateful if the reviewer could correct us.
>
> Under this interpretation, the comment is closely related to Q3.  Extending the analysis to heterogeneous drift parameter $A^i$ introduces some conceptual and technical challenges. There are several different conditions:
>
> (1) If players cannot observe each other's states, then they only interact through the cost functional. They observes their own trajectory and cannot learn other players' parameters from their observations, which means beliefs about other players' dynamics never concentrate.
>
> (2) If players do observe each other's states, then the problem becomes much more interesting and challenging. In this setting, each player need to infer not only its own drift parameter, but also others' $A^j$, and this would require modeling each other's strategy and belief process. The underlying model of full-information heterogeneous $A^i$ SDG has, to the best of our knowledge, not been studied in the literature, which makes it harder for the learning problem to be built on top of the baseline. This problem of hererogeneous $A^i$ is truly an intriguing direction for future work, but would require some new tools to address these difficulties.
>
> *W2*: We thank the reviewer for this helpful suggestion, we agree that additional intuition would improve the accessibility of Section 2. In the updated version, we have added a short explanatory paragraph that clarifies the meaning of parameters right after the HJB-KFP equation (Equation (2.5)). We hope this improves the readability of the section. If there are other parts of Section 2 that remain unclear, we would be happy to address them further.
>
> *W3 \& Q1*: In Section 2, all matrix norms refer to the Frobenius norm. Since expect for the one in Assumption A.4, the remaining norms apply to vectors, for which the Frobenius and spectral norms coincide. Assumption A.4 is newly introduced, which does not appear in Bardi \& Priuli (2014), to prove Lemma A.3. We also thank the reviewer for the two additional points, the corresponding corrections have been made in the revised version.
>
> *Q2*: The regret bound in Theorem 3.1 was intentionaly stated in a simplied form to emphasize the overall $\sqrt{T \log T}$ dependence. However, the reviewer is correct that Proposition 3.3 yields a sharper bound of $O \left( \min\big( \|\sigma\|^3 \sqrt{d}, d \big) \sqrt{T \log T} \right)$, and we have updated the statement of Theorem 3.1 to reflect this more precise bound.
>
> *Q3*: See W1.
>
> *Comments \& Typos*: We thank the reviewer for the careful and detailed reading, and we have revised the manuscript accordingly. In particular,  Abeille \& Lazaric (2018) has been removed from the Bayesian regret examples. We have also cleand up the duplicated references and corrected all listed typos.
>
> These are our responses to all comments raised by the reviewer. We are happy to address any further questions or suggestions.

---

> > ### Comment · Reviewer_6YKS · 2025-11-24
> >
> > Thank you for the answers. Most of my concerns have been resolved, and I have also read the discussion with the other reviewers.
> >
> > I agree with the concern raised by Reviewer fXfr regarding the fit of this paper.
> > I understand that the paper studies a partial-feedback extension of the full-information game considered in Bardi and Priuli, which, as the authors mentioned, serves as a bridge between stochastic differential games and multi-agent RL.
> > However, while the manuscript provides detailed explanations of previously obtained results, it does not clearly convey this bridging perspective to readers of ICLR.
> > A more general introduction to stochastic differential games, rather than focusing on specific existing results, would improve the accessibility of the paper.
> > For this reason, I will keep my current score.

---

> > > ### Author Response · Authors · 2025-11-24
> > >
> > > Thank you for raising this point about the fit. We agree that clarifying the bridging perspective would strengthen the paper.
> > >
> > > In response, we have added several references in the introduction to guide readers toward classical works on general stochastic differential game (SDG) theory. we have also added a paragraph to the introduction that highlights the connection between our SDG formulation and multi-agent reinforcement learning, and clarifies the role of our setting as a continuous-time multi-agent learning problem. We hope these additions make the positioning and motivation of the paper clearer for the ICLR audience.
> > >
> > > We believe ICLR is an appropriate venue because the community is increasingly seeking theoretical foundations for RL, especially in continuous-time and multi-agent settings. Our goal is precisely to fill this gap by showing how tools from stochastic differential games can support rigorous learning algorithms. In particular, the framework we study is closely connected to multi-agent reinforcement learning, and our results could potentially benefit colleagues working on multi-agent RL from a theoretical perspective.
> > >
> > > Importantly, our contribution is not another extension of SDG theory, but a learning algorithm with regret guarantees in an SDG environment. The mathematical results serve as tools to establish the performance of the algorithm, rather than being the main focus of the paper. Submitting this work to a control journal would shift the emphasis toward existence and uniqueness results, whereas here the central object is the learning procedure itself.

---

### Official Review · Reviewer_MFdF · 2025-10-31

**Soundness:** 4
**Presentation:** 3
**Contribution:** 4
**Rating:** 8
**Confidence:** 2

**Summary:**

This paper propose a Thompson sampling method for solving stochastic different games (with linear–quadratic dynamics) with homogeneous player setting in a Bayesian inference framework. The authors show that the proposed method enjoys state of the art regret bound and leads to Nash equilibrium.

**Strengths:**

1. This paper propose a novel approach for solving multi-player SDGs using Thompson Sampling, extending prior work's approach.
2. The framework also relax assumptions in previous work on independence of the players.
3. The results on Nash equilibrium is very interesting and potentially impactful to the field.

**Weaknesses:**

I am not in this field, so my feedback might be limited and please correct me if I am wrong. That said, there are a few of my concerns
1. The scope of this work is quite limited. The authors assume there is no coupling between the players from the dynamics side, but only through costs. Although the authors motivates the scenarios in the intro, but the applicability of such framework remains elusive, and quite restrictive.
2. The tools the authors used seem to be borrowed from prior works. I.e., the adaptive episode length idea, the continuous time framework, and TS itself is extensively studies in the literature. So what's the true contribution of this work, besides attaching all the techniques together?

**Questions:**

See above.

---

> ### Author Response · Authors · 2025-11-21
>
> We thank the reviewer for their careful reading and detailed feedback. Below we respond to all comments, denoting the weaknesses as W1 and W2.
>
> *W1*: We want to clarify that in our model, the underlying processes are coupled. The players dynamics do not explicitly depend on other players' states, however, they share the same unknown drift matrix $A$, which makes their underlying dynamics strongly coupled. In addition, as the referee points out, they are strategically coupled via their cost functionals. This setup is natural when agents operate in the same environment but do not directly influence each other's dynamics. In addition, even in this seemingly ``less-coupled'' case, the learning problem is non-trivial.
>
> We acknowledge that several alternative modeling frameworks can be considered beyond the current setting. For instance, one may study (i) games with common noise and/or (ii) games with closed-loop controls, in which players can observe the dynamics of others. However, addressing these extensions requires substantially more technical machinery. Our present work serves as a natural first step toward analyzing technically more involved cases such as (i), and exploring these directions remains an important avenue for future research.
>
> Regarding (ii), we emphasize that allowing players to observe each other’s dynamics leads to an immediate and significant increase in analytical complexity, and the ergodic formulation would require a fundamentally new approach. One of the authors is currently working on this extension. Notably, under our current framework, we obtain a clean Bayesian regret bound that remains independent of the number of players, a structural advantage that may be lost in more general settings.
>
> *W2*: We agree that our work builds on and bridges several ideas from existing literature. However, we would like to stress that our contribution is not simply pasting exsiting techniques, but to develop new results and proof arguments in the cases of ergodic N-player LQ game with an unknown drift, which has not been studied before. Concretely, we list a few examples where our analysis is completely new:
> - The control of the propagation of the discrepancy $||A-\hat{A}||$ is essential for the analysis of both the regret and the Nash Equilibrium. Our proof of this argument is novel, and this has not been done even in the discrete case. Our intuition comes from the $1/t$ growth rate of this propagation in dimension one. However, it turns out to be much more difficult in multi-dimension to obtain such bounds, as we need to carefully control the growth of the trace of the posterior variance matrix. We had to bound the smallest eigenvalue uniformly using concentration inequalities. See our proof of Proposition 3.4 for details (Appendix A5, P24).
> - The idea of ``adaptive episode lengths'' is similar to previous control papers, but the second part regarding the Nash Equilibrium is completely new. In fact, the proof of the Nash equilibrium (Theorem 3.2) is, to the best of our knowledge, the first equilibrium result in competitive LQ games with partial information. And to prove these results, we needed new techniques that were not used in previous work.
>
>
> These are our responses to all comments raised by the reviewer. We are happy to address any further questions or suggestions.

---

### Official Review · Reviewer_fXfr · 2025-11-01

**Soundness:** 3
**Presentation:** 3
**Contribution:** 3
**Rating:** 4
**Confidence:** 3

**Summary:**

This paper treats the question of equilibrium learning in general, non-zero-sum stochastic differential games, that is, games where the players' payoff depends on the actions chosen at each instance of time and a state that evolves following a stochastic differential equation (an open-loop control).

An equilibrium strategy in this context is an admissible selection of actions in $\mathbb{R}^d$ over time which is maximizes unilaterally the long-run (Cesàro) average payoff of each player over time. This is in turn characterized by an associated ensemble of Hamilton-Jacobi-Bellman equations, a set of nonlinear partial differential equations that provide necessary and sufficient conditions for unilateral optimality.

The specific state evolution model of the authors is of the form
$$
dX(t) = (AX(t) - \alpha(t)) \, dt + \sigma\, dW(t)
$$
where $X(t)$ is the state of the game at time $t$, $\alpha(t)$ is the players' action profile at time $t$, $A$ is a coupling matrix, and $W(t)$ is an ordinary Brownian motion. Under certain assumptions for $A$ and the (quadratic) cost functions of the game, it is known by prior work that the associated set of HJB equations admits a unique solution (also of quadratic form).

The authors consider the case where the coupling matrix $A$ is unknown, so the players cannot solve the associated HJB equations. Their main contribution is an episodic Thompson sampling (TS) algorithm which starts by sampling from an initial Gaussian prior distribution and proceeds episode-by-episode by observing the state and updating the posterior following a specific stopping criterion. Based on this algorithm, the authors are able to show that (i) the players' regret is bounded as $\mathcal{O}(\sqrt{T \log T})$; and (ii) the proposed algorithmic policy is a Nash equilibrium with probability $1$. The authors also validate these results through numerical simulations in games with 10, 50 and 100 players.

**Strengths:**

I am not sufficiently knowledgeable in stochastic differential games to provide an expert opinion but, as far as I could tell, the authors' analysis is sound, and the positioning of their contributions in the surrounding literature is fair. The contribution itself seems in line with what could be expected from a good paper in the field.

**Weaknesses:**

My main concern with this paper is its thematic alignment with ICLR. Even though I could easily see this paper published in a top-tier control venue (IEEE CDC, TAC or SIOPT SICON), the fit with ICLR is very slim. This would be less of an issue if the field of (stochastic) differential games were more accessible from a technical standpoint but, as it currently stands, the paper's technical content and contributions would only be accessible to an infinitesimally thin slice of ICLR's generalist audience—even its more theoretical side.

My "reject" recommendation reflects precisely this: it should not be taken as a criticism of the paper's technical content (which I cannot assess at the level of a dedicated expert), but as an assessment of the suitability of this paper for ICLR as a whole.

**Questions:**

None (see above), except for a remark: the paper is not about *stochastic* games in the sense of Shapley (also referred to as Markov games in ML), but stochastic *differential* games. This should be made clear in the title, as the two fields are quite disjoint (Ι was definitely expecting something different coming in from the title).

---

> ### Author Response · Authors · 2025-11-21
>
> We thank the reviewer for their careful reading and detailed feedback. Below we respond to all comments. We address the weaknesses (W) first, followed by the questions (Q).
>
> *W1*:  We appreciate the reviewer's comment regarding the fit of our work to the venue, and agree that our work would, indeed, fit well in journals such as SICON. At the same time, we do believe that the paper is strongly connected to the reinforcement learning theory theme at ICLR. We submit to the current venue for two reasons:
> - The central focus of our work is learning in stochastic differential games with unknown dynamics. We propose a numerical scheme and show a Bayesian regret bound as well as a Nash Equilibrium characterization of this learning based policy. In this sense, our contribution is not just a control/game-theoretic existence result, but in a learning result with a concrete construction. We believe that our work integrates classical stochastic games into multi-agent reinforcement learning.
> - Our work bring tools from the classic control, filtering, and differential games community to answer questions central to ICLR: how agents should learn and react in unknown environments. We hope we can help bridge the field of stochastic control and reinforcement learning: the colleagues in the learning theory community can take advantage of these existing tools while studying similar problems, and we can bring ideas and new questions, tools, and techniques to the control-theoretic community as well.
>
> *Q1*:  We thank the reviewer for this helpful remark and we agree with the concern. Our work is indeed in stochastic differential games (SDGs), and we refer to SDGs in the abstract and the introduction. We will update the title (if allowed) and make this clarification in the introduction.
>
> These are our responses to all comments raised by the reviewer. We are happy to address any further questions or suggestions.

---

> > ### Comment · Reviewer_fXfr · 2025-11-25
> >
> > Dear authors,
> >
> > Thank you for your reply.
> >
> > While I appreciate the technical content of your paper (to the extent that I was able to position it in the relevant literature, on which I am not an expert), I do not agree that its setting is sufficiently close to the question of "how agents should learn and react in unknown environments" in the way this is interpreted e.g., in RL (or other areas of ICLR).
> >
> > SDG models are highly stylized and specific, and albeit very rich from a mathematical viepoint, the end results tend to reflect their narrow starting point. This is also the case for the current paper: In the end, I do not see any concrete link (even once removed) between your paper and the core audience of ICLR, and the high cost of entry of your paper's mathematical formalism (SDEs, PDEs, martingale limit theory, ...) would, I believe, make it inaccessible to said audience.
> >
> > I have also read the other reviews and your responses. I do not agree with your assessment that "submitting this work to a control journal would shift the emphasis toward existence and uniqueness results", as there is a vast number of similarly themed papers that have appeared in control-theoretic venues—both journals and conferences.
> >
> > All in all, my suggestion to submit your paper to a top-tier control venue is just that, a suggestion, made in the spirit of what I would consider the best fit for your paper. On the other hand, on the question of whether I find this paper suitable for ICLR, my assessment remains that it isn't.
> >
> > I would not wish to lower my score, as I would consider this unfair to the overall quality of your work. However, in terms of thematic alignment, the current paper does not quite provide a bridge between the SDG literature and ICLR's centers of interest, so I remain on the "reject" side of the equation.
> >
> > Regards,
> >
> > Reviewer fXfr

---

> > > ### Author Response · Authors · 2025-11-25
> > >
> > > Dear reviewer,
> > >
> > > Thank you very much for your follow-up and for acknowledging the technical quality of our work. We fully respect your final assessment. Since your concerns focus on the thematic alignment with ICLR, we hope to clarify how we see the positioning of the paper in the learning-theory landscape, and how our perspective differs.
> > >
> > > **1. On accessibility and mathematical formalism.**
> > >
> > > We understand your concern that the SDE- and PDE-based presentation may raise the entry barrier for some readers. In our revision, we have already added intuition and reorganized parts of the exposition to reduce this burden. At the same time, we note that several active research areas at ICLR, such as diffusion models, continuous-time RL, and SDE-based optimization, routinely use the same mathematical tools. Our work builds on this existing trend: it studies learning and exploration in a stochastic dynamical environment, but in a continuous-time multi-agent setting. We therefore believe that the formalism, while technical, remains compatible with current ICLR directions.
> > >
> > > **2. On stylization and the relationship to learning in unknown environments.**
> > >
> > > We completely agree that SDG models are stylized, and our intention is not to dispute this. Rather, we wish to clarify that our contribution does not lie in solving a classical SDG: the agents face an unknown drift matrix, maintain Bayesian posteriors, and adapt their policies through Thompson Sampling. This is directly analogous to learning unknown dynamics in RL, but in continuous time and within a competitive multi-agent environment. The game-theoretic structure is essential for our results on equilibrium and learning dynamics, yet the novelty lies in analyzing how agents learn and adapt when the underlying dynamics of this stochastic game are unknown.
> > >
> > > Stylized abstractions are fundamental to learning theory, as multi-armed bandits, LQRs, and MDPs are all simplified models that allow researchers to isolate the essential challenges of exploration and adaptation. Our model follows this tradition by providing a structured stochastic environment in which the learning dynamics can be rigorously analyzed. In this sense, the SDG structure serves as the environment for the learning problem, rather than the goal of the paper.
> > >
> > > To make this clear, in the revision we added references to classical SDG works as well as recent RL-inspired multi-agent learning formulations, and a dedicated paragraph explaining how our setting aligns with continuous-time RL and multi-agent learning theory.
> > >
> > > **3. On venue suitability.**
> > >
> > > We genuinely appreciate your suggestion about control-theoretic venues. Our motivation for submitting to ICLR is that the core contributions (posterior updates, regret bounds, equilibrium guarantees, and multi-agent learning under partial information) belong to the learning-theory side rather than to classical control analysis. Control-theoretic venues also study systems with unknown dynamics; however, the emphasis in those works tends to be on establishing properties of optimal controls or value functions under structural assumptions. In contrast, our paper adopts a learning-theoretic perspective: the central objects are posterior updates, Thompson sampling behavior, regret, and equilibrium under uncertainty, which are much closer in spirit to RL, MARL, and learning in dynamical systems. Our goal is to contribute to this line of research with a rigorous continuous-time counterpart.
> > >
> > > We thank you again for your thoughtful perspective. We will continue improving the exposition to make the work clearer and more accessible, and we hope that the clarified positioning helps convey why we view the learning-theory track at ICLR as an appropriate venue for this line of results.

---

### Official Review · Reviewer_YWJY · 2025-11-03

**Soundness:** 3
**Presentation:** 2
**Contribution:** 2
**Rating:** 4
**Confidence:** 3

**Summary:**

This paper studies an $N$-player linear–quadratic stochastic differential game with ergodic cost where the drift matrix $A$ is unknown but common to all players. Each player observes only their own state, maintains a Bayesian posterior over  $A$ with Gaussian prior, and plays using a Thompson Sampling policy with adaptive episode lengths. The authors prove a Bayesian regret bound $\mathcal{O}(\sqrt{T}\log T)$ for each player, which is independent of $N$. They show that the TS policy profile constitutes a Nash equilibrium. They provide a continuous-time posterior update (Proposition 3.1), decompose the regret into sampling/strategy/mismatch terms (Proposition 3.2) and bound each (Proposition 3.3), plus a coupling argument establishing equilibrium (Theorem 3.2). Experiments simulate regret scaling and show normalization by $\sqrt{T}\log T$ remains bounded.

**Strengths:**

* The paper proves the Bayesian regret bound for ergodic $N$-player games, matching best-known orders for LQ control while not depending on number of players $N$.
* Propose Thomspon sampling algorithm to handle the unknown $A$ setting, proving the Nash equilibrium of the TS profile under additional stability conditions.

**Weaknesses:**

* The model is based on the previous work Bardi & Priuli (2014a). This paper handles the setting where the matrix $A$ is unknown but does not explain why this setting is meaningful in practice.
* This paper focus on the theoretical side, the author use a full section (Section 2) to introduce existing results but does not claim their technique contribution over previous work.

**Questions:**

* The numerical experiment section is confusing. It lack explanation to help the reader understand how the experiment is conducted.
* Assumption 3.1 (2) seems non-trival, can you provide concrete examples or experiments to verify this assumption?
* The proof technique seems standard, please explain the technique contribution.

---

> ### Author Response · Authors · 2025-11-21
>
> We thank the reviewer for their careful reading and detailed feedback. Below we respond to all comments. We address the weaknesses (W) first, followed by the questions (Q).
>
> *W1*: Our formulation is the natural partial-information extension of the Bardi-Priuli game. The drift $A$ is not directly observable and must be inferred from observing the trajectories of the agents. We want to point out that this setting is, in fact, more realistic than the case where $A$ is known. It is a very strong assumption to assume that we have the full knowledge of the distributional information of the whole underlying system. In addition, it is natural to assume that agents in the same system are driven by some shared but unknown features, and that these must be learned over time. Partial-information extensions of stochastic games arise, for example, in investment problems with unknown profit characteristics, or in auction/market settings with unobservable value components.
>
> In the introduction (P2, L56), we already mention a concrete application in the ergodic setting where the decision makers observe some common unknown factor and optimize their cost functions while updating their beliefs regarding the unknown. For example, minimizing one's long-term average electricity cost given certain regulatory or market conditions. In the revised version, we will highlight this example and make it explicit that it directly motivates our partial-information extension of the Bardi–Priuli framework.
>
> *W2*: Section 2 is intended to make the paper self-contained by recalling the game structure and characterization of Nash Equilibria in the Bardi-Priuli setting, as well as the assumptions necessary for the current setup. We want to emphasize that we formulate the game in a competitive setting. In order to deal with the common unknown drift, we apply filtering techniques and propose a Thompson-like numerical scheme. To the best of our knowledge, our work provides one of the first approaches in a competitive game setting that yields an equilibrium and a sublinear regret bound, extending beyond single-agent optimal control (Gagrani et al (2021)), cooperative games (Ouyang et al (2020)), and full-information games (Bardi-Priuli (2014)). In addition, this regret bound is independent of the number of agents.

---

> > ### Author Response · Authors · 2025-11-21
> >
> > *Q1*: In the current version, some details of the numerical experiments are only given briefly due to the page limit. We have added a short introductory paragraph and given the full parameter specification in the appendix (A.9.1) in the revised version, and explicitly point to it in the main text. We hope that these additions will make it much easier for readers to understand how the experiments are done and how to reproduce them. We are happy to provide any additional details the reviewer finds helpful.
> >
> >
> > *Q2*: Assumption 3.1(2) enforces exponential decay to assure the stability of the following TS algorithm, which is the continuous-time equivalent of Ouyang et al. (2020, Assumption 2). Related stability assumptions, though not identically formulated, are standard in the literature on LQ control and learning in continuous time. In fact, in the numerical experiment A.9.5, we observe that the scheme often remains stable even in settings where Assumption 3.1(2) is not strictly satisfied. This suggests that one of the goals in the future is perhaps working on removing or weakening the current assumption, but it might require more work to show stability. However, in the present work we retain Assumption 3.1(2) as a convenient sufficient condition ensuring the stability needed for our theoretical analysis, and we view this as a natural first step toward a more general stability theory for learning in SDGs.
> >
> > *Q3*: The main technical novelty of our work is to extend Thompson Sampling to a competitive continuous-time N-player stochastic differential game under partial information, while relaxing key structural assumptions on the unknown drift compared to prior works. In this setting, we establish a sublinear Bayesian regret bound of order independent of the number of players, and prove that the online learning strategies constitute a Nash equilibrium-a guarantee not previously achieved for posterior-based learning in continuous-time stochastic games.
> >
> > While the structure of the regret bound seems similar to our references, several key technical ingredients in our proofs are different. In particular, (i) the proof of the Nash equilibrium (Theorem 3.2) is, to the best of our knowledge, the first equilibrium result in competitive LQ games with partial information. (ii) The control of the propagation of the discrepancy $||A-\hat{A}||$ is essential for the analysis of both the Bayesian regret and the Nash Equilibrium. Our proof of this argument is novel, and this has not been done even in the discrete case. Our intuition comes from the $1/t$ growth rate of this propagation in dimension one. However, it turns out to be much more difficult in multi-dimension to obtain such bounds, as we need to carefully control the growth of the trace of the posterior variance matrix. We had to bound the smallest eigenvalue uniformly using concentration inequalities. See our proof of Proposition 3.4 for details (Appendix A5, P24).
> >
> > These are our responses to all comments raised by the reviewer. We are happy to address any further questions or suggestions.

---

### Author Response · Authors · 2025-12-02

Dear Area Chair,

Thank you for handling our submission. Below is a brief summary of the main improvements made in our rebuttal and revised manuscript.

**1. Improved motivation and accessibility.**

We clarified why learning an unknown drift is meaningful, added concrete motivating examples, and expanded the introduction with additional general references. Section 2 was significantly improved with clearer intuition around the HJB–KFP system, better explanation of parameters, and a fully cleaned-up notation flow.

**2. Clearer positioning and contribution.**

Reviewers asked us to clarify which parts of the analysis are genuinely new and how our work goes beyond existing TS, control, and SDG literature. We made explicit that the paper provides:

	•	the first competitive continuous-time N-player partial-information SDG with both NE characterization and sublinear Bayesian regret;

	•	new control of posterior drift discrepancy using eigenvalue concentration;

	•	the first equilibrium proof in this competitive partial-information setting.

These clarifications were incorporated directly into the revised text.

**3. Expanded and clarified experiments.**

To address requests for clarity and empirical support, we expanded the experiment section by adding:

	•	a detailed description of the experimental setup and parameters (A.9.1),

	•	new hyper-parameter ablations (A.9.6), and

	•	new experiments on empirical convergence to Nash equilibrium (A.9.7).

These additions significantly improve transparency, allow full reproducibility, and strengthen the empirical validation of the main theoretical results.

**4. Strengthened theoretical clarity.**

We updated Theorem 3.1 to reflect the sharper bound pointed out by the reviewers and clarified the role of Assumption 3.1(2). We also thank the reviewers for their helpful comments regarding notation and typos; all identified issues have now been corrected in the revised manuscript.

**5. Clarified connection to learning and venue relevance.**

In response to concerns about fit, we added a dedicated paragraph explaining how the work connects to continuous-time RL and multi-agent learning (posterior updates, Thompson sampling, regret, equilibrium). This was incorporated without changing the technical scope.

Overall, the revision substantially improves clarity, experimental completeness, and the explicit presentation of the paper’s contributions.

Thanks,

Authors of submission 14044

---

### Meta-Review · Area_Chair_Xatr · 2026-01-07

**Summary:**

- This is a theoretical paper on stochastic differrential game in a linear quadratic framework with a Gaussian prior on the drift. The paper provides a Thompson sampling based method with a proved Bayesian regret bound and a Nash equilibrium.

- Five reviews were collected. Three reviewers were positive with scores of 8, 8, 6. Two reviewers are negative: both giving scores of 4.

- Three reviewers (with scores 8,6,4)  raise concerns about the fit of this paper with ICLR. After carefully reading the paper and the discussions, I am inclined to agree with these concerns. The authors didn't provide a convincing justification of why this paper is well-aligned with the scope of ICLR either in the paper or during the discussion. The connection with continuous time RL and multi-agent learning is rather vague because this paper focuses on linear quadratic games and no experimental results were provided on more general settings beyond LQ games that could potentially be more related to ICLR's scope.

- Moreover, the two reviewers who gave the highest scores 8 either explicitly note that they are not an expert in this field, or only provide a very brief comments focussing on missing experimental details rather than commenting on the core theoretical contributions. This further suggests that the primary audience for this work may not be the ICLR community. Besides, this limits the weight placed on these highest-score reviews when I make the final decision.

- As noted by Reviewer fXfr, the paper appears more suitable for control-oriented venues such as TAC or L4DC. I believe the paper will receive both higher-quality feedback and a better-aligned audience in these venues.

**Reviewer Concerns:**

Reviewers' concerns on missing experiment details and confusing technical explanantions are successfully addressed in the rebuttal.

**Unsolved concerns**
- The major concerns on the alignment of this paper with the scope of ICLR is not resolved successfully. The authors could either provide more evidence on why this paper fits with ICLR by adding more general simulation experiments or submitting this paper to control venues.

- Reviewer YWJY's concern on Assumption 3.1 could be better addressed by providing some concrete examples.

- Several reviewers question the novelty of the proof techniques because they seem to be borrowed from prior work. The rebuttal didn't provide a satisfying explanation on the novelty.

**Reviewer Scores:**

- Reviewer YWJY (score 4) is not likely to change the score.
- Reviewer fXfr (score 4) is not going to change the score according to the discussion.
- Reviewer MFdF (score 8) may decrease the score because the rebuttal didn't address the two major concerns successfully: the limited scope of this work and the lack of novelty in the technical tools.
- Reviewer 6YKS (score 6) may decrease the score because they agreed with Reviewer fXfr on the poor fit of this paper with ICLR after participating in the discussion.
- Reviewer ywtJ (score 8) may decrease or keep the score because this reviewer only questions the experiment setting instead of the core theoretical part.

---

### Decision · Program_Chairs · 2026-01-26

Reject